# TRAINABLE TRANSFORMER IN TRANSFORMER

## ABSTRACT

Recent works attribute the capability of in-context learning (ICL) in large pre-trained language models to implicitly simulating and fine-tuning an internal model (e.g., linear or 2-layer MLP) during inference. However, such constructions require large memory overhead, which makes simulation of more sophisticated internal models intractable. In this work, we propose a new efficient construction, *Transformer in Transformer* (in short, TINT), that allows a transformer to simulate and fine-tune more complex models during inference (e.g., pre-trained language models). In particular, we introduce innovative approximation techniques that allow a TINT model with less than 2 billion parameters to simulate and fine-tune a 125 million parameter transformer model within a single forward pass. TINT accommodates many common transformer variants and its design ideas also improve the efficiency of past instantiations of simple models inside transformers. We conduct end-to-end experiments to validate the internal fine-tuning procedure of TINT on various language modeling and downstream tasks. For example, even with a limited one-step budget, we observe TINT for a OPT-125M model improves performance by $4 - 16\%$ absolute on average compared to OPT-125M. These findings suggest that large pre-trained language models are capable of performing intricate subroutines. To facilitate further work, a modular and extensible codebase for TINT is included.

## 1 INTRODUCTION

Transformers (Vaswani et al., 2017) have brought about a revolution in language modeling, and scaling model size has enabled significant advancements in capabilities (Brown et al., 2020; Chowdhery et al., 2022). One such capability (Wei et al., 2022) is in-context learning (ICL), where language models "learn" from given training exemplars in the context and subsequently predict the label of a test example within a single inference pass.

Several works (Akyurek et al., 2022; Dai et al., 2022; von Oswald et al., 2022) propose that ICL occurs when the large ("simulator") model mimics and trains a smaller and simpler auxiliary model —such as a linear or 2-layer MLP model— on the in-context data. A crucial limitation of previous works is the large number of parameters needed for a simulator to perform such a complex subroutine during its forward pass, which restricts the simulator to performing very few training steps on fairly simple models. For example, simulating training of a linear layer can require tens of millions of parameters (Akyurek et al., 2022), and extending the simulator to train a larger model would require a simulator with trillions of parameters.

The current work shows that minor modifications to the standard transformer architecture allow it to efficiently simulate and approximately train an internal *auxiliary* transformer during a single inference pass (Section 2). We call our architecture *Transformer in Transformer*, or TINT in short. We show how TINT can internally simulate and train several popular and capable Transformer models such as GPT (Radford et al., 2019), OPT (Zhang et al., 2022), and other variants (Touvron et al., 2023; Scao et al., 2022). TINT requires fewer than two billion parameters to internally simulate and update an auxiliary transformer with 125 million parameters (e.g., GPT-2 or OPT-125M). The scale of our construction is crucial to its significance, as it suggests that even transformers of moderate scale can explicitly learn from context during inference. TINT can also be understood as a meta-learning architecture (Finn et al., 2017; Kirsch et al., 2022). As our experiments show, TINT can adapt a small pre-trained model on-the-fly using just the first few tokens of a given context.

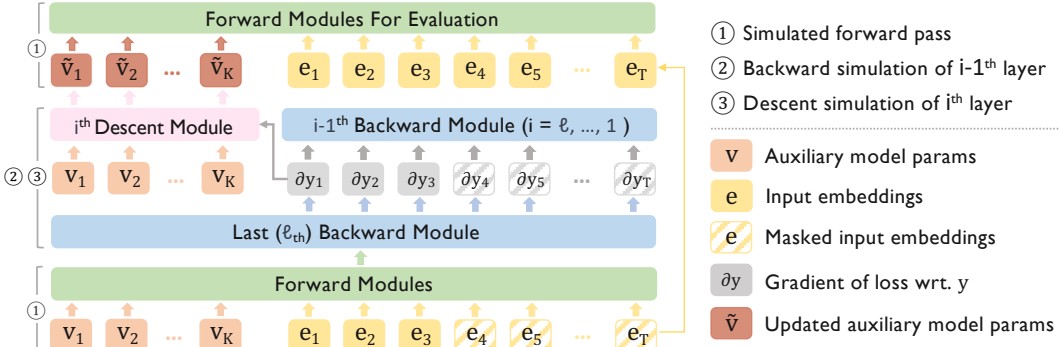

Figure 1: The overall structure of TINT. Each Forward, Backward, and Descent module is represented using combinations of linear, self-attention, layernorm, and activation layers. The input consists of prefix embeddings, that represent relevant auxiliary model parameters in each layer, input token embeddings, and a binary prefix mask to separate the train and evaluation segments of the input. Auxiliary model parameters are updated in the descent module using the training part, and the updated prefix tokens are transferred to the forward modules via residual connections for evaluating the rest.

We validate our approach with end-to-end experiments on many language modeling and downstream tasks. Results demonstrate that a TINT model constructed to simulate and tune an OPT-125M model leads to a perplexity reduction of 0.3 to 0.7 points in language modeling. Additionally, TINT learns from in-context exemplars in the few-shot setting, resulting in an absolute gain of 12% to 16% over the auxiliary model. TINT can also learn from the context tokens of the evaluation inputs in the zero-shot setting, leading to an absolute performance improvement of up to 4% when no explicit exemplars are provided (Section 3.2). To the best of our knowledge, TINT is the first simulator to undergo such a comprehensive end-to-end evaluation on standard language tasks. In contrast, previous studies primarily conducted probing tests on transformers pre-trained using synthetic datasets or lacked empirical validation (Akyurek et al., 2022; von Oswald et al., 2022; Giannou et al., 2023), likely due to the immense scale required by their constructions.

To summarize, our work improves over prior works in three crucial ways.

1. **Expressiveness**: We allow the auxiliary model to be a complex, general-purpose transformer, which requires significant technical innovation beyond past work using linear and MLP auxiliary models. TINT can internally simulate and train popular moderate- and large-scale transformers, including OPT (Zhang et al., 2022), LLaMA (Touvron et al., 2023), BLOOM (Scao et al., 2022), and Pythia (Biderman et al., 2023).

2. **Efficiency**: We focus on making TINT orders of magnitude smaller, reducing the construction size from trillions of parameters to a few billion, despite simulating much complex auxiliary models. This efficiency is driven by a number of approximations (see Section 2.2).

3. **Empirical Validation**: TINT can match the performance of fine-tuning the auxiliary model explicitly, thereby validating the approximations used in the construction. In addition, we provide an extensible codebase for further training and evaluation of our proposed architecture (Section 3).

## 2 OUR CONSTRUCTION

### 2.1 OVERVIEW

We design a transformer architecture, dubbed TINT, to perform the forward, backward, and parameter update operations of a smaller so-called *auxiliary* model over the course of a single inference pass (Figure 1). The first few layers of TINT simulate the forward pass, and then backpropagation and gradient updates are performed in parallel in the next several layers of the model. The final layers of TINT simulate another forward pass through the auxiliary model with the updated parameters, and TINT directly outputs the result. This procedure requires that TINT has (1) read and write access to the auxiliary model weights, and (2) read access to labelled training data.

For (1), we read from and write to the token embeddings of the first few tokens in the input (i.e., *prefix embeddings*, see Definition 2.1 and Section 2.3). For (2), we designate the first few input tokens as training data. If the input contains in-context demonstrations, then TINT performs a natural operation: training on in-context demonstrations and testing on the last example. However, the input can also be standard text tokens without additional formatting or labelling, in which case TINT performs gradient descent to adapt to the first few tokens in a context before being evaluated on the rest of the context. This procedure is known as dynamic evaluation, which we describe in further detail in Section 3.1.

## 2.2 KEY COMPONENTS

Due to space constraints, we defer a complete description of the TINT to the appendix. Here, we detail the modifications and approximations introduced in our architecture to enable efficient simulation of training an internal auxiliary model. Experiments in Section 3 verify that these approximations do not harm performance. Numbers in parentheses indicate the parameter saving factor when relevant.

1. **Prefix embeddings** ($5\times$ compared to Wei et al. (2021); Perez et al. (2021)): As described in Section 2.3, we use the token embeddings of the first few inputs (i.e., the *prefix*, see Definition 2.1) to represent the relevant auxiliary model weights at each layer.

2. $H_{\text{sim}}$**-split linear operations** ($H_{\text{sim}}\times$): We parallelize expensive linear operations by splitting them across attention heads (Section 2.4).

3. **Linear attention**: We use linear attention modules to perform the forward, backward, and gradient operations for an auxiliary model linear layer. Softmax attention also suffices but requires more parameters and incurs an approximation error (Theorem 2.5). We use softmax attention modules to simulate the auxiliary model attention modules in TINT.

4. **First order gradients** ($4\times$): We use the first-order term of the gradient for layer normalization and activation functions (Section 2.5).

5. **Only train the value vectors of the attention** ($5\times$): We only update and backpropagate through the value vectors of the attention layers. TINT is designed to simulate only a few steps of training the auxiliary model, so we expect this approximation not to drastically modify the final performance. Our experiments in Tables 1 and 2 validate that it does not hurt performance. Moreover, in Theorem 2.13, we formally show that under certain conditions, backpropagating through just the value vectors can be arbitrarily accurate to standard backpropagation.

6. **Parameter sharing** ($3\times$ or $4\times$): We save $3\times$ parameters by applying the same forward module in TINT to simulate the query, key, and value computation of the auxiliary model's self-attention module (Section 2.7). Similarly, we divide the feedforward layer in the auxiliary model into 4 sub-layers and save $4\times$ parameters by employing a single TINT module to simulate each sub-layer.

**Notation:** Let $D$ denote the embedding dimension for a token and $T$ denote the length of an input sequence. $H$ denotes the number of attention heads. With the exception of contextual embeddings, we use subscripts to indicate if the quantity is from TINT or from the auxiliary model. For example, $D_{\text{aux}}$ refers to the embedding dimension and $D_{\text{sim}}$ refers to the TINT embedding dimension. For contextual embeddings, we use $e_t^{(\ell)} \in \mathbb{R}^{D_{\text{sim}}}$ to denote activations in TINT and $x_t^{(\ell)} \in \mathbb{R}^{D_{\text{aux}}}$ to denote activations in the auxiliary model, where $\ell$ is the layer and $t$ is the sequence position. When convenient, we drop the superscript that represents the layer index and the subscript that represents the position index. For a matrix $A$, $a_j$ refers to its $j$th row, and for any vector $b$, $b_j$ refers to its $j$th element. TINT uses one-hot positional embeddings $\{p_i^{\text{TINT}} \in \mathbb{R}^{T_{\text{sim}}}\}_{i \leq T_{\text{sim}}}$. For illustration, we ignore the bias parameters here but discuss them in the appendix.

## 2.3 OPERATING ON AN AUXILIARY MODEL WITH PREFIX EMBEDDINGS

The straightforward way to simulate the forward pass of the auxiliary model would be to store its weights in the simulator's weights and run a forward pass as usual. However, this gives the simulator no way to update the weights of the auxiliary model, since the simulator cannot modify its own weights during a forward pass. The only way to update the auxiliary model weights is by storing them in model *activations* that can be accessed and modified over the course of a forward pass.

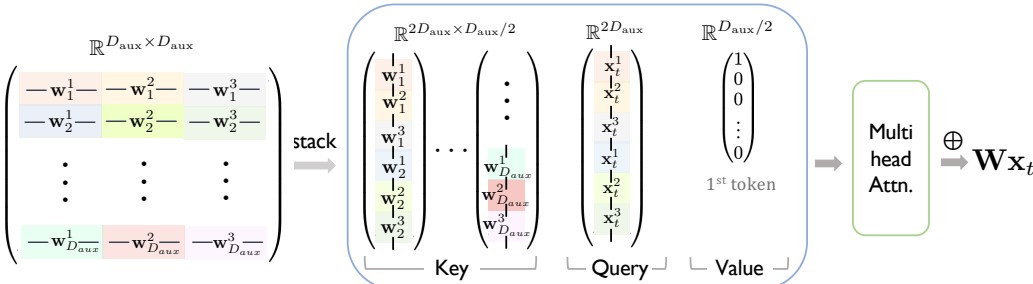

Figure 2: TINT simulates the forward pass of a linear layer as a $H$-head ($H = 6$ here) attention layer, with parameters of the auxiliary model as the key, the encodings of input tokens as the query, and the positional one-hot vector of the prefix embeddings as the value. We omitted the identical transformation for key, query, and value matrices for simplicity.

Wei et al. (2021); Perez et al. (2021) model the simulator after a Turing machine, where each $e_t^{(\ell)} \in \mathbb{R}^{D_{\text{sim}}}$ acts as a workspace for operations, and data is copied between workspaces and memory using attention operations. Memory can either be allocated in a token embedding, thereby increasing the embedding size $D_{\text{sim}}$ (Akyurek et al., 2022), or passed into the model as additional tokens, thereby increasing the simulator's input sequence length $T_{\text{sim}}$. Both strategies increase the size of the construction, and using attention modules for copy operations results in a drastic scaling. For example, if $D_{\text{aux}} = 768$, a dot product with weight $\boldsymbol{w} \in \mathbb{R}^{768}$, i.e. $\langle \boldsymbol{w}, \boldsymbol{x}_t^{(\ell)} \rangle$, requires at least $8.7$ million parameters in the simulator[1].

Alternatively, storing memory as context tokens and allowing the attention modules to attend to those tokens removes the need for copying operations (Giannou et al., 2023). Then, a dot product with weight $\boldsymbol{w} \in \mathbb{R}^{768}$, i.e. $\langle \boldsymbol{w}, \boldsymbol{x}_t^{(\ell)} \rangle$, requires $1.7$ million parameters only. Naive implementation is problematic since the TINT attention module grows quadratically with the sequence length $T_{\text{sim}}$. We thus define *prefix embeddings* in TINT to contain only the relevant auxiliary parameters at each layer.

**Definition 2.1** (Prefix Embeddings). $\{\boldsymbol{v}_j^{(\ell)}\}_{j=1}^K$ denotes the $K$ prefix embeddings at the $\ell$th layer in TINT. These contain relevant auxiliary model weights or simulated activations.

Prefix embeddings permit efficient parallelizaton across attention heads while keeping the embedding dimension $D_{\text{sim}}$ small, as we demonstrate below.

## 2.4 STACKING IN PREFIX-TOKENS, $H_{\text{SIM}}$-SPLIT LINEAR OPERATIONS AND LINEAR ATTENTION

We motivate three parameter-efficient techniques using a $D_{\text{aux}} \times D_{\text{aux}}$ linear layer as a case study. The linear layer is applied token-wise, so we consider a single position $t$ without loss of generality.

**Definition 2.2** (Linear layer). For a weight $\boldsymbol{W} \in \mathbb{R}^{D_{\text{aux}} \times D_{\text{aux}}}$, a linear layer takes $\boldsymbol{x} \in \mathbb{R}^{D_{\text{aux}}}$ as input and outputs $\boldsymbol{y} = \boldsymbol{W}\boldsymbol{x}$.

**Stacking:** We compute $\langle \boldsymbol{w}_i, \boldsymbol{x}_t \rangle$ for all $i \in [D_{\text{aux}}]$, where $\boldsymbol{w}_i$ denotes the $i$th row of $\boldsymbol{W}$. To do so, the TINT input embedding $e_t$ must contain $\boldsymbol{x}_t$ in its first $D_{\text{aux}}$ coordinates, and the weights $\{\boldsymbol{w}_i\}$ are in prefix embeddings $\{\boldsymbol{v}_j\}$ (Definition 2.1). A first attempt is to put each $\boldsymbol{w}_i$ in its own $\boldsymbol{v}_i$ vector, which requires $K = D_{\text{aux}}$ prefix embeddings at the start of the sequence. For GPT-2, $D_{\text{aux}} = 768$, so this strategy will not allow the TINT to accept many standard language context tokens; moreover, the attention modules in the TINT will grow quadratically with input length. Therefore, we stack $S$ weights on top of each other to form each prefix embedding $\boldsymbol{v}_i$. $S$ drives a trade-off between the embedding dimension of the TINT, $D_{\text{sim}} := D_{\text{aux}}S$, and the context length to the TINT, $T_{\text{sim}} := K + T_{\text{aux}}$. We set $S = 4$.

---

[1]The copy attention will require 1.7 million pdarameters, while the dot product with a feedforward module (following Akyurek et al. (2022)) will require $> 7$ million parameters.

**Attention Module:** We use a self-attention module in TINT to compute the matrix-vector product. We modify the usual attention layer to also include the one-hot position embeddings $\{\boldsymbol{p}_i^{\text{TINT}} \in \mathbb{R}^{T_{\text{sim}}}\}_{i \leq T_{\text{sim}}}$. Here, we use a single attention head (see Definition A.1 for multi-head attention).

**Definition 2.3** (TINT self-attention with single head). For parameters $\{\boldsymbol{W}_Q^{\text{TINT}}, \boldsymbol{W}_K^{\text{TINT}}, \boldsymbol{W}_V^{\text{TINT}} \in \mathbb{R}^{D_{\text{sim}} \times D_{\text{sim}}}\}$, $\{\boldsymbol{W}_Q^p, \boldsymbol{W}_K^p, \boldsymbol{W}_V^p \in \mathbb{R}^{D_{\text{sim}} \times T_{\text{sim}}}\}$, the self-attention layer with single attention head and a function $f_{\text{attn}} : \mathbb{R}^{T_{\text{sim}}} \to \mathbb{R}^{T_{\text{sim}}}$ takes a sequence $\{\widehat{\boldsymbol{e}}_t \in \mathbb{R}^{D_{\text{sim}}}\}_{t \leq T_{\text{sim}}}$ as input and outputs $\{\widetilde{\boldsymbol{e}}_t \in \mathbb{R}^{D_{\text{sim}}}\}_{t \leq T_{\text{sim}}}$, such that

$$\widetilde{\boldsymbol{e}}_t = \sum_{j \leq T_{\text{sim}}} a_{t,j} \boldsymbol{v}_j, \qquad \text{where } a_{t,j} = f_{\text{attn}}(\boldsymbol{K}\boldsymbol{q}_t)_j,$$

$$\boldsymbol{q}_t = \boldsymbol{W}_Q^{\text{TINT}}\widehat{\boldsymbol{e}}_t + \boldsymbol{W}_Q^p \boldsymbol{p}_t, \qquad \boldsymbol{k}_t = \boldsymbol{W}_K^{\text{TINT}}\widehat{\boldsymbol{e}}_t + \boldsymbol{W}_K^p \boldsymbol{p}_t, \qquad \boldsymbol{v}_t = \boldsymbol{W}_V^{\text{TINT}}\widehat{\boldsymbol{e}}_t + \boldsymbol{W}_V^p \boldsymbol{p}_t \text{ for all } t \leq T_{\text{sim}},$$

and $\boldsymbol{K} \in \mathbb{R}^{T_{\text{sim}} \times D_{\text{sim}}}$ is the key matrix defined with its rows as $\{\boldsymbol{k}_t\}_{t \leq T_{\text{sim}}}$.

$\boldsymbol{q}_t, \boldsymbol{k}_t, \boldsymbol{v}_t$ are the query, key, and value vectors at position $t$, and $a_{t,j}$ is the attention score between tokens at position $t$ and $j$. $f_{\text{attn}}$ can be either linear or softmax functions, and the corresponding layers are referred to as linear and softmax self-attention respectively.

*Remark* 2.4. In order to compute and backpropagate the loss during inference, the self-attention layers in TINT need to be non-causal on the first few tokens of the input.[2] Explicit masks apply bidirectional attention to the input to backpropagate auxiliary self-attention layers (Section 2.6).

**TINT Linear Forward module (Figure 2):** A first attempt would be to use different attention heads to operate on different rows; however, this uses $S$ attention heads whereas large transformers usually have many more heads. Moreover, the output from the multi-head attention would need to be reorganized by a $D_{\text{sim}} \times D_{\text{sim}}$ linear layer before it could be reduced efficiently via summation. We instead parallelize across more attention heads to ensure the resulting output can easily be compiled: crucially, we shard each individual weight into $S'$ parts. We set $S$ and $S'$ such that $H_{\text{sim}} = S \times S'$, using all available heads to parallelize the dot products.

**$H_{\text{sim}}$-split linear operations (Appendix B.1):** The output resulting from the attention module has shape $(D_{\text{sim}}/H_{\text{sim}}) \times H_{\text{sim}}$ and is sparse. To complete linear forward pass, we need to sum and aggregate the appropriate terms to form a $D_{\text{sim}}$-length vector with $\boldsymbol{W}\boldsymbol{x}$ in the first $D_{\text{aux}}$ coordinates. Straightforwardly summing along an axis aggregates incorrect terms, since the model was sharded. Rearranging the entire matrix to enable easy summation requires an additional $D_{\text{sim}} \times D_{\text{sim}}$ linear layer. But we can save a $H_{\text{sim}} \times$ parameters by leveraging the local structure of the attention output. We space out the results across the $D_{\text{sim}}/H_{\text{sim}}$ rows and then sum along the $H_{\text{sim}}$ columns to get the desired $D_{\text{sim}}$-length vector. This requires $D_{\text{sim}}^2/H_{\text{sim}} + D_{\text{sim}}H_{\text{sim}}$ parameters. Note that leveraging local structure also compresses the constructions for the TINT's backpropagation modules of layer normalization and activation operations (Appendices D and E).

**Linear Attention:** The above construction uses a linear attention mechanism instead of the canonical softmax. Here, we show that any linear attention module with bounded entries can be approximated by softmax attention with a few additional parameters. Thus, we often use linear attention in TINT.

**Theorem 2.5** (Informal, c.f. Theorem A.2). *For any $\epsilon > 0$, $B > 0$, and a linear attention module with $H$ heads and bounded parameters, there exists a softmax attention module with $2H_{sim}$ attention heads and $4\times$ additional parameters, such that on every sequence of inputs with $B$-bounded norm, the output sequences of the softmax attention and the linear attention differ by $\mathcal{O}(\epsilon)$ at each position.*

## 2.5 FIRST ORDER GRADIENTS FOR LAYER NORMALIZATION

Below, we show that computing exact gradients for layer normalization is expensive, so we efficiently approximate backpropagation by computing the dominating term.

**Definition 2.6.** [Layer Normalization] Define a normalization function $f : \mathbb{R}^d \to \mathbb{R}^d$ that performs $f(\boldsymbol{x}) = (\boldsymbol{x} - \mu)/\sigma$, where $\mu$ and $\sigma$ are the mean and standard deviation of $\boldsymbol{x}$, respectively. Then, layer normalization with parameters $\gamma, \boldsymbol{b} \in \mathbb{R}^{D_{\text{aux}}}$ takes as input $\boldsymbol{x} \in \mathbb{R}^{D_{\text{aux}}}$ and outputs $\boldsymbol{y} \in \mathbb{R}^{D_{\text{aux}}}$, which is computed as $\boldsymbol{z} = f(\boldsymbol{x}), \boldsymbol{y} = \gamma \odot \boldsymbol{z} + \boldsymbol{b}$.

---

[2]Similar prefix models have been developed in (Raffel et al., 2020; Liu et al., 2018).

**Definition 2.7.** [Exact Gradient for Layer Normalization] Using notations in Definition 2.6, given the gradient of the loss w.r.t the output of the Layer Normalization $\partial_{\boldsymbol{y}}$, backpropagation computes $\partial_{\boldsymbol{x}}$ as

$$\partial_{\boldsymbol{x}} = (\partial_{\boldsymbol{z}} - D_{\text{aux}}{}^{-1} \sum_{i=1}^{D_{\text{aux}}} \partial_{z_i} - \langle \partial_{\boldsymbol{z}}, \boldsymbol{z} \rangle \boldsymbol{z})/\sigma \qquad \partial_{\boldsymbol{z}} = \gamma \odot \partial_{\boldsymbol{y}}.$$

Exact backpropagation is expensive because $\langle \partial_{\boldsymbol{z}}, \boldsymbol{z} \rangle \boldsymbol{z}$ requires using at least two sequential MLPs. We thus approximate it with a first-order Taylor expansion, which is entry-wise close to the true gradient.

**Definition 2.8.** [$\epsilon$-approximate Layer Normalization Gradient] With notations defined above, this layer takes $\partial_{\boldsymbol{y}}, \boldsymbol{x} \in \mathbb{R}^{D_{\text{aux}}}$ as input and outputs $\widehat{\partial_{\boldsymbol{x}}} = \frac{1}{\epsilon}(f(\boldsymbol{x} + \epsilon \gamma \odot \partial_{\boldsymbol{y}}) - f(\boldsymbol{x}))$.

**Theorem 2.9** (Informal, c.f. Thm D.1). *With bounded $\ell_2$-norms of $\boldsymbol{x}, \partial_{\boldsymbol{y}}, \gamma, \boldsymbol{b}, \left\| \partial_{\boldsymbol{x}} - \widehat{\partial_{\boldsymbol{x}}} \right\|_{\infty} \leq \mathcal{O}(\epsilon)$.*

This approximation only works for symmetric Jacobian, so it does not apply when backpropagating the linear and self-attention layers. We use a TINT module with 2 linear layers, separated by Group Normalization (Wu and He, 2018), to compute $\widehat{\partial_{\boldsymbol{x}}}$, resulting in a $4\times$ parameter reduction.

## 2.6 BACKPROPAGATION THROUGH ATTENTION VALUE VECTORS

For simplicity, we show a single head self-attention layer (multi-head attention is in Appendix C). The self-attention in the auxiliary model is the same as in TINT (Definition 2.3) without a position vector.

**Definition 2.10** (Auxiliary model softmax self-attention). A self-attention layer with parameters $\{\boldsymbol{W}_Q, \boldsymbol{W}_K, \boldsymbol{W}_V\}$ takes a sequence $\{\boldsymbol{x}_t\}_{t \leq T_{\text{aux}}}$ and outputs a sequence $\{\boldsymbol{y}_t\}_{t \leq T_{\text{aux}}}$, such that

$$\boldsymbol{y}_t = \sum_j a_{t,j} \boldsymbol{v}_j, \qquad \text{with } a_{t,j} = \text{softmax}(\boldsymbol{K} \boldsymbol{q}_t)_j, \quad \boldsymbol{q}_t = \boldsymbol{W}_Q \boldsymbol{x}_t, \quad \boldsymbol{k}_t = \boldsymbol{W}_K \boldsymbol{x}_t, \quad \boldsymbol{v}_t = \boldsymbol{W}_V \boldsymbol{x}_t,$$

for all $t \leq T_{\text{aux}}$, and $\boldsymbol{K} \in \mathbb{R}^{T_{\text{aux}} \times D_{\text{aux}}}$ defined with rows $\{\boldsymbol{k}_t\}_{t=1}^{T_{\text{aux}}}$.

**Definition 2.11** (Exact gradient for softmax self-attention). Given the gradients of the loss w.r.t the output sequence $\{\partial_{\boldsymbol{y}_t}\}_{t=1}^{T}$, backpropagation computes $\{\partial_{\boldsymbol{x}_t}\}_{t=1}^{T}$, with

$$\partial_{\boldsymbol{x}_t} = \boldsymbol{W}_Q^{\top} \partial_{\boldsymbol{q}_t} + \boldsymbol{W}_K^{\top} \partial_{\boldsymbol{k}_t} + \boldsymbol{W}_V^{\top} \partial_{\boldsymbol{v}_t}, \qquad \partial_{\boldsymbol{v}_t} = \sum_j a_{j,t} \partial_{\boldsymbol{y}_j},$$

$$\partial_{\boldsymbol{q}_t} := \sum_j a_{t,j}((\partial_{\boldsymbol{y}_t})^{\top} \boldsymbol{v}_j)[\boldsymbol{k}_j - \sum_{j'} a_{t,j'} \boldsymbol{k}_{j'}], \qquad \partial_{\boldsymbol{k}_t} := \sum_j a_{t,j}((\partial_{\boldsymbol{y}_t})^{\top}(\boldsymbol{v}_j - \sum_{j'} a_{t,j'} \boldsymbol{v}_{j'}))\boldsymbol{q}_j$$

for all $t \leq T_{\text{aux}}$, with $\boldsymbol{K} \in \mathbb{R}^{T_{\text{aux}} \times D_{\text{aux}}}$ defined with rows $\{\boldsymbol{k}_t\}_{t=1}^{T_{\text{aux}}}$.

The computation of $\partial_{\boldsymbol{q}_t} := \sum_j a_{t,j}((\partial_{\boldsymbol{y}_t})^{\top} \boldsymbol{v}_j)[\boldsymbol{k}_j - \sum_{j'} a_{t,j'} \boldsymbol{k}_{j'}]$ (and similarly $\partial_{\boldsymbol{k}_t}$) requires at least 2 self-attention layers and an MLP layer , as we must compute and multiply attention scores $a_{t,j}$ and $(\partial_{\boldsymbol{y}_t})^{\top} \boldsymbol{v}_j$ before computing $\partial_{\boldsymbol{q}_t}$. Thus, we only update the self-attention using the gradients w.r.t. $\boldsymbol{v}_t$.

**Definition 2.12** (Approximate Self-Attention Backpropagation). With the notations defined above, this layer takes a sequence $\{\partial_{\boldsymbol{y}_t} \in \mathbb{R}^{D_{\text{aux}}}\}_{t \leq T_{\text{aux}}}$ and $\{\boldsymbol{x}_t \in \mathbb{R}^{D_{\text{aux}}}\}_{t \leq T_{\text{aux}}}$ as input and outputs $\{\widehat{\partial_{\boldsymbol{x}_t}}\}_{t \leq T}$, with $\widehat{\partial_{\boldsymbol{x}_t}} = \boldsymbol{W}_V^{\top} \partial_{\boldsymbol{v}_t}$, where $\partial_{\boldsymbol{v}_t} = \sum_j a_{j,t} \partial_{\boldsymbol{y}_j}$.

We formally show that when the attention head for each position pays a lot of attention to a single token (i.e., behaves like hard attention (Perez et al., 2021)), $\widehat{\partial_{\boldsymbol{x}_t}}$ is entry-wise close to $\partial_{\boldsymbol{x}_t}$ for all $t$. Computing $\{\widehat{\partial_{\boldsymbol{x}_t}}\}_{t=1}^{T}$ instead of $\{\partial_{\boldsymbol{x}_t}\}_{t=1}^{T}$ induces a $5\times$ parameter reduction.

**Theorem 2.13** (Informal, c.f. Theorem C.4). *If on input sequence $\{\boldsymbol{x}_t\}_{t \leq T_{aux}}$, the attention scores are $\varepsilon$-close to a hard-attention at each position, then for all $t$, $\|\partial_{\boldsymbol{x}_t} - \widehat{\partial_{\boldsymbol{x}_t}}\| \leq \mathcal{O}(\varepsilon)$.*

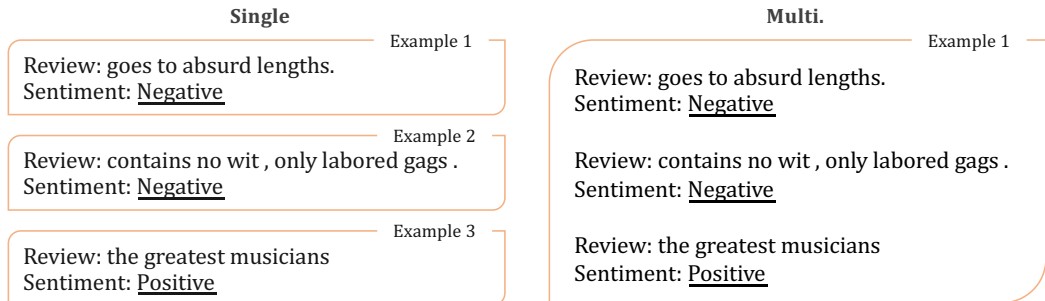

Figure 3: This illustration showcases different settings in few-shot learning ($k = 3$) using TINT. The **Single** mode (left) has one example for each input, and the auxiliary model is updated with a batch of inputs. The **Multi.** mode (right) concatenates all examples to form a single input. For **Label loss**, only underlined label words are used for internal training, while **full context loss** includes all tokens.

## 2.7 PARAMETER SHARING IN THE TINT

Consider the self-attention layer (Section 2.6). The relevant TINT module performs linear operations with $\boldsymbol{W}_Q, \boldsymbol{W}_K, \boldsymbol{W}_V$ to compute query, key, and value vectors at each position $t$ (Definition 2.2) and hence can be simulated with the Linear Forward module (Section 2.4). We additionally leverage parameter sharing to apply a single Linear Forward module for each of the three computations, changing only the prefix embeddings to correspond to $\boldsymbol{W}_Q, \boldsymbol{W}_K$, or $\boldsymbol{W}_V$. Applying the same structure to feed-forward linear layers results in a $4\times$ reduction in the number of necessary modules (Appendix G).

## 3 EXPERIMENTS

We conduct experiments on TINT constructed using GPT2 and OPT-125M as auxiliary models to validate that the approximations introduced (Section 2.2) do not significantly harm the capability of TINT to simulate and train an internal model.

### 3.1 EXPERIMENTAL SETUP

We introduce dynamic evaluation (Krause et al., 2019), which updates the model with a segment of the input and evaluates the updated model on the rest of the input.

**Definition 3.1** (Dynamic Evaluation (Krause et al., 2019))**.** For a sequence $s \in \mathcal{S}$ of $T$ tokens $s_1, .., s_T$ and hyperparameters $i$ and $\eta$, *dynamic evaluation* of any model $f$ with parameters $\theta \in \Theta$ is defined as follows. Let $\mathcal{L} : \{\mathcal{S}, \Theta\} \to \mathbb{R}$ be the cross-entropy language modeling loss. Compute

$$\theta' = \theta - \eta \nabla_\theta \mathcal{L}((s_1, ..., s_i), \theta).$$

Then, the dynamic evaluation loss is $\mathcal{L}((s_{i+1}, ..., s_T), \theta')$. For in-context learning, the first $i$ *examples* are used to update the parameters.

For example, suppose we perform dynamic evaluation with $i = 5$ using the cross-entropy objective on Machine learning is a useful tool for solving problems. The red part is used to update the pre-trained model, and the brown part is used for evaluation. This operation is equivalent to TINTinference, which implicitly updates a model on the red portion during the forward pass and then evaluates the updated model on the brown portion. Therefore, we use dynamic evaluation as a baseline for TINT: matching dynamic evaluation performance provides strong evidence that TINT closely approximates explicitly fine-tuning the auxiliary model.

**Tasks:** We perform language modeling experiments on Wikitext-103 (Merity et al., 2016) and evaluate on 7 downstream tasks in zero-shot and few-shot settings: SST-2 (Socher et al., 2013), MR (Pang and Lee, 2004), CR (Hu and Liu, 2004), MPQA (Wiebe et al., 2005), Amazon Polarity (Zhang et al., 2015), AGNews (Zhang et al., 2015), and Subj (Pang and Lee, 2005).

**Model:** We compare a TINT model that tunes an OPT-125M pre-trained model internally to dynamic evaluation of OPT-125M and standard evaluation of OPT-1.3B.[3] Given a downstream task input

---

[3]Our construction is generally applicable to diverse variants of pre-trained language models (Appendix I).

Table 1: Language modeling results on WIKITEXT-103. We use $30\%, 50\%, 70\%$ and $90\%$ of sequences for training in dynamic eval and TINT and the rest of the sequence for evaluation. TINT improves upon the auxiliary model perplexities by $0.3 - 0.7$ absolute on average. The small perplexity difference between the TINTand dynamic evaluation suggests that the approximations introduced in the descent algorithm (Section 2.2) can still effectively fine-tune the auxiliary model.

| Training proportion | GPT2 | | | | OPT-125m | | | |
|---|---|---|---|---|---|---|---|---|
| | 30% | 50% | 70% | 90% | 30% | 50% | 70% | 90% |
| VANILLA MODEL | 25.6 | 24.9 | 24.5 | 23.3 | 29.6 | 28.8 | 28.0 | 28.0 |
| DYNA. EVAL | 24.9 | 24.0 | 23.5 | 22.2 | 29.0 | 28.2 | 27.4 | 27.4 |
| TINT | 25.1 | 24.3 | 23.8 | 22.6 | 29.3 | 28.4 | 27.5 | 27.4 |

(e.g., a movie review), the model's predicted label is computed as follows. First, we design a simple task-specific prompt (e.g., "Sentiment:") and select label words $c_1, ..., c_n$ to serve as surrogates for each class (e.g., "positive" and "negative"). Then, we provide the input along with the prompt to the model, and the label word assigned the highest probability is treated as the model's prediction. If using calibration, then the probabilities are normalized using just the prompt as input.[4] Mathematically, the model's prediction is computed as

$$\text{No calibration: } \arg\max_{c_i} \Pr[c_i \mid \text{input, prompt}] \qquad \text{Calibration: } \arg\max_{c_i} \frac{\Pr[c_i \mid \text{input, prompt}]}{\Pr[c_i \mid \text{prompt}]}$$

This is a widely used calibration technique (Holtzman et al., 2021) for prompting language models.

**Settings (Figure 3):** We explore the following settings in downstream tasks: 1) Single and Multi.: We finetune the auxiliary model using either separate single examples or concatenated examples within each input; 2) Label loss and full-context loss: We finetune on the loss either from only label words or the entire context (Figure 3). We evaluate both zero-shot and few-shot settings, using the context of the evaluation example and 32 training examples for internal learning respectively.

### 3.2 VERIFICATION OF TINT

In language modeling (Table 1), the perplexity decreases using TINT, especially as the training proportion increases. For downstream tasks (Table 2), explicit internal training within TINT surpasses vanilla zero-shot evaluation and in-context learning, even with a limited budget of a single forward pass. Moreover, TINT achieves a performance comparable to dynamic evaluation, indicating that the approximations made during its construction largely preserve its effectiveness for fine-tuning. Though calibration may not always be beneficial in every setting,[5] we observe that the efficacy of TINT remains comparable to dynamic evaluation. Additionally, we find that TINT outperforms or is on par with a similarly sized pre-trained model (OPT-1.3B) except in the calibrated few-shot setting. This suggests that the capabilities of existing pre-trained models may be understood via the simulation of smaller auxiliary models. Please refer to Appendix J for more experiment details.

### 4 RELATED WORK

**Interpretability:** Mechanistic interpretability works reverse-engineer the algorithms simulated by these models (Elhage et al., 2021; Olsson et al., 2022; Wang et al., 2022; Nanda et al., 2023; Chughtai et al., 2023). These works study local patterns, e.g. activations and attention heads, to derive interpretable insights. Other works (Weiss et al., 2021; Lindner et al., 2023) use declarative programs to algorithmically describe transformer models.

**Transformer Expressivity:** Perez et al. (2021); Pérez et al. (2019) show that Transformers with hard attention are Turing complete, with Wei et al. (2021) showing statistically meaningful transformer constructions for Turing machines for statistical learnability. In Section 2.3, we point out that this scheme often results in gigantic constructions. To understand the behavior of moderate sized models,

---

[4]Calibration is not applied to the language modeling evaluation.

[5]Such inconsistencies in the calibration method have been observed in previous works (Brown et al., 2020).

Table 2: Zero-shot and few-shot in-context learning results across 7 downstream tasks. All the few-shot results are averaged over three training seeds. TINT consistently surpasses its auxiliary model and achieves comparable performance to dynamic evaluation. TINT outperforms auxiliary models by $3-4\%$ and $12-16\%$ absolute points on average in $0$-shot and $32$-shot experiments respectively. TINT performs competitively with a similar-sized pre-trained model (OPT-1.3B) in both $0$-shot and $32$-shot settings. We show the standard deviation for few-shot settings in parentheses.

| Model | Shots | Subj | AGNews | SST2 | CR | MR | MPQA | Amazon | Avg. |
|---|---|---|---|---|---|---|---|---|---|
| | | | | *Without Calibration* | | | | | |
| OPT-125M | 0 | 64.0 | 66.0 | 70.5 | 64.5 | 71.0 | 68.0 | 76.5 | 68.6 |
| OPT-1.3B | 0 | 59.0 | 55.5 | 54.0 | 50.5 | 52.5 | 74.0 | 57.0 | 57.5 |
| OPT-125M DYNA. EVAL | 0 | 71.0 | 67.0 | 79.5 | 71.5 | 70.0 | 68.0 | 85.5 | 73.2 |
| OPT-125M TINT | 0 | 67.5 | 66.0 | 76.5 | 69.0 | 76.0 | 70.5 | 78.5 | 72.0 |
| OPT-125M | 32 | $58.7_{(4.9)}$ | $33.7_{(8.4)}$ | $50.8_{(1.2)}$ | $51.3_{(1.9)}$ | $50.0_{(0.0)}$ | $54.3_{(2.5)}$ | $55.0_{(6.7)}$ | $50.5_{(1.9)}$ |
| OPT-1.3B | 32 | $74.2_{(6.1)}$ | $71.3_{(5.3)}$ | $89.8_{(3.6)}$ | $71.5_{(4.5)}$ | $68.3_{(6.1)}$ | $81.7_{(3.3)}$ | $70.3_{(9.9)}$ | $75.3_{(0.4)}$ |
| OPT-125M DYNA. EVAL | 32 | $78.0_{(1.4)}$ | $66.7_{(1.6)}$ | $71.5_{(1.4)}$ | $73.7_{(3.3)}$ | $72.0_{(0.0)}$ | $80.7_{(0.6)}$ | $79.8_{(0.2)}$ | $74.6_{(2.7)}$ |
| OPT-125M TINT | 32 | $82.3_{(2.7)}$ | $69.3_{(0.9)}$ | $73.7_{(0.8)}$ | $75.7_{(1.9)}$ | $72.3_{(1.2)}$ | $83.2_{(1.0)}$ | $78.2_{(0.2)}$ | $76.4_{(0.7)}$ |
| | | | | *With Calibration* | | | | | |
| OPT-125M | 0 | 64.0 | 66.0 | 53.0 | 54.5 | 52.5 | 55.5 | 58.0 | 57.6 |
| OPT-1.3B | 0 | 73.5 | 61.5 | 57.5 | 53.0 | 54.5 | 79.5 | 61.0 | 62.9 |
| OPT-125M DYNA. EVAL | 0 | 62.5 | 66.0 | 60.5 | 53.5 | 54.0 | 56.5 | 74.5 | 61.1 |
| OPT-125M TINT | 0 | 64.0 | 66.0 | 56.5 | 59.0 | 53.5 | 62.0 | 66.5 | 61.1 |
| OPT-125M | 32 | $83.5_{(2.4)}$ | $40.7_{(10.4)}$ | $50.8_{(0.8)}$ | $67.7_{(4.1)}$ | $57.7_{(10.8)}$ | $79.2_{(8.4)}$ | $56.0_{(8.1)}$ | $62.2_{(2.7)}$ |
| OPT-1.3B | 32 | $51.8_{(1.9)}$ | $66.2_{(3.1)}$ | $93.7_{(1.0)}$ | $82.8_{(2.8)}$ | $91.3_{(1.9)}$ | $83.5_{(2.5)}$ | $92.0_{(2.9)}$ | $80.2_{(0.7)}$ |
| OPT-125M DYNA. EVAL | 32 | $87.2_{(0.2)}$ | $67.2_{(0.6)}$ | $72.8_{(5.9)}$ | $73.3_{(2.6)}$ | $66.7_{(7.4)}$ | $81.5_{(3.7)}$ | $70.3_{(2.1)}$ | $74.1_{(2.9)}$ |
| OPT-125M TINT | 32 | $85.3_{(1.9)}$ | $67.3_{(0.6)}$ | $71.8_{(3.8)}$ | $70.7_{(1.9)}$ | $63.7_{(0.2)}$ | $83.5_{(1.6)}$ | $77.5_{(1.2)}$ | $74.3_{(1.4)}$ |

other works have investigated specific classes of algorithms, e.g. bounded-depth Dyck languages (Yao et al., 2021), modular prefix sums (Anil et al., 2022), adders (Nanda et al., 2023), regular languages (Bhattamishra et al., 2020), and sparse logical predicates (Edelman et al., 2022). Liu et al. (2023) provide a unified theory to understand automata-like mechanisms within transformers.

Fast Weight Programmers (FWPs) enable input-dependent weight updates during inference. Ba et al. (2016) connect self-attention and FWPs, and follow-up works (Schlag et al., 2021; Irie et al., 2021) show the efficacy of self-attention layers to update linear and recurrent networks during inference. Clark et al. (2022) added Fast Weights Layers (FWL) to a frozen pre-trained model, and efficiently fine-tune FWL as the model processes the sequence.

**Alternative Explanations for ICL:** Some works study ICL using a Bayesian framework. Xie et al. (2022) model pretraining data as a mixture of HMMs and cast ICL identifying one such component. Hahn and Goyal (2023) later modeled language as a compositional grammar, and propose ICL as a composition of operations. On the other hand, careful experiments in Chan et al. (2022) show that data distributional properties (e.g. Zipf's law) drive in-context learning in transformers.

**Transfer learning:** Our construction uses a pre-trained model to initialize a larger transformer, which is similar to several other more empirically oriented works (Gong et al., 2019; Reddi et al., 2023).

## 5 DISCUSSION

We present a parameter-efficient construction TINT capable of simulating gradient descent on an internal transformer modelduring inference. Using fewer than 2 billion parameters, it can simulate fine-tuning a 125 million transformer (e.g., GPT-2) internally, dramatically reducing the scale required by previous works. Language modeling and in-context learning experiments demonstrate that the efficient approximations still allow the TINT to fine-tune the model. Our work emphasizes that the inference behavior of complex models may rely on the training dynamics of smaller models. As such, the existence of TINT has strong implications for interpretability and AI alignment research.

While our work represents a significant improvement over previous simulations in terms of auxiliary model complexity, similar to prior research in this area, our insights into existing pre-trained models are limited. Furthermore, we have not yet examined potential biases that may arise in the auxiliary models due to one-step gradient descent. We plan to investigate these aspects in future work.

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

CONTENTS

## A    ADDITIONAL NOTATIONS

We differentiate the parameters of the auxiliary model and TINT by using an explicit superscript TINT for TINT parameters, for example, the weights of a linear layer in TINT will be represented by $\boldsymbol{W}^{\text{TINT}}$. We use two operations throughout: $\text{SPLIT}_h$ and $\text{VECTORIZE}$. Function $\text{SPLIT}_h : \mathbb{R}^d \to \mathbb{R}^{h \times \lfloor d/h \rfloor}$ takes an input $\boldsymbol{x} \in \mathbb{R}^d$ and outputs $H$ equal splits of $\boldsymbol{x}$, for any arbitrary dimension $d$. Function $\text{VECTORIZE} : \mathbb{R}^{h \times d} \to \mathbb{R}^{dh}$ concatenates the elements of a sequence $\{\boldsymbol{x}_i \in \mathbb{R}^d\}_{i \leq h}$ into one single vector, for any arbitrary $d$ and $h$. Recall that for a matrix $\boldsymbol{A}$, $\boldsymbol{a}_j$ refers to its $j$th row, and for any vector $\boldsymbol{b}$, $b_j$ refers to its $j$th element. However, at a few places in the appendix, for typographical reasons, for a matrix $\boldsymbol{A}$, we have also used $(\boldsymbol{A})_j$ to refer to its $j$th row, and for any vector $\boldsymbol{b}$, $(\boldsymbol{b})_j$ to refer to its $j$th element.

**TINTAttention Module**    We modify the usual attention module to include the position embeddings $\{\boldsymbol{p}_i^{\text{TINT}} \in \mathbb{R}^{T_{\text{sim}}}\}_{i \leq T_{\text{sim}}}$. In usual self-attention modules, the query, key, and value vectors at each position are computed by token-wise linear transformations of the input embeddings. In TINT's Attention Module, we perform additional linear transformations on the position embeddings, using parameters $\boldsymbol{W}_Q^p, \boldsymbol{W}_K^p, \boldsymbol{W}_V^p$, and decision vectors $\lambda^Q, \lambda^K, \lambda^V \in \mathbb{R}^{H_{\text{sim}}}$ decide whether to add these transformed position vectors to the query, key, and value vectors of different attention heads. The following definition generalizes single-head attention (Definition 2.3). For the following definition, we use $\widehat{e}$ to represent input sequence and $\widetilde{e}$ to represent the output sequence: we introduce these general notations below to avoid confusion with the notations for token and prefix embeddings for TINTillustrated in Figure 1.

**Definition A.1** (TINT's self-attention with $H_{\text{sim}}$ heads). For parameters $\{\boldsymbol{W}_Q^{\text{TINT}}, \boldsymbol{W}_K^{\text{TINT}}, \boldsymbol{W}_V^{\text{TINT}} \in \mathbb{R}^{D_{\text{sim}} \times D_{\text{sim}}}\}$, $\{\boldsymbol{b}_Q^{\text{TINT}}, \boldsymbol{b}_K^{\text{TINT}}, \boldsymbol{b}_V^{\text{TINT}} \in \mathbb{R}^{D_{\text{sim}}}\}$, $\{\boldsymbol{W}_Q^p, \boldsymbol{W}_K^p, \boldsymbol{W}_V^p \in \mathbb{R}^{T_{\text{sim}} \times D_{\text{sim}}/H_{\text{sim}}}\}$ and $\{\lambda^Q, \lambda^K, \lambda^V \in \mathbb{R}^{H_{\text{sim}}}\}$, TINT self-attention with $H_{\text{sim}}$ attention heads and a function $f_{\text{attn}} : \mathbb{R}^{T_{\text{sim}}} \to \mathbb{R}^{T_{\text{sim}}}$ takes a sequence $\{\widehat{\boldsymbol{e}}_t \in \mathbb{R}^{D_{\text{sim}}}\}_{t \leq T_{\text{sim}}}$ as input and outputs $\{\widetilde{\boldsymbol{e}}_t \in \mathbb{R}^{D_{\text{sim}}}\}_{t \leq T_{\text{sim}}}$, with

$$\widetilde{\boldsymbol{e}}_t = \text{VECTORIZE}(\{ \sum_{j \leq T_{\text{sim}}} a_{t,j}^h \widetilde{\boldsymbol{v}}_j^h)_h \}_{h \leq H_{\text{sim}}}), \text{ with } a_{t,j}^h = f_{\text{attn}}(\widetilde{\boldsymbol{K}}^h \widetilde{\boldsymbol{q}}_t^h)_j$$

$$\widetilde{\boldsymbol{q}}_t^h = \text{SPLIT}_H(\boldsymbol{q}_t)_h + \lambda_h^Q \boldsymbol{W}_Q^p \boldsymbol{p}_t^{\text{TINT}}; \quad \widetilde{\boldsymbol{k}}_t^h = \text{SPLIT}_H(\boldsymbol{k}_t)_h + \lambda_h^K \boldsymbol{W}_K^p \boldsymbol{p}_t^{\text{TINT}};$$

$$\widetilde{\boldsymbol{v}}_t^h = \text{SPLIT}_H(\boldsymbol{v}_t)_h + \lambda_h^V \boldsymbol{W}_v^p \boldsymbol{p}_t^{\text{TINT}}.$$

Here, $\boldsymbol{q}_t, \boldsymbol{k}_t, \boldsymbol{v}_t$ denote the query, key, and value vectors at each position $t$, computed as $\boldsymbol{W}_Q^{\text{TINT}} \widehat{\boldsymbol{e}}_t + \boldsymbol{b}_Q^{\text{TINT}}$, $\boldsymbol{W}_K^{\text{TINT}} \widehat{\boldsymbol{e}}_t + \boldsymbol{b}_K^{\text{TINT}}$, and $\boldsymbol{W}_V^{\text{TINT}} \widehat{\boldsymbol{e}}_t + \boldsymbol{b}_V^{\text{TINT}}$ respectively. $\widetilde{\boldsymbol{K}}^h \in \mathbb{R}^{T_{\text{sim}} \times D_{\text{sim}}/H_{\text{sim}}}$ is defined with its rows as $\{\widetilde{\boldsymbol{k}}_t^h\}_{t \leq T_{\text{sim}}}$ for all $h \leq H_{\text{sim}}$.

$f_{\text{attn}}$ can be either linear or softmax function.

**Bounded parameters and input sequence:**    We define a linear self-attention layer to be $B_w$-bounded, if the $\ell_2$ norms of all the parameters are bounded by $B_w$. Going by Definition A.1, this implies

$$\max\{\left\|\boldsymbol{W}_Q^{\text{TINT}}\right\|_2, \left\|\boldsymbol{W}_K^{\text{TINT}}\right\|_2, \left\|\boldsymbol{W}_V^{\text{TINT}}\right\|_2\} \leq B_w, \quad \max\{\left\|\boldsymbol{b}_Q^{\text{TINT}}\right\|_2, \left\|\boldsymbol{b}_K^{\text{TINT}}\right\|_2, \left\|\boldsymbol{b}_V^{\text{TINT}}\right\|_2\} \leq B_w$$

$$\max\{\left\|\boldsymbol{W}_Q^p\right\|_2, \|\boldsymbol{W}_K^p\|_2, \|\boldsymbol{W}_V^p\|_2\} \leq B_w, \quad \max\{\left\|\lambda^Q\right\|_2, \left\|\lambda^K\right\|_2, \left\|\lambda^V\right\|_2\} \leq B_w.$$

Furthermore, we define an input sequence $\{\widehat{e}_t\}_{t \leq T_{\text{sim}}}$ to $B_x$-bounded, if $\|\widehat{e}_t\|_2 \leq B_x$ for all $t$.

Recall from the main paper (Section 2.4), we used Linear TINT Self-Attention layer to represent the linear operations of the auxiliary model. In the following theorem, we show that a linear attention layer can be represented as a softmax attention layer that uses an additional attention head and an extra token $u$, followed by a linear layer. Therefore, replacing softmax attention with linear attention does not deviate too far from the canonical transformer. We use the Linear TINT Self-Attention layers in several places throughout the model.

**Theorem A.2.** *For any $B_w > 0$, consider a $B_w$-bounded linear self-attention layer that returns $\{\widetilde{e}_t^{linear} \in \mathbb{R}_{sim}^D\}_{t \leq T_{sim}}$ on any input $\{\widehat{e}_t \in \mathbb{R}_{sim}^D\}_{t \leq T_{sim}}$. Consider a softmax self-attention layer with $2H_{sim}$ attention heads and an additional token $u \in \mathbb{R}^{2D_{sim}}$ such that for any $B_x$-bounded input $\{\widehat{e}_t\}_{t \leq T_{sim}}$, it takes a modified input sequence $\{\bar{e}_1, \cdots, \bar{e}_{T_{sim}}, u\}$, and returns $\{\widetilde{e}_t^{softmax} \in \mathbb{R}^{2D_{sim}}\}_{t \leq T_{sim}}$. Each modified input token $\bar{e}_t \in \mathbb{R}^{2D_{sim}}$ is obtained by concatenating additional $0$s to $\widehat{e}_t$. Then, for any $B_x > 0$, and $\epsilon \leq \mathcal{O}(T_{sim}^{-2} B_w^{-5} B_x^{-5})$, there exists $W_O \in \mathbb{R}^{D_{sim} \times 2D_{sim}}$ and such a softmax self-attention layer such that*

$$\left\| W_O \widetilde{e}_t^{softmax} - \widetilde{e}_t^{linear} \right\|_2 \leq \mathcal{O}(\sqrt{\epsilon}),$$

*for all $t \leq T_{sim}$.*

*Proof.* Consider an input sequence $\{x_t\}_{t \leq T_{\text{sim}}}$. Let the attention scores of any linear head $h \leq H_{\text{sim}}$ in the linear attention layer be given by $\{a_{t,j}^h\}_{j \leq T_{\text{sim}}}$, at any given position $t$. Additionally, let the value vectors for the linear attention be given by $v_t$. To repeat our self-attention definition, the output of the attention layer at any position $t$ is given by $\text{VECTORIZE}(\{\widetilde{e}_t^{linear,h}\}_{h \leq H_{\text{sim}}})$, where

$$\widetilde{e}_t^{linear,h} = \sum_{j \leq T_{\text{sim}}} a_{t,j}^h v_j^h.$$

Under our assumption, $B_w$ denotes the maximum $\ell_2$ norm of all the parameters in the linear self-attention layer and $B_x$ the maximum $\ell_2$ norm in the input sequence, i.e. $\max_{t \leq T_{\text{sim}}} \|x_t\|_2 \leq B_x$. With a simple application of Cauchy-Schwartz inequality, we can show that $\max_{j \leq T_{\text{sim}}} |a_{t,j}^h| \leq \mathcal{O}(B_w^2 B_x^2)$, and $\max_{t \leq T_{\text{sim}}} \|v_t^h\|_2 \leq \mathcal{O}(B_w B_x)$.

For $\epsilon \leq \mathcal{O}(T_{\text{sim}}^{-10/9} B_w^{-40/9} B_x^{-40/9})$, we can then use Lemma A.3 to represent for each $t, j \leq T_{\text{sim}}$,

$$a_{t,j}^h = \frac{\epsilon^{-3} e^{\epsilon a_{t,j}}}{\sum_{t' \leq T_{\text{sim}}} e^{\epsilon a_{t,t'}^h} + e^{-2\log \epsilon}} - \epsilon^{-1} + \mathcal{O}\left(\epsilon(T_{\text{sim}} + a_{t,j}^h)\right)$$

$$:= \epsilon^{-3} \text{softmax}\left(\{\epsilon a_{t,1}^h, \epsilon a_{t,2}^h, \cdots, \epsilon a_{t,T_{\text{sim}}}^h, -2\log \epsilon\}\right)_j - \epsilon^{-1} + \mathcal{O}\left(\epsilon^{0.9}\right).$$

**Softmax attention construction:** We define $u$, and the query and key parameters of the softmax attention layer such that for the first $H_{\text{sim}}$ attention heads, the query-key dot products for all the attention heads between any pairs $\{(\bar{e}_t, \bar{e}_j)\}_{t,j \leq T_{\text{sim}}}$ is given by $\{\epsilon a_{t,j}^h\}_{h \leq H_{\text{sim}}}$, while being $-2\log \epsilon$ between $u$ and any token $\bar{e}_t$, with $t \leq T_{\text{sim}}$. For the rest of $H_{\text{sim}}$ attention heads, the attention scores are uniformly distributed across all pairs of tokens (attention score between any pair of tokens is given by $\frac{1}{T_{\text{sim}}+1}$).

We set the value parameters of the softmax attention layer such that at any position $t \leq T_{\text{sim}}$, the value vector is given by $\text{VECTORIZE}(\{\epsilon^{-3} v_t, v_t\})$. The value vector returned for $u$ contains all $0$s.

**Softmax attention computation:** Consider an attention head $h \leq H_{\text{sim}}$ in the softmax attention layer now. The output of the attention head at any position $t \leq T_{\text{sim}}$ is given by

$$\widetilde{e}_t^{softmax,h} = \sum_{j \leq T_{\text{sim}}} \text{softmax}\left(\{\epsilon a_{t,1}^h, \epsilon a_{t,2}^h, \cdots, \epsilon a_{t,T_{\text{sim}}}^h, -2\log \epsilon\}\right)_j \epsilon^{-3} v_j^h$$

$$= \sum_{j \leq T_{\text{sim}}} \left(a_{t,j}^h + \epsilon^{-1} + \mathcal{O}(\epsilon^{0.9})\right) v_j^h.$$

This has an additional $\sum_{j \leq T_{\text{sim}}} \left( \epsilon^{-1} + \mathcal{O}(\epsilon^{0.9}) \right) \boldsymbol{v}_j^h$, compared to $\widetilde{\boldsymbol{e}}_t^{linear,h}$. However, consider the output of the attention head $H_{\text{sim}} + h$ at the same position:

$$\widetilde{\boldsymbol{e}}_t^{softmax,H_{\text{sim}}+h} = \frac{1}{T_{\text{sim}} + 1} \sum_{j \leq T_{\text{sim}}} \boldsymbol{v}_j^h.$$

Hence, we can use the output matrix $\boldsymbol{W}_O$ to get $\widetilde{\boldsymbol{e}}_t^{softmax,h} - \frac{T_{\text{sim}}+1}{\epsilon} \widetilde{\boldsymbol{e}}_t^{softmax,H_{\text{sim}}+h} = \sum_{j \leq T_{\text{sim}}} \left( a_{t,j}^h + \mathcal{O}(\epsilon^{0.9}) \right) \boldsymbol{v}_j^h$. The additional term $\mathcal{O}(\epsilon^{0.9}) \sum_{j \leq T_{\text{sim}}} \boldsymbol{v}_j^h$ can be further shown to be $\mathcal{O}(\epsilon^{0.5})$ small with the assumed bound of $\epsilon$, since each $\boldsymbol{v}_j^h$ is almost $\mathcal{O}(B_w B_x)$ in $\ell_2$ norm with a Cauchy Schwartz inequality. $\qquad \square$

**Lemma A.3.** *For $\epsilon > 0$, $B > 0$, and a sequence $\{a_1, a_2, \cdots, a_T\}$ with each $a_i \in \mathbb{R}$ and $|a_i| \leq B$, the following holds true for all $i \leq T$,*

$$\frac{\epsilon^{-3} e^{\epsilon a_i}}{\sum_{t' \leq T} e^{\epsilon a_{t'}} + e^{-2 \log \epsilon}} = a_i + \frac{1}{\epsilon} + \mathcal{O}\left(\epsilon^{0.9}\right),$$

*provided $\epsilon \leq \mathcal{O}(T^{-10/9} B^{-20/9})$.*

*Proof.* We will use the following first-order Taylor expansions:

$$e^x = 1 + x + \mathcal{O}(x^2). \tag{1}$$

$$\frac{1}{1+x} = 1 - \mathcal{O}(x). \tag{2}$$

Hence, for any $x \ll 1$, $x \approx e^x - 1$.

Simplifying the L.H.S. of the desired bound, we have

$$\frac{\epsilon^{-3} e^{\epsilon a_i}}{\sum_{t' \leq T} e^{\epsilon a_{t'}} + e^{-2 \log \epsilon}} = \frac{\epsilon^{-3}(1 + \epsilon a_i + \mathcal{O}(\epsilon^2 a_i^2))}{\sum_{t' \leq T}(1 + \epsilon a_{t'} + \mathcal{O}(\epsilon^2 a_{t'}^2)) + e^{-2 \log \epsilon}} \tag{3}$$

$$= \frac{\epsilon^{-1} + a_i + \mathcal{O}(\epsilon a_i^2)}{\sum_{t' \leq T}(\epsilon^2 + \epsilon^3 a_{t'} + \mathcal{O}(\epsilon^4 a_{t'}^2)) + 1} \tag{4}$$

$$= \left(\epsilon^{-1} + a_i + \mathcal{O}(\epsilon a_i^2)\right)\left(1 + \mathcal{O}(\epsilon^2 T)\right) \tag{5}$$

$$= \epsilon^{-1} + a_i + \mathcal{O}(\epsilon T + a_i^2 T \epsilon^2 + a_i^2 T \epsilon^3 + \epsilon a_i^2) = \epsilon^{-1} + a_i + \mathcal{O}(\epsilon^{0.9}).$$

We used taylor expansion of exponential function( Equation (1) ) in Equation (3) to get Equation (4), and taylor expansion of inverse function(Equation (2)) to get Equation (5) from Equation (4). Furthermore, with the lower bound assumption on $\epsilon$, $\sum_{t' \leq T}(\epsilon^2 + \epsilon^3 a_{t'} + \mathcal{O}(\epsilon^4 a_{t'}^2))$ can be shown to be atmost $3\epsilon^2 T$, which amounts to $\mathcal{O}(\epsilon^2 T)$ error in Equation (5). The final error bound has again been simplified using the lower bound assumption on $\epsilon$. $\qquad \square$

## A.1 SIMULATING MULTIPLICATION FROM AKYUREK ET AL. (2022)

We refer to the multiplication strategy of Akyurek et al. (2022) at various places.

**Lemma A.4.** *[Lemma 4 in Akyurek et al. (2022)] The $GeLU$ Hendrycks and Gimpel (2016) nonlinearity can be used to perform multiplication: specifically,*

$$\sqrt{\pi/2}(GeLU(x+y) - GeLU(y)) = xy + \mathcal{O}(x^3 y^3).$$

Thus, to represent an element-wise product or a dot product between two sub-vectors in a token embedding, we can use a MLP with a $GeLU$ activation.

## B LINEAR LAYER

In the main paper, we defined the linear layer without the bias term for simplicity (Definition 2.2). In this section, we will redefine the linear layer with the bias term and present a comprehensive construction of the Linear Forward module.

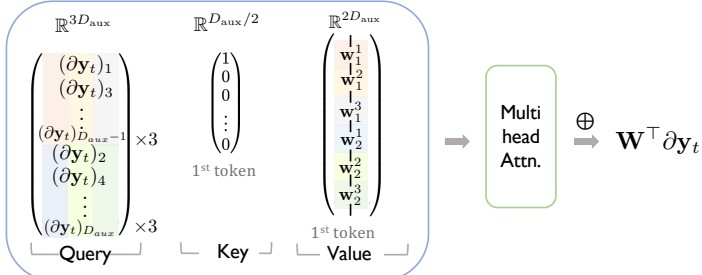

Figure 4: TINT simulates the backward pass of a linear layer as a $H$-head attention layer ($H = 6$ pictured), with the gradient of the loss w.r.t. linear layer output ($\partial_{\boldsymbol{y}_t}$) as the query, the positional one-hot vector of prefix embeddings as the key, and the parameters of the auxiliary model stored in the prefix embeddings as the value. Similar to the Linear Forward module (Figure 2), we distribute the dot product computations across all attention heads by sharding the vectors into $S'$ ($S' = 3$ here) parts. We omitted the identical transformation for query, and value matrices, and permutation-based transformation for key matrix for illustration purposes.

**Definition B.1** (Linear layer). For a weight $\boldsymbol{W} \in \mathbb{R}^{D_{\mathrm{aux}} \times D_{\mathrm{aux}}}$ and bias $\boldsymbol{b} \in \mathbb{R}^{D_{\mathrm{aux}}}$, a linear layer takes $\boldsymbol{x} \in \mathbb{R}^{D_{\mathrm{aux}}}$ as input and outputs $\boldsymbol{y} = \boldsymbol{W}\boldsymbol{x} + \boldsymbol{b}$.

In the discussions below, we consider a linear layer in the auxiliary model with parameters $\{\boldsymbol{W}, \boldsymbol{b}\}$ that takes in input sequence $\boldsymbol{x}_1, \cdots, \boldsymbol{x}_{T_{\mathrm{aux}}}$ and outputs $\boldsymbol{y}_1, \cdots, \boldsymbol{y}_{T_{\mathrm{aux}}}$, with $\boldsymbol{y}_t = \boldsymbol{W}\boldsymbol{x}_t + \boldsymbol{b}$ for each $t \le T_{\mathrm{aux}}$. Since this involves a token-wise operation, we will present our constructed modules with a general token position $t$ and the prefix tokens $\{\boldsymbol{v}_j\}$.

**TINT Linear Forward module**  Continuing our discussion from Section 2.4, we represent $S$ stacked rows of $\boldsymbol{W}$ as a prefix embedding. In addition, we store the bias $\boldsymbol{b}$ in the first prefix embedding ($\boldsymbol{v}_1$).

Using a set of $S'$ unique attention heads in a TINT attention module (Definition A.1), we copy the bias $\boldsymbol{b}$ to respective token embeddings and use a TINT linear layer to add the biases to the final output.

**Auxiliary's backpropagation through linear layer**  For a linear layer as defined in Definition B.1, the linear backpropagation layer takes in the loss gradient w.r.t. output ($\partial_{\boldsymbol{y}}$) and computes the loss gradient w.r.t. input ($\partial_{\boldsymbol{x}}$).

**Definition B.2** (Linear backpropagation ). For a weight $\boldsymbol{W} \in \mathbb{R}^{D_{\mathrm{aux}} \times D_{\mathrm{aux}}}$, the linear backpropagation layer takes $\partial_{\boldsymbol{y}} \in \mathbb{R}^{D_{\mathrm{aux}}}$ as input and outputs $\partial_{\boldsymbol{x}} = \boldsymbol{W}^{\top} \partial_{\boldsymbol{y}}$.

**TINT Linear backpropagation module**  This module will aim to simulate the auxiliary's linear backpropagation. The input embedding $\boldsymbol{e}_t$ to this module will contain the gradient of the loss w.r.t. $\boldsymbol{y}_t$, i.e. $\partial_{\boldsymbol{y}_t}$. As given in Definition B.2, this module will output the gradient of the loss w.r.t. $\boldsymbol{x}_t$, given by $\partial_{\boldsymbol{x}_t} = \boldsymbol{W}^{\top} \partial_{\boldsymbol{y}_t}$.

We first use the residual connection to copy the prefix embeddings $\{\boldsymbol{v}_j\}$ (i.e., the rows of $\boldsymbol{W}$) from the forward propagation module. A straightforward construction would be to use the Linear Forward module but with the columns of $\boldsymbol{W}$ stored in the prefix tokens, thereby simulating multiplication with $\boldsymbol{W}^{\top}$. However, such a construction requires applying attention to the prefix tokens, which increases the size of the construction substantially.

We instead perform the operation more efficiently by splitting it across attention heads. In particular, once we view the operation as $\partial_{\boldsymbol{x}_t} = \sum_i (\partial_{\boldsymbol{y}_t})_i \boldsymbol{w}_i$, we can see that the attention score between the current token and the prefix token containing $\boldsymbol{w}_i$ must be $(\partial_{\boldsymbol{y}_t})_i$. Using value vectors as rows of $\boldsymbol{W}$ returns the desired output. Similar to the Linear Forward module, we shard the weights into $S'$ parts to parallelize across more attention heads. Please see Figure 4.

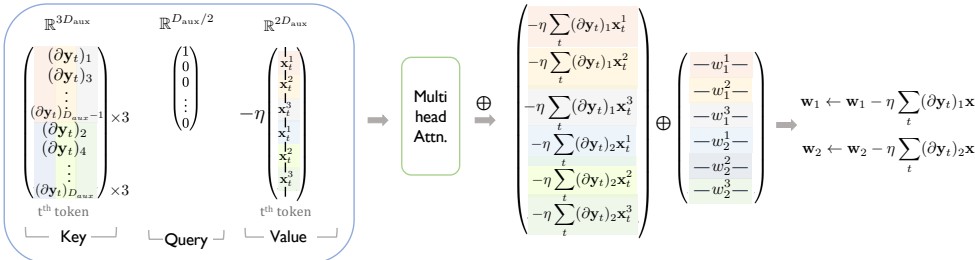

Figure 5: TINT computes the parameter gradients for a linear layer as a $H$-head attention layer ($H = 6$ pictured), with the gradient of the loss w.r.t. linear layer output ($\partial_{\boldsymbol{y}_t}$) as the query, the positional one-hot vector of prefix embeddings as the key, and the input to the linear layer ($\boldsymbol{x}_t$) as the value. The auxiliary model parameters in the prefix embeddings are then updated using a residual connection. Similar to the Linear Forward module (Figure 2), we distribute the dot product computations across all attention heads, by sharding the vectors into $S'$ ($S' = 3$ here) parts. We omitted the identical transformation for query, and value matrices, and permutation-based transformation for key matrix for simplicity.

**Auxiliary's linear descent update**  Finally, the linear descent layer updates the weight and the bias parameters using a batch of inputs $\{\boldsymbol{x}_t\}_{t \leq T_{\mathrm{aux}}}$ and the loss gradient w.r.t. the corresponding outputs $\{\partial_{\boldsymbol{y}_t}\}_{t \leq T_{\mathrm{aux}}}$.

**Definition B.3** (Linear descent). For a weight $\boldsymbol{W} \in \mathbb{R}^{D_{\mathrm{aux}} \times D_{\mathrm{aux}}}$ and a bias $\boldsymbol{b} \in \mathbb{R}^{D_{\mathrm{aux}}}$, the linear descent layer takes in a batch of inputs $\{\boldsymbol{x}_t \in \mathbb{R}^D_{\mathrm{aux}}\}_{t \leq T_{\mathrm{aux}}}$ and gradients $\{\partial_{\boldsymbol{y}_t} \in \mathbb{R}^D_{\mathrm{aux}}\}_{t \leq T_{\mathrm{aux}}}$ and updates the parameters as follows:

$$\boldsymbol{W} \leftarrow \boldsymbol{W} - \eta \sum_{t \leq T_{\mathrm{aux}}} \partial_{\boldsymbol{y}_t} \boldsymbol{x}_t^\top; \qquad \boldsymbol{b} \leftarrow \boldsymbol{b} - \eta \sum_{t \leq T_{\mathrm{aux}}} \partial_{\boldsymbol{y}_t}.$$

**TINT Linear descent module**  The input embedding $\boldsymbol{e}_t$ to this module will contain the gradient of the loss w.r.t. $\boldsymbol{y}_t$, i.e. $\partial_{\boldsymbol{y}_t}$.

As in the Linear backpropagation module, the prefix tokens $\{\boldsymbol{v}_j\}$ will contain the rows of $\boldsymbol{W}$ and $\boldsymbol{b}$, which have been copied from the Linear forward module using residual connections. Since, in addition to the gradients, we also require the input to the linear layer, we will use residual connections to copy the input $\{\boldsymbol{x}_t\}$ to their respective embeddings $\{\boldsymbol{e}_t\}$, from the Linear Forward module. As given in Definition B.3, this module will update $\boldsymbol{W}$ and $\boldsymbol{b}$ using the gradient descent rule.

Focusing on $\boldsymbol{w}_i$, the descent update is given by $\boldsymbol{w}_i \leftarrow \boldsymbol{w}_i - \eta \sum_t (\partial_{\boldsymbol{y}_t})_i \boldsymbol{x}_t$. For the prefix token $\boldsymbol{v}_j$ that contains $\boldsymbol{w}_i$, the update term $-\eta \sum_t (\partial_{\boldsymbol{y}_t})_i \boldsymbol{x}_t$ can be expressed with an attention head that represents the attention between the prefix token $\boldsymbol{v}_j$ and any token $\boldsymbol{e}_t$ with score $(\partial_{\boldsymbol{y}_t})_i$ and value $-\eta \boldsymbol{x}_t$. The residual connection can then be used to update the weights $\boldsymbol{w}_i$ in $\boldsymbol{v}_j$.

For the bias $\boldsymbol{b}$, the descent update is give by $\boldsymbol{b} \leftarrow \boldsymbol{b} - \eta \sum_t \partial_{\boldsymbol{y}_t}$. With $\boldsymbol{b}$ present in $\boldsymbol{v}_1$, we use one attention head to represent the attention score between prefix token $\boldsymbol{v}_1$ and any token $\boldsymbol{e}_t$ as 1, with the value being $-\eta \partial_{\boldsymbol{y}_t}$. The residual connection can then be used to update the weights $\boldsymbol{b}$ in $\boldsymbol{v}_1$.

The above process can be further parallelized across multiple attention heads, by sharding each weight computation into $S'$ parts. Please see Figure 5.

### B.1  $H_{\mathrm{SIM}}$-SPLIT OPERATION

We leverage local structure within the linear operations of TINT to make the construction smaller. We build two $H_{\mathrm{sim}}$-split operations to replace all the linear operations. We use $d_{\mathrm{sim}}$ to denote $D_{\mathrm{sim}}/H_{\mathrm{sim}}$ in the following definitions.

**Definition B.4** (Split-wise $H_{\mathrm{sim}}$-split Linear operation). For weight and bias parameters $\boldsymbol{W}^{\mathrm{TINT}} \in \mathbb{R}^{H_{\mathrm{sim}} \times d_{\mathrm{sim}} \times d_{\mathrm{sim}}}, \boldsymbol{B}^{\mathrm{TINT}} \in \mathbb{R}^{H_{\mathrm{sim}} \times d_{\mathrm{sim}}}$, this layer takes in input $\boldsymbol{e} \in \mathbb{R}^{D_{\mathrm{sim}}}$ and returns $\widetilde{\boldsymbol{e}} = \mathrm{VECTORIZE}(\widetilde{\boldsymbol{S}} + \boldsymbol{B}^{\mathrm{TINT}})$, with $\widetilde{\boldsymbol{S}} \in \mathbb{R}^{H_{\mathrm{sim}} \times d_{\mathrm{sim}}}$ defined with rows $\{\boldsymbol{W}_h^{\mathrm{TINT}} \mathrm{SPLIT}_{H_{\mathrm{sim}}}(\boldsymbol{e})_h\}_{h \leq H_{\mathrm{sim}}}$.

**Definition B.5** (Dimension-wise $H_{\text{sim}}$-split Linear operation). For weight and bias parameters $\boldsymbol{W}^{\text{TINT}} \in \mathbb{R}^{d_{\text{sim}} \times H_{\text{sim}} \times H_{\text{sim}}}$, $\boldsymbol{B}^{\text{TINT}} \in \mathbb{R}^{d_{\text{sim}} \times H_{\text{sim}}}$, this layer takes in input $\boldsymbol{e} \in \mathbb{R}^{D_{\text{sim}}}$, defines $\boldsymbol{S} \in \mathbb{R}^{d_{\text{sim}} \times H_{\text{sim}}}$ with columns $\{\text{SPLIT}_{H_{\text{sim}}}(\boldsymbol{e})_h\}_{h \leq H_{\text{sim}}}$, and returns $\widetilde{\boldsymbol{e}} = \text{VECTORIZE}((\widetilde{\boldsymbol{S}} + \boldsymbol{B}^{\text{TINT}})^{\top})$, where $\widetilde{\boldsymbol{S}} \in \mathbb{R}^{d_{\text{sim}} \times H_{\text{sim}}}$ is defined with rows $\{\boldsymbol{W}_d^{\text{TINT}} \boldsymbol{s}_d^{\text{TINT}}\}_{d \leq d_{\text{sim}}}$.

We find that we can replace all the linear operations with a splitwise $H_{\text{sim}}$-split Linear operation followed by a dimensionwise $H_{\text{sim}}$-split Linear operation, and an additional splitwise $H_{\text{sim}}$-split Linear operation, if necessary. A linear operation on $D_{\text{sim}}$-dimensional space involves $D_{\text{sim}}^2$ parameters, while its replacement requires $D_{\text{sim}}^2 / H_{\text{sim}} + 2D_{\text{sim}} H_{\text{sim}}$ parameters, effectively reducing the total number of necessary parameters by $H_{\text{sim}}$.

We motivate the $H_{\text{sim}}$-split linear operations with an example. We consider the Linear Forward module in Figure 2 for simulating a linear operation with parameters $\boldsymbol{W} \in \mathbb{R}^{D_{\text{aux}} \times D_{\text{aux}}}$ and no biases. For simplicity of presentation, we assume $D_{\text{aux}}$ is divisible by 4. We stack 2 rows of weights per prefix embedding. We distribute the dot-product computation across the $H_{\text{sim}} = 6$ attention heads, by sharding each weight into 3 parts. Since we require to have enough space to store all the sharded computation from the linear attention heads, we require $D_{\text{sim}} = 3D_{\text{aux}}$ (we get 3 values for each of the $D_{\text{aux}}$ weights in $\boldsymbol{W}$). For presentation, for a given vector $\boldsymbol{v} \in \mathbb{R}^{D_{\text{aux}}}$, we represent $\text{SPLIT}_3(\boldsymbol{v})_i$ by $\boldsymbol{v}^i$ for all $1 \leq i \leq 3$.

Now, consider the final linear operation responsible for combining the output of the attention heads. The output, after the linear operation, should contain $\boldsymbol{W}\boldsymbol{x}_t$ in the first $D_{\text{aux}}$ coordinates. At any position $t$, if we stack the output of the linear attention heads as rows of a matrix $\boldsymbol{S}_t \in \mathbb{R}^{H_{\text{sim}} \times D_{\text{sim}}/H_{\text{sim}}}$ we get

$$\boldsymbol{S}_t = \begin{bmatrix} \langle \boldsymbol{w}_1^1, \boldsymbol{x}_t^1 \rangle & \langle \boldsymbol{w}_3^1, \boldsymbol{x}_t^1 \rangle & \langle \boldsymbol{w}_5^1, \boldsymbol{x}_t^1 \rangle & \cdots & \langle \boldsymbol{w}_{D_{\text{aux}}-1}^1, \boldsymbol{x}_t^1 \rangle \\ \langle \boldsymbol{w}_1^2, \boldsymbol{x}_t^2 \rangle & \langle \boldsymbol{w}_3^2, \boldsymbol{x}_t^2 \rangle & \langle \boldsymbol{w}_5^2, \boldsymbol{x}_t^2 \rangle & \cdots & \langle \boldsymbol{w}_{D_{\text{aux}}-1}^2, \boldsymbol{x}_t^2 \rangle \\ \langle \boldsymbol{w}_1^3, \boldsymbol{x}_t^3 \rangle & \langle \boldsymbol{w}_3^3, \boldsymbol{x}_t^3 \rangle & \langle \boldsymbol{w}_5^3, \boldsymbol{x}_t^3 \rangle & \cdots & \langle \boldsymbol{w}_{D_{\text{aux}}-1}^3, \boldsymbol{x}_t^3 \rangle \\ \langle \boldsymbol{w}_2^1, \boldsymbol{x}_t^1 \rangle & \langle \boldsymbol{w}_4^1, \boldsymbol{x}_t^1 \rangle & \langle \boldsymbol{w}_6^1, \boldsymbol{x}_t^1 \rangle & \cdots & \langle \boldsymbol{w}_{D_{\text{aux}}}^1, \boldsymbol{x}_t^1 \rangle \\ \langle \boldsymbol{w}_2^2, \boldsymbol{x}_t^2 \rangle & \langle \boldsymbol{w}_4^2, \boldsymbol{x}_t^2 \rangle & \langle \boldsymbol{w}_6^2, \boldsymbol{x}_t^2 \rangle & \cdots & \langle \boldsymbol{w}_{D_{\text{aux}}}^2, \boldsymbol{x}_t^2 \rangle \\ \langle \boldsymbol{w}_2^3, \boldsymbol{x}_t^3 \rangle & \langle \boldsymbol{w}_4^3, \boldsymbol{x}_t^3 \rangle & \langle \boldsymbol{w}_6^3, \boldsymbol{x}_t^3 \rangle & \cdots & \langle \boldsymbol{w}_{D_{\text{aux}}}^3, \boldsymbol{x}_t^3 \rangle \end{bmatrix}$$

Note that for each $j \leq D_{\text{aux}}$, we have $\langle \boldsymbol{w}_j, \boldsymbol{x}_t \rangle = \sum_{i=1}^3 \langle \boldsymbol{w}_j^i, \boldsymbol{x}_t^i \rangle$. Thus, with a column-wise linear operation on $\boldsymbol{S}_t$, we can sum the relevant elements in each column to get

$\boldsymbol{S}_t^{col} =$

$$\begin{bmatrix} \langle \boldsymbol{w}_1, \boldsymbol{x}_t \rangle & \langle \boldsymbol{w}_3, \boldsymbol{x}_t \rangle & \cdots & \langle \boldsymbol{w}_{D_{\text{aux}}/2-1}, \boldsymbol{x}_t \rangle & 0 & 0 & \cdots & 0 \\ \langle \boldsymbol{w}_2, \boldsymbol{x}_t \rangle & \langle \boldsymbol{w}_4, \boldsymbol{x}_t \rangle & \cdots & \langle \boldsymbol{w}_{D_{\text{aux}}/2}, \boldsymbol{x}_t \rangle & 0 & 0 & \cdots & 0 \\ 0 & 0 & \cdots & 0 & \langle \boldsymbol{w}_{D_{\text{aux}}/2+1}, \boldsymbol{x}_t \rangle & \langle \boldsymbol{w}_{D_{\text{aux}}/2+3}, \boldsymbol{x}_t \rangle & \cdots & \langle \boldsymbol{w}_{D_{\text{aux}}-1}, \boldsymbol{x}_t \rangle \\ 0 & 0 & \cdots & 0 & \langle \boldsymbol{w}_{D_{\text{aux}}/2+2}, \boldsymbol{x}_t \rangle & \langle \boldsymbol{w}_{D_{\text{aux}}/2+4}, \boldsymbol{x}_t \rangle & \cdots & \langle \boldsymbol{w}_{D_{\text{aux}}}, \boldsymbol{x}_t \rangle \\ 0 & 0 & \cdots & 0 & 0 & 0 & \cdots & 0 \\ 0 & 0 & \cdots & 0 & 0 & 0 & \cdots & 0 \end{bmatrix}$$

A row-wise linear operation on $\boldsymbol{S}_t^{col}$ can space out the non-zero elements in the matrix and give us

$\boldsymbol{S}_t^{row} =$

$$\begin{bmatrix} \langle \boldsymbol{w}_1, \boldsymbol{x}_t \rangle & 0 & \langle \boldsymbol{w}_3, \boldsymbol{x}_t \rangle & 0 & \cdots & \langle \boldsymbol{w}_{D_{\text{aux}}/2-1}, \boldsymbol{x}_t \rangle & 0 \\ 0 & \langle \boldsymbol{w}_2, \boldsymbol{x}_t \rangle & 0 & \langle \boldsymbol{w}_4, \boldsymbol{x}_t \rangle & \cdots & 0 & \langle \boldsymbol{w}_{D_{\text{aux}}/2}, \boldsymbol{x}_t \rangle \\ \langle \boldsymbol{w}_{D_{\text{aux}}/2+1}, \boldsymbol{x}_t \rangle & 0 & \langle \boldsymbol{w}_{D_{\text{aux}}/2+3}, \boldsymbol{x}_t \rangle & 0 & \cdots & \langle \boldsymbol{w}_{D_{\text{aux}}-1}, \boldsymbol{x}_t \rangle & 0 \\ 0 & \langle \boldsymbol{w}_{D_{\text{aux}}/2+2}, \boldsymbol{x}_t \rangle & 0 & \langle \boldsymbol{w}_{D_{\text{aux}}/2+4}, \boldsymbol{x}_t \rangle & \cdots & 0 & \langle \boldsymbol{w}_{D_{\text{aux}}}, \boldsymbol{x}_t \rangle \\ 0 & 0 & \cdots & 0 & \cdots & 0 & 0 \\ 0 & 0 & \cdots & 0 & \cdots & 0 & 0 \end{bmatrix}$$

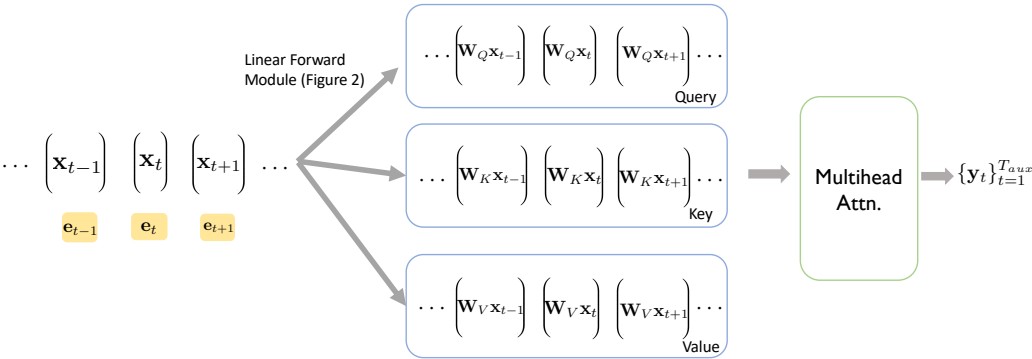

Figure 6: TINT simulates the forward pass of a self-attention layer of the auxiliary model with a Linear Forward module (Figure 2) and a TINT softmax attention layer (Definition A.1). The Linear Forward module computes the query, key, and value vectors using a Linear Forward module on the current embeddings, changing the prefix embeddings to correspond to $\boldsymbol{W}_Q$, $\boldsymbol{W}_K$, and $\boldsymbol{W}_K$ respectively.

Finally, a column-wise linear operation on $\boldsymbol{S}_t^{row}$ helps to get the non-zero elements in the correct order.

$$\bar{\boldsymbol{S}}_t^{col} =$$

$$\begin{bmatrix} \langle \boldsymbol{w}_1, \boldsymbol{x}_t \rangle & \langle \boldsymbol{w}_2, \boldsymbol{x}_t \rangle & \langle \boldsymbol{w}_3, \boldsymbol{x}_t \rangle & \langle \boldsymbol{w}_4, \boldsymbol{x}_t \rangle & \cdots & \langle \boldsymbol{w}_{D_{\mathrm{aux}}/2-1}, \boldsymbol{x}_t \rangle & \langle \boldsymbol{w}_{D_{\mathrm{aux}}/2}, \boldsymbol{x}_t \rangle \\ \langle \boldsymbol{w}_{D_{\mathrm{aux}}/2+1}, \boldsymbol{x}_t \rangle & \langle \boldsymbol{w}_{D_{\mathrm{aux}}/2+2}, \boldsymbol{x}_t \rangle & \langle \boldsymbol{w}_{D_{\mathrm{aux}}/2+3}, \boldsymbol{x}_t \rangle & \langle \boldsymbol{w}_{D_{\mathrm{aux}}/2+4}, \boldsymbol{x}_t \rangle & \cdots & \langle \boldsymbol{w}_{D_{\mathrm{aux}}-1}, \boldsymbol{x}_t \rangle & \langle \boldsymbol{w}_{D_{\mathrm{aux}}}, \boldsymbol{x}_t \rangle \\ 0 & 0 & 0 & 0 & \cdots & 0 & 0 \\ \vdots & \vdots & \vdots & \vdots & \cdots & \vdots & \vdots \\ 0 & 0 & 0 & 0 & \cdots & 0 & 0 \end{bmatrix}$$

The desired output is then given by VECTORIZE($\{\bar{\boldsymbol{s}}_{t,j}^{col}\}_{j=1}^{D_{\mathrm{aux}}}$), which contains $\boldsymbol{W}\boldsymbol{x}_t$ in the first $D_{\mathrm{aux}}$ coordinates. The operations that convert $\boldsymbol{S}_t$ to $\boldsymbol{S}_t^{col}$ and $\boldsymbol{S}_t^{row}$ to $\bar{\boldsymbol{S}}_t^{row}$ represents a split-wise 6-split linear operation, while the operation that converts $\boldsymbol{S}_t^{col}$ to $\boldsymbol{S}_t^{row}$ represents a dimension-wise 6-split linear operation. A naive linear operation on the output of the attention heads would require $D_{\mathrm{sim}}^2$ parameters, while its replacement requires $D_{\mathrm{sim}}^2/6$ parameters to represent a dimension-wise 6-split linear operation, and an additional $12D_{\mathrm{sim}}$ parameters to represent the split-wise 6-split linear operations.

## C SELF-ATTENTION LAYER

We first introduce multi-head attention, generalizing single-head attention (Definition 2.10).

**Definition C.1** (Auxiliary self-attention with $H_{\mathrm{aux}}$ heads). For query, key, and value weights $\boldsymbol{W}_Q, \boldsymbol{W}_K, \boldsymbol{W}_V \in \mathbb{R}^{D_{\mathrm{aux}} \times D_{\mathrm{aux}}}$ and bias $\boldsymbol{b}_Q, \boldsymbol{b}_K, \boldsymbol{b}_V \in \mathbb{R}^{D_{\mathrm{aux}}}$, a self-attention layer with $H_{\mathrm{aux}}$ attention heads and a function $f_{\mathrm{attn}} : \mathbb{R}^{T_{\mathrm{aux}}} \to \mathbb{R}^{T_{\mathrm{aux}}}$ takes a sequence $\{\boldsymbol{x}_t \in \mathbb{R}^{D_{\mathrm{aux}}}\}_{t \leq T_{\mathrm{aux}}}$ as input and outputs $\{\boldsymbol{y}_t\}_{t \leq T_{\mathrm{aux}}}$, with

$$\boldsymbol{y}_t = \mathrm{VECTORIZE}(\{ \sum_{j \leq T_{\mathrm{aux}}} a_{t,j}^h \boldsymbol{v}_j^h \}_{h \leq H_{\mathrm{aux}}}). \tag{6}$$

$a_{t,j}^h$ is defined as the attention score of head $h$ between tokens at positions $t$ and $j$, and is given by

$$a_{t,j}^h = \mathrm{softmax}(\boldsymbol{K}^h \boldsymbol{q}_t^h)_j. \tag{7}$$

Here, $\boldsymbol{q}_t$, $\boldsymbol{k}_t$, $\boldsymbol{v}_t$ denote the query, key, and value vectors at each position $t$, computed as $\boldsymbol{W}_Q \boldsymbol{x}_t + \boldsymbol{b}_Q$, $\boldsymbol{W}_K \boldsymbol{x}_t + \boldsymbol{b}_K$, and $\boldsymbol{W}_V \boldsymbol{x}_t + \boldsymbol{b}_V$ respectively. In addition, $\boldsymbol{q}_t^h$, $\boldsymbol{k}_t^h$, $\boldsymbol{v}_t^h$ denote $\mathrm{SPLIT}_{H_{\mathrm{aux}}}(\boldsymbol{q}_t)_h$, $\mathrm{SPLIT}_{H_{\mathrm{aux}}}(\boldsymbol{k}_t)_h$, and $\mathrm{SPLIT}_{H_{\mathrm{aux}}}(\boldsymbol{v}_t)_h$ respectively for all $t \leq T_{\mathrm{aux}}$, and $h \leq H_{\mathrm{aux}}$. $\boldsymbol{K}^h \in \mathbb{R}^{T_{\mathrm{aux}} \times D_{\mathrm{aux}}}$ is defined with its rows as $\{\boldsymbol{k}_t^h\}_{t \leq T_{\mathrm{aux}}}$ for all $h \leq H_{\mathrm{aux}}$.

Figure 7: The gradient w.r.t. the value vectors $\{\partial_{\boldsymbol{v}_t}\}$ (Definition C.2) forms the integral component for both TINT self-attention backward and descent update modules. TINT computes $\{\partial_{\boldsymbol{v}_t}\}$ using a softmax attention and a linear attention layer. We first use residual connections to copy the query and key vectors to the current embeddings from the TINT Self-attention Forward module (Figure 6). The softmax attention layer re-computes the attention scores $\{a_{t,j}^h\}$ between all token pairs $\{(t,j)\}$ and stores them in the token embeddings. The linear attention layer uses the one-hot position embeddings of the input tokens as the query to use the transposed attention scores $\{a_{j,t}^h\}$ for all token pairs $\{(t,j)\}$ and use the gradients $\{\partial_{\boldsymbol{y}_t}\}$ as the value vectors to compute $\{\partial_{\boldsymbol{v}_t}\}$.

In the discussions below, we consider a self-attention layer in the auxiliary model with parameters $\{\boldsymbol{W}_Q, \boldsymbol{b}_Q, \boldsymbol{W}_K, \boldsymbol{b}_K, \boldsymbol{W}_V, \boldsymbol{b}_V\}$ that takes in input sequence $\boldsymbol{x}_1, \cdots, \boldsymbol{x}_{T_{\text{aux}}}$ and outputs $\boldsymbol{y}_1, \cdots, \boldsymbol{y}_{T_{\text{aux}}}$, with $\{\boldsymbol{y}_t\}_{t=1}^{T_{\text{aux}}}$ given by (6). As in the definition, $\boldsymbol{q}_t, \boldsymbol{k}_t, \boldsymbol{v}_t$ denote the query, key, and value vectors for position $t$. We will use TINT self-attention modules in order to simulate the operations on the auxiliary's self-attention layer. To do so, we will need $H_{\text{sim}} \geq H_{\text{aux}}$ in the corresponding TINT self-attention modules.

**TINT Self-attention forward module** The input embedding to this module $\boldsymbol{e}_t$ at each position $t$ will contain $\boldsymbol{x}_t$ in its first $D_{\text{aux}}$ coordinates. The self-attention module can be divided into four sub-operations: Computation of (a) query vectors $\{\boldsymbol{q}_t\}_{t \leq T}$, (b) key vectors $\{\boldsymbol{k}_t\}_{t \leq T}$, (c) value vectors $\{\boldsymbol{v}_t\}_{t \leq T}$, and (d) $\{\boldsymbol{y}_t\}_{t \leq T}$ using (6). Please see Figure 6.

- Sub-operations (a): The computation of query vector $\boldsymbol{q}_t := \boldsymbol{W}_Q \boldsymbol{x}_t + \boldsymbol{b}_Q$ at each position $t$ is a linear operation involving parameters $\boldsymbol{W}_Q, \boldsymbol{b}_Q$. Thus, we can first feed in the stacked rows of $\boldsymbol{W}_Q$ and $\boldsymbol{b}_Q$ onto the prefix embeddings $\{\boldsymbol{v}_j\}$. We use a Linear Forward module (Appendix B) on the current embeddings and the prefix embeddings to get embedding $\boldsymbol{e}_t^q$ at each position $t$ that contains $\boldsymbol{q}_t$ in the first $D_{\text{aux}}$ coordinates.

- Sub-operations (b, c): Similar to (a), we feed in the stacked rows of the necessary parameters onto the prefix embeddings $\{\boldsymbol{v}_j\}$, and call two Linear Forward Modules (Appendix B) independently to get embeddings $\boldsymbol{e}_t^k$, and $\boldsymbol{e}_t^v$ containing $\boldsymbol{k}_t$ and $\boldsymbol{v}_t$ respectively.
  We now combine the embeddings $\boldsymbol{e}_t^q$, $\boldsymbol{e}_t^k$, and $\boldsymbol{e}_t^v$ to get an embedding $\boldsymbol{e}_t$ that contain $\boldsymbol{q}_t, \boldsymbol{k}_t, \boldsymbol{v}_t$ in the first $3D_{\text{aux}}$ coordinates.

- Sub-operation (d): Finally, we call a TINT self-attention module (Definition A.1) on our current embeddings $\{\boldsymbol{e}_t\}_{t \leq T}$ to compute $\{\boldsymbol{y}_t\}_{t \leq T}$. The query, key, and value parameters in the self-attention module contain sub-Identity blocks that pick out the relevant information from $\boldsymbol{q}_t, \boldsymbol{k}_t, \boldsymbol{v}_t$ stored in $\boldsymbol{e}_t$.

*Remark:* Sub-operations (a), (b), and (c) can be represented as a single linear operation with a weight $\boldsymbol{W} \in \mathbb{R}^{3D_{\text{aux}} \times D_{\text{aux}}}$ by concatenating the rows of $\{\boldsymbol{W}_Q, \boldsymbol{W}_K, \boldsymbol{W}_V\}$ and a bias $\boldsymbol{b} \in \mathbb{R}^{3D_{\text{aux}}}$ that concatenates $\{\boldsymbol{b}_Q, \boldsymbol{b}_K, \boldsymbol{b}_V\}$. Thus, they can be simulated with a single Linear Forward Module, with $\boldsymbol{W}, \boldsymbol{b}$ fed into the prefix embeddings. However, we decide to separate them in order to limit the number of prefix embeddings and the embedding size. E.g. for GPT-2, $D_{\text{aux}} = 768$. This demands either a $3\times$ increase in the embedding size in TINT or a $3\times$ increase in the number of prefix embeddings. Hence, in order to minimize the parameter cost, we call Linear Forward Module separately to compute $\boldsymbol{q}_t, \boldsymbol{k}_t,$ and $\boldsymbol{v}_t$ at each position $t$.

**Auxiliary's backpropagation through self-attention**    For an auxiliary self-attention layer as defined in Definition C.1, the backpropagation layer takes in the loss gradient w.r.t. output ($\{\partial_{\boldsymbol{y}_t}\}_{t \leq T_{\text{aux}}}$) and computes the loss gradient w.r.t. input token ($\{\partial_{\boldsymbol{x}_t}\}_{t \leq T_{\text{aux}}}$).

**Definition C.2.** [Auxiliary self-attention backpropagation] For query, key, and value weights $\boldsymbol{W}_Q, \boldsymbol{W}_K, \boldsymbol{W}_V \in \mathbb{R}^{D_{\text{aux}} \times D_{\text{aux}}}$ and bias $\boldsymbol{b}_Q, \boldsymbol{b}_K, \boldsymbol{b}_V \in \mathbb{R}^{D_{\text{aux}}}$, the backpropagation layer corresponding to a self-attention layer with $H_{\text{aux}}$ attention heads takes a sequence $\{\partial_{\boldsymbol{y}_t} \in \mathbb{R}^{D_{\text{aux}}}\}_{t \leq T_{\text{aux}}}$ and $\{\boldsymbol{x}_t \in \mathbb{R}^{D_{\text{aux}}}\}_{t \leq T_{\text{aux}}}$ as input and outputs $\{\partial_{\boldsymbol{x}_t}\}_{t \leq T_{\text{aux}}}$, with

$$\partial_{\boldsymbol{x}_t} = \boldsymbol{W}_Q^\top \partial_{\boldsymbol{q}_t} + \boldsymbol{W}_K^\top \partial_{\boldsymbol{k}_t} + \boldsymbol{W}_V^\top \partial_{\boldsymbol{v}_t}, \quad \text{with}$$

$$\partial_{\boldsymbol{q}_t} = \text{VECTORIZE}(\{\sum_j a_{t,j}^h ((\partial_{\boldsymbol{y}_t^h})^\top \boldsymbol{v}_j^h)[\boldsymbol{k}_j^h - \sum_{j'} a_{t,j'}^h \boldsymbol{k}_{j'}^h]\}_{h \leq H_{\text{aux}}});$$

$$\partial_{\boldsymbol{k}_t} = \text{VECTORIZE}(\{\sum_j a_{j,t}^h \boldsymbol{q}_j^h [(\partial_{\boldsymbol{y}_j^h})^\top (\boldsymbol{v}_t^h - \sum_{j'} a_{j,j'}^h \boldsymbol{v}_{j'}^h)]\}_{h \leq H_{\text{aux}}});$$

$$\partial_{\boldsymbol{v}_t} = \text{VECTORIZE}(\{\sum_j a_{j,t}^h \partial_{\boldsymbol{y}_j^h}\}_{h \leq H_{\text{aux}}})$$

Here, $\boldsymbol{q}_t, \boldsymbol{k}_t$, and $\boldsymbol{v}_t$ refer to query, key, and value vectors at each position $t$, with the attention scores $\{a_{t,j}^h\}_{t,j \leq T_{\text{aux}}, h \leq H_{\text{aux}}}$.

**Complexity of true backpropagation**    The much-involved computation in the above operation is due to the computation of $\partial_{\boldsymbol{q}_t}$ and $\partial_{\boldsymbol{k}_t}$ at each position $t$. For the following discussion, we assume that our current embeddings $\boldsymbol{e}_t$ contain $\boldsymbol{q}_t, \boldsymbol{k}_t, \boldsymbol{v}_t$, in addition to the gradient $\partial_{\boldsymbol{y}_t}$. The computation of $\partial_{\boldsymbol{q}_t}$ (and similarly $\partial_{\boldsymbol{k}_t}$) at any position $t$ involves the following sequential computations and the necessary TINT modules.

- $\{\{(\partial_{\boldsymbol{y}_t^h})^\top \boldsymbol{v}_j^h\}_{j \leq T_{\text{aux}}}\}_{h \leq H_{\text{aux}}}$ with a TINT linear self-attention module (Definition A.1), with atleast $H_{\text{aux}}$ attention heads that represent the attention score between $\boldsymbol{e}_t$ and any other token $\boldsymbol{e}_j$, by $\{(\partial_{\boldsymbol{y}_t^h})^\top \boldsymbol{v}_j^h\}_{h \leq H_{\text{aux}}}$.

- Attention scores $\{a_{t,j}^h\}_{h \leq H_{\text{aux}}}$, which requires a TINT softmax self-attention module (Definition A.1), with at least $H_{\text{aux}}$ heads, that uses the already present $\{\boldsymbol{q}_t, \boldsymbol{k}_t, \boldsymbol{v}_t\}$ in the current embeddings $\boldsymbol{e}_t$ to re-compute the attention scores.

- $\{a_{t,j}^h (\partial_{\boldsymbol{y}_t^h})^\top \boldsymbol{v}_j^h\}_{h \leq H_{\text{aux}}}$ for all $j \leq T_{\text{aux}}$ by multiplying the attention scores $\{a_{t,j}^h\}_{h \leq H_{\text{aux}}}$ with $\{(\partial_{\boldsymbol{y}_t^h})^\top \boldsymbol{v}_j^h\}_{h \leq H_{\text{aux}}}$ using an MLP layer (Lemma A.4). Furthermore, $\{\sum_j a_{t,j}^h \boldsymbol{k}_j^h\}_{h \leq H_{\text{aux}}}$ needs to be computed in parallel as well, with additional attention heads.

- $\partial_{\boldsymbol{y}_t}$ with a TINT linear self-attention module (Definition A.1), with atleast $H_{\text{aux}}$ attention heads that represent the attention score between any token $\boldsymbol{e}_j$ and $\boldsymbol{e}_t$ by $\{a_{t,j}^h (\partial_{\boldsymbol{y}_t^h})^\top \boldsymbol{v}_j^h\}_{h \leq H_{\text{aux}}}$, with value vectors given by $\{\boldsymbol{k}_j^h - \sum_{j'} a_{t,j'}^h \boldsymbol{k}_{j'}^h\}_{h \leq H_{\text{aux}}}$.

The sequential computation requires the simulator to store $\{\{(\partial_{\boldsymbol{y}_t^h})^\top \boldsymbol{v}_j^h\}_{j \leq T_{\text{aux}}}\}_{h \leq H_{\text{aux}}}$ and $\{a_{t,j}^h\}_{h \leq H_{\text{aux}}}$ in the token embedding $\boldsymbol{e}_t$, which requires an additional $2 T_{\text{aux}} H_{\text{aux}}$ embedding dimension size. To avoid the much-involved computation for the true gradient propagation, we instead only use the gradients w.r.t. $\boldsymbol{v}_t$.

**Approximate auxiliary self-attention backpropagation**    We formally extend the definition of approximate gradients $\{\partial_{\boldsymbol{x}_t}\}_{t=1}^{T_{\text{aux}}}$ from Definition 2.12 to multi-head attention in Definition C.6.

In the upcoming theorem, we formally show that if on a given sequence $\{\boldsymbol{x}_t\}_{t \leq T_{\text{aux}}}$, for all token positions all the attention heads in a self-attention layer primarily attend to a single token, then the approximate gradient $\widehat{\partial_{\boldsymbol{x}_t}}$ is close to the true gradient $\partial_{\boldsymbol{x}_t}$ at each position $t$.

**Definition C.3** ($\varepsilon$-hard attention head)**.**    For the Self-Attention layer of $H_{\text{aux}}$ heads in Definition C.1, on a given input sequence $\{\boldsymbol{x}_t\}_{t=1}^{T_{\text{aux}}}$, an attention head $h \leq H_{\text{aux}}$ is defined to be $\varepsilon$-hard on the input sequence, if for all positions $t \leq T_{\text{aux}}$, there exists a position $t_0 \leq T_{\text{aux}}$ such that $a_{t,t_0}^h \geq 1 - \varepsilon$.

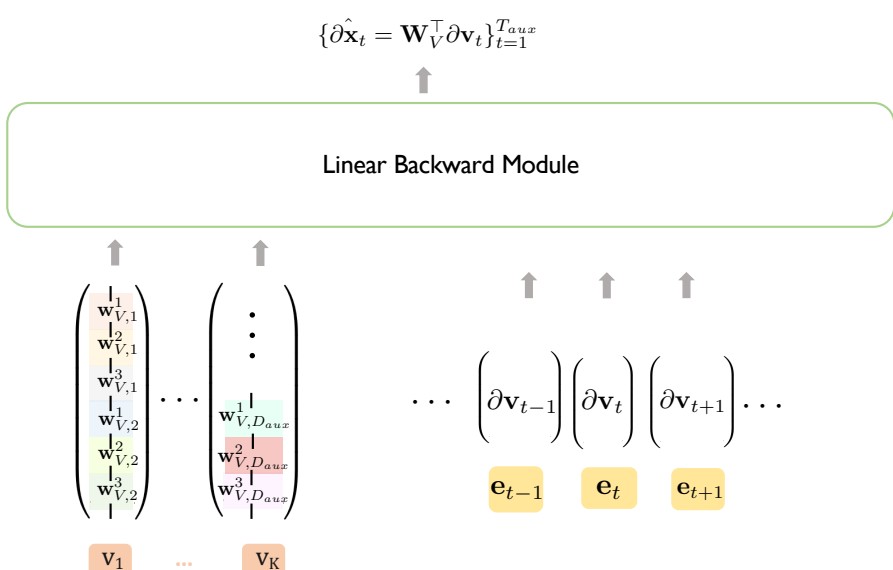

Figure 8: TINT simulates the backward pass of a self-attention layer of the auxiliary model using a Linear Backward module (Figure 4). The input embeddings contain the gradient of the loss w.r.t. the value vectors ($\partial_{v_t}$) computed in Figure 7. The value matrix $W_V$ is encoded in the prefix embeddings. We call the Linear Backward module on this sequence.

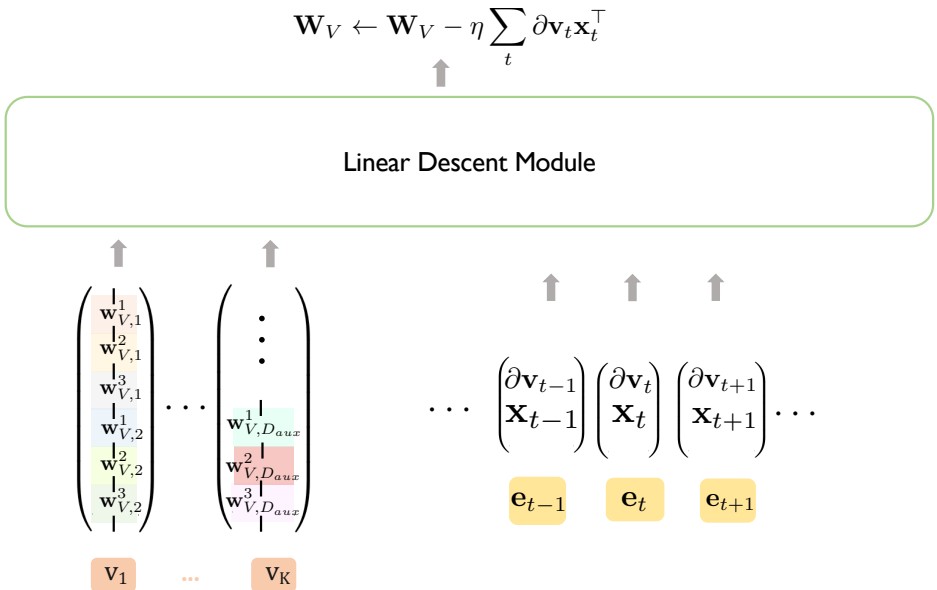

Figure 9: TINT simulates the backward pass of the self-attention layer in the auxiliary model by employing the Linear Descent module (Figure 5). The input embeddings consist of the gradient of the loss with respect to the value vectors ($\partial_{v_t}$) computed in Figure 7. Additionally, we incorporate a residual connection to copy the input from the Self-attention Forward module (Figure 6) into $x_t$. Before invoking the Linear Descent module, we represent the value parameters ($W_V$) into the prefix embeddings. TINT simulates the backward pass of a self-attention layer of the auxiliary model using a Linear Descent module (Figure 5).

**Theorem C.4.** *With the notations in Definitions 2.12, C.1 and C.2, if on a given input sequence $\{\boldsymbol{x}_t\}_{t=1}^{T_{aux}}$, with its query, key, and value vectors $\{\boldsymbol{q}_t, \boldsymbol{k}_t, \boldsymbol{v}_t\}_{t=1}^{T_{aux}}$, all the $H_{aux}$ attention heads are $\varepsilon$-hard for some $\varepsilon > 0$, then for a given sequence of gradients $\{\partial_{\boldsymbol{y}_t}\}_{t=1}^{T_{aux}}$,*

$$\|\partial_{\boldsymbol{q}_t}\|_2, \|\partial_{\boldsymbol{k}_t}\|_2 \leq \mathcal{O}(\varepsilon B_x^2 B_w^2 B_y), \quad \text{for all } t \leq T_{aux},$$

*where* $B_x = \max_{t \leq T_{aux}} \|\boldsymbol{x}_t\|_2$, $B_y = \max_{t \leq T_{aux}} \|\partial_{\boldsymbol{y}_t}\|_2$, *and* $B_w = \max\{\|\boldsymbol{W}_K\|_2, \|\boldsymbol{W}_Q\|_2, \|\boldsymbol{W}_V\|_2, \|\boldsymbol{b}_V\|_2, \|\boldsymbol{b}_K\|_2, \|\boldsymbol{b}_V\|_2\}$.

*This implies, for each position $t$,* $\left\|\widehat{\partial_{\boldsymbol{x}_t}} - \partial_{\boldsymbol{x}_t}\right\|_2 \leq \mathcal{O}(\varepsilon B_x^2 B_w^3 B_y)$.

**TINT Self-attention backpropagation module** The input embeddings $\boldsymbol{e}_t$ contain $\partial_{\boldsymbol{y}_t}$ in the first $D_{\text{aux}}$ coordinates. Since we require to re-compute the attention scores $\{a_{t,j}^h\}_{j \leq T_{\text{aux}}, h \leq H_{\text{aux}}}$, we need to copy the query, key, and value vectors $\boldsymbol{q}_t, \boldsymbol{k}_t$, and $\boldsymbol{v}_t$ from the TINT self-attention Forward module at each position $t$. Furthermore, we use the residual connection to copy the prefix embeddings $\{\boldsymbol{v}_j\}$, which contain the rows of $\boldsymbol{W}_V$, from the TINT self-attention Forward module.

The operation can be divided into three sub-operations: Computing (a) attention scores $\{a_{t,j}^h\}_{h \leq H_{\text{aux}}}$ for all $j \leq T_{\text{aux}}$, at each position $t$, (b) $\partial_{\boldsymbol{v}_t}$ from $\{a_{t,j}^h\}_{h \leq H_{\text{aux}}}$ and $\partial_{\boldsymbol{y}_t}$, and (c) $\widehat{\partial_{\boldsymbol{x}_t}}$ from $\partial_{\boldsymbol{v}_t}$.

- Sub-operation (a): Since, the current embeddings $\boldsymbol{e}_t$ contain $\boldsymbol{q}_t, \boldsymbol{k}_t$, we can simply call a self-attention attention module to compute the attention scores $\{a_{t,j}^h\}_{h \leq H_{\text{aux}}}$ for all $j \leq T$ and store them in the current embeddings. We further retain $\partial_{\boldsymbol{y}_t}$ and $\boldsymbol{v}_t$ for further operations using residual connections.

- Sub-operation (b): With the current embeddings $\boldsymbol{e}_t$ containing the attention scores $\{a_{t,j}^h\}_{h \leq H_{\text{aux}}}$ for all $j \leq T$, and the gradient $\partial_{\boldsymbol{y}_t}$, we can compute $\partial_{\boldsymbol{v}_t}$ using a TINT linear self-attention module with atleast $H_{\text{aux}}$ attention heads, that represent the attention scores between tokens $\boldsymbol{e}_t$ and $\boldsymbol{e}_j$ for any $j$ as $\{a_{j,t}^h\}_{h \leq H_{\text{aux}}}$ and use $\text{SPLIT}_{H_{\text{aux}}}(\partial_{\boldsymbol{y}_t})$ as their value vectors.

- Sub-operation (c): And finally, the computation of $\widehat{\partial_{\boldsymbol{x}_t}}$ is identical to the backpropagation through a linear layer, with parameters $\boldsymbol{W}_V$ and $\boldsymbol{b}_V$. Hence, we call a Linear backpropagation module on the current embeddings, that contain $\partial_{\boldsymbol{y}_t}$ and the prefix embeddings that contain $\boldsymbol{W}_V$ and $\boldsymbol{b}_V$.

**Separating sub-operations (a) and (b)** The operation for computing $\partial_{\boldsymbol{v}_t}$ in Definition 2.12 looks very similar to the computation of $\boldsymbol{y}_t$ in Equation (6). However, the major difference is that instead of the attention scores being $\{a_{t,j}^h\}_{h \leq H_{\text{aux}}}$ between token $t$ and any token $j$, we need the attention scores to be $\{a_{j,t}^h\}_{h \leq H_{\text{aux}}}$. Thus, unless our model allows a transpose operation on the attention scores, we need to first store them in our embeddings and then use an additional self-attention module that can pick the right attention scores between tokens using position embeddings. Please see Figure 8.

**Auxiliary's value descent update** Similar to the complexity of true backpropagation, the descent updates for $\boldsymbol{W}_Q, \boldsymbol{b}_Q, \boldsymbol{W}_K, \boldsymbol{b}_K$ are quite expensive to express with the transformer layers. Hence, we focus simply on updating on $\boldsymbol{W}_V, \boldsymbol{b}_V$, while keeping the others fixed.

**Definition C.5** (Auxiliary self-attention value descent). For query, key, and value weights $\boldsymbol{W}_Q, \boldsymbol{W}_K, \boldsymbol{W}_V \in \mathbb{R}^{D_{\text{aux}} \times D_{\text{aux}}}$ and bias $\boldsymbol{b}_Q, \boldsymbol{b}_K, \boldsymbol{b}_V \in \mathbb{R}^{D_{\text{aux}}}$, the value descent layer corresponding to a self-attention layer with $H_{\text{aux}}$ attention heads and any function $f_{\text{attn}} : \mathbb{R}^{T_{\text{aux}}} \to \mathbb{R}^{T_{\text{aux}}}$ takes in a batch of gradients $\{\partial_{\boldsymbol{y}_t} \in \mathbb{R}^{D_{\text{aux}}}\}_{t \leq T_{\text{aux}}}$ and inputs $\{\boldsymbol{x}_t \in \mathbb{R}^{D_{\text{aux}}}\}_{t \leq T_{\text{aux}}}$ and updates $\boldsymbol{W}_V, \boldsymbol{b}_V$ as follows:

$$\boldsymbol{W}_V \leftarrow \boldsymbol{W}_V - \eta \sum_{t \leq T_{\text{aux}}} \partial_{\boldsymbol{v}_t} \boldsymbol{x}_t^\top, \quad \boldsymbol{b}_V \leftarrow \boldsymbol{b}_V - \eta \sum_{t \leq T_{\text{aux}}} \partial_{\boldsymbol{v}_t},$$

$$\text{where } \partial_{\boldsymbol{v}_t} = \text{VECTORIZE}(\{\sum_j a_{j,t}^h \partial_{\boldsymbol{y}_j^h}\}_{h \leq H_{\text{aux}}})$$

Here, $\boldsymbol{v}_t$ refers to value vectors at each position $t$, as defined in Definition C.1.

**TINT Self-attention descent module**   The input embeddings contain $\partial_{\boldsymbol{v}_t}$ in the first $D_{\text{aux}}$ coordinates, from the TINT self-attention backpropagation module. Furthermore, the prefix embeddings $\{\boldsymbol{v}_j\}$ contain the stacked rows of $\boldsymbol{W}_V$ and $\boldsymbol{b}_V$, continuing from the TINT self-attention backpropagation module.

Since we further need the input $\boldsymbol{x}_t$ to the auxiliary self-attention layer under consideration, we use residual connections to copy $\boldsymbol{x}_t$ from the TINT self-attention Forward module at each position $t$.

The updates of $\boldsymbol{W}_V$ and $\boldsymbol{b}_V$ are equivalent to the parameter update in a linear layer, involving gradients $\{\partial_{\boldsymbol{v}_t}\}$ and input $\{\boldsymbol{x}_t\}$. Thus, we call a Linear descent module on the current embeddings and the prefix embeddings to get the updated value parameters. Please see Figure 9.

### C.1   APPROXIMATE AUXILIARY SELF-ATTENTION BACKPROPAGATION

**Definition C.6.** For query, key, and value weights $\boldsymbol{W}_Q, \boldsymbol{W}_K, \boldsymbol{W}_V \in \mathbb{R}^{D_{\text{aux}} \times D_{\text{aux}}}$ and bias $\boldsymbol{b}_Q, \boldsymbol{b}_K, \boldsymbol{b}_V \in \mathbb{R}^{D_{\text{aux}}}$, the approximate backpropagation layer corresponding to a self-attention layer with $H_{\text{aux}}$ attention heads takes a sequence $\{\partial_{\boldsymbol{y}_t} \in \mathbb{R}^{D_{\text{aux}}}\}_{t \leq T_{\text{aux}}}$ and $\{\boldsymbol{x}_t \in \mathbb{R}^{D_{\text{aux}}}\}_{t \leq T_{\text{aux}}}$ as input and outputs $\{\partial_{\boldsymbol{x}_t} := \text{VECTORIZE}(\{\partial_{\boldsymbol{x}_t^h}\}_{h \leq H_{\text{aux}}})\}_{t \leq T_{\text{aux}}}$, with

$$\widehat{\partial_{\boldsymbol{x}_t}} = \boldsymbol{W}_V^\top \partial_{\boldsymbol{v}_t}, \quad \text{where } \partial_{\boldsymbol{v}_t} = \text{VECTORIZE}(\{\sum_j a_{j,t}^h \partial_{\boldsymbol{y}_j^h}\}_{h \leq H_{\text{aux}}})$$

Here, $\boldsymbol{q}_t$, $\boldsymbol{k}_t$, and $\boldsymbol{v}_t$ refer to query, key, and value vectors at each position $t$, as defined in Definition C.1, with the attention scores $\{a_{t,j}^h\}_{t,j \leq T_{\text{aux}}, h \leq H_{\text{aux}}}$ defined in Equation (7).

### C.2   PROOFS OF THEOREMS AND GRADIENT DEFINITIONS

We restate the theorems and definitions, before presenting their proofs for easy referencing.

**Definition C.2.** [Auxiliary self-attention backpropagation] For query, key, and value weights $\boldsymbol{W}_Q, \boldsymbol{W}_K, \boldsymbol{W}_V \in \mathbb{R}^{D_{\text{aux}} \times D_{\text{aux}}}$ and bias $\boldsymbol{b}_Q, \boldsymbol{b}_K, \boldsymbol{b}_V \in \mathbb{R}^{D_{\text{aux}}}$, the backpropagation layer corresponding to a self-attention layer with $H_{\text{aux}}$ attention heads takes a sequence $\{\partial_{\boldsymbol{y}_t} \in \mathbb{R}^{D_{\text{aux}}}\}_{t \leq T_{\text{aux}}}$ and $\{\boldsymbol{x}_t \in \mathbb{R}^{D_{\text{aux}}}\}_{t \leq T_{\text{aux}}}$ as input and outputs $\{\partial_{\boldsymbol{x}_t}\}_{t \leq T_{\text{aux}}}$, with

$$\partial_{\boldsymbol{x}_t} = \boldsymbol{W}_Q^\top \partial_{\boldsymbol{q}_t} + \boldsymbol{W}_K^\top \partial_{\boldsymbol{k}_t} + \boldsymbol{W}_V^\top \partial_{\boldsymbol{v}_t}, \quad \text{with}$$

$$\partial_{\boldsymbol{q}_t} = \text{VECTORIZE}(\{\sum_j a_{t,j}^h((\partial_{\boldsymbol{y}_t^h})^\top \boldsymbol{v}_j^h)[\boldsymbol{k}_j^h - \sum_{j'} a_{t,j'}^h \boldsymbol{k}_{j'}^h]\}_{h \leq H_{\text{aux}}});$$

$$\partial_{\boldsymbol{k}_t} = \text{VECTORIZE}(\{\sum_j a_{j,t}^h \boldsymbol{q}_j^h[(\partial_{\boldsymbol{y}_j^h})^\top(\boldsymbol{v}_t^h - \sum_{j'} a_{j,j'}^h \boldsymbol{v}_{j'}^h)]\}_{h \leq H_{\text{aux}}});$$

$$\partial_{\boldsymbol{v}_t} = \text{VECTORIZE}(\{\sum_j a_{j,t}^h \partial_{\boldsymbol{y}_j^h}\}_{h \leq H_{\text{aux}}})$$

Here, $\boldsymbol{q}_t$, $\boldsymbol{k}_t$, and $\boldsymbol{v}_t$ refer to query, key, and value vectors at each position $t$, with the attention scores $\{a_{t,j}^h\}_{t,j \leq T_{\text{aux}}, h \leq H_{\text{aux}}}$.

*Derivation of gradient in Definition C.2.* Recalling the definition of $\boldsymbol{y}_t$ from Definition C.1,

$$\boldsymbol{y}_t = \text{VECTORIZE}(\{\sum_{j \leq T_{\text{aux}}} a_{t,j}^h \boldsymbol{v}_j^h\}_{h \leq H_{\text{aux}}}); \quad a_{t,j}^h = \text{softmax}(\boldsymbol{K}^h \boldsymbol{q}_t^h)_j,$$

$$\boldsymbol{q}_t = \boldsymbol{W}_Q \boldsymbol{x}_t + \boldsymbol{b}_Q \quad \boldsymbol{k}_t = \boldsymbol{W}_K \boldsymbol{x}_t + \boldsymbol{b}_K, \quad \boldsymbol{v}_t = \boldsymbol{W}_V \boldsymbol{x}_t + \boldsymbol{b}_V.$$

$\boldsymbol{q}_t^h, \boldsymbol{k}_t^h, \boldsymbol{v}_t^h$ denote $\text{SPLIT}_{H_{\text{aux}}}(\boldsymbol{q}_t)_h$, $\text{SPLIT}_{H_{\text{aux}}}(\boldsymbol{k}_t)_h$, and $\text{SPLIT}_{H_{\text{aux}}}(\boldsymbol{v}_t)_h$ respectively for all $t \leq T_{\text{aux}}$, and $h \leq H_{\text{aux}}$. $\boldsymbol{K}^h \in \mathbb{R}^{T_{\text{aux}} \times D_{\text{aux}}}$ is defined with its rows as $\{\boldsymbol{k}_t^h\}_{t \leq T_{\text{aux}}}$ for all $h \leq H_{\text{aux}}$.

We explain the proof for an arbitrary token position $t$. With the application of the chain rule, we have

$$\partial_{\boldsymbol{x}_t} = (\frac{\partial \boldsymbol{q}_t}{\partial \boldsymbol{x}_t})^\top \partial_{\boldsymbol{q}_t} + (\frac{\partial \boldsymbol{k}_t}{\partial \boldsymbol{x}_t})^\top \partial_{\boldsymbol{k}_t} + (\frac{\partial \boldsymbol{v}_t}{\partial \boldsymbol{x}_t})^\top \partial_{\boldsymbol{v}_t}$$

$$= \boldsymbol{W}_Q^\top \partial_{\boldsymbol{q}_t} + \boldsymbol{W}_K^\top \partial_{\boldsymbol{k}_t} + \boldsymbol{W}_V^\top \partial_{\boldsymbol{v}_t},$$

where the second step follows from the definitions of $\boldsymbol{q}_t$, $\boldsymbol{k}_t$, and $\boldsymbol{v}_t$ respectively.

**Computation of $\partial_{\boldsymbol{q}_t}$:**    With the SPLIT operation of $\boldsymbol{q}_t$ across $H_{\text{aux}}$ heads for the computation of $\boldsymbol{y}_t$, the computation of the backpropagated gradient $\partial_{\boldsymbol{q}_t}$ itself needs to be split across $H_{\text{aux}}$ heads. Furthermore, query vector $\boldsymbol{q}_t$ only affects $\boldsymbol{y}_t$, implying $\frac{\partial \boldsymbol{y}_{t'}}{\partial \boldsymbol{q}_t} = 0$ for any $t' \neq t$. Thus, we have for any head $h \leq H_{\text{aux}}$, if $\boldsymbol{y}_t^h$ represents the output of attention head $h$, given by $\sum_{j \leq T_{\text{aux}}} a_{t,j}^h \boldsymbol{v}_j^h$,

$$
\partial_{\boldsymbol{q}_t^h} = (\frac{\partial \boldsymbol{y}_t^h}{\partial \boldsymbol{q}_t^h})^\top \partial_{\boldsymbol{y}_t^h}
$$

$$
= \sum_{j \leq T_{\text{aux}}} \langle \boldsymbol{v}_j^h, \partial_{\boldsymbol{y}_t^h} \rangle \frac{\partial a_{t,j}^h}{\partial \boldsymbol{q}_t^h}
$$

$$
= \sum_{j \leq T_{\text{aux}}} \langle \boldsymbol{v}_j^h, \partial_{\boldsymbol{y}_t^h} \rangle \frac{\partial}{\partial \boldsymbol{q}_t^h} \left( \frac{e^{\langle \boldsymbol{k}_j^h, \boldsymbol{q}_t^h \rangle}}{\sum_{t' \leq T_{\text{aux}}} e^{\langle \boldsymbol{k}_{t'}^h, \boldsymbol{q}_t^h \rangle}} \right) \tag{8}
$$

$$
= \sum_{j \leq T_{\text{aux}}} \langle \boldsymbol{v}_j^h, \partial_{\boldsymbol{y}_t^h} \rangle \left[ \frac{1}{\sum_{t' \leq T_{\text{aux}}} e^{\langle \boldsymbol{k}_{t'}^h, \boldsymbol{q}_t^h \rangle}} \frac{\partial e^{\langle \boldsymbol{k}_j^h, \boldsymbol{q}_t^h \rangle}}{\partial \boldsymbol{q}_t^h} - \left( \frac{e^{\langle \boldsymbol{k}_j^h, \boldsymbol{q}_t^h \rangle}}{(\sum_{t' \leq T_{\text{aux}}} e^{\langle \boldsymbol{k}_{t'}^h, \boldsymbol{q}_t^h \rangle})^2} \right) \sum_{j' \leq T_{\text{aux}}} \frac{\partial e^{\langle \boldsymbol{k}_{j'}^h, \boldsymbol{q}_t^h \rangle}}{\partial \boldsymbol{q}_t^h} \right] \tag{9}
$$

$$
= \sum_{j \leq T_{\text{aux}}} \langle \boldsymbol{v}_j^h, \partial_{\boldsymbol{y}_t^h} \rangle \left[ \left( \frac{e^{\langle \boldsymbol{k}_j^h, \boldsymbol{q}_t^h \rangle}}{\sum_{t' \leq T_{\text{aux}}} e^{\langle \boldsymbol{k}_{t'}^h, \boldsymbol{q}_t^h \rangle}} \right) \boldsymbol{k}_j^h - \left( \frac{e^{\langle \boldsymbol{k}_j^h, \boldsymbol{q}_t^h \rangle}}{\sum_{t' \leq T_{\text{aux}}} e^{\langle \boldsymbol{k}_{t'}^h, \boldsymbol{q}_t^h \rangle}} \right) \sum_{j' \leq T_{\text{aux}}} \left( \frac{e^{\langle \boldsymbol{k}_{j'}^h, \boldsymbol{q}_t^h \rangle}}{\sum_{t' \leq T_{\text{aux}}} e^{\langle \boldsymbol{k}_{t'}^h, \boldsymbol{q}_t^h \rangle}} \right) \boldsymbol{k}_{j'}^h \right] \tag{10}
$$

$$
= \sum_{j \leq T_{\text{aux}}} a_{t,j}^h \langle \boldsymbol{v}_j^h, \partial_{\boldsymbol{y}_t^h} \rangle \left( \boldsymbol{k}_j^h - \sum_{j' \leq T_{\text{aux}}} a_{t,j'}^h \boldsymbol{k}_{j'}^h \right).
$$

In Equation (8), we have expanded the definition of $\text{softmax}$ in $a_{t,j}^h := \text{softmax}(\boldsymbol{K}^h \boldsymbol{q}_t^h)_j$ in order to better motivate the derivative of $a_{t,j}^h$ w.r.t. $\boldsymbol{q}_t^h$. Finally, $\partial_{\boldsymbol{q}_t}$ is given by $\text{VECTORIZE}(\{\partial_{\boldsymbol{q}_t^h}\}_{h \leq H_{\text{aux}}})$.

**Computation of $\partial_{\boldsymbol{k}_t}$:**    Continuing as the computation of $\partial_{\boldsymbol{q}_t}$, we split the computation of $\partial_{\boldsymbol{k}_t}$ across the $H_{\text{aux}}$ attention heads. However, unlike $\boldsymbol{q}_t$, $\boldsymbol{k}_t$ affects $\boldsymbol{y}_j$ for all $j \leq T_{\text{aux}}$. For any head $h \leq H_{\text{aux}}$, we follow the chain-rule step by step to get

$$
\partial_{\boldsymbol{k}_t^h} = \sum_{j \leq T_{\text{aux}}} (\frac{\partial \boldsymbol{y}_j^h}{\partial \boldsymbol{k}_t^h})^\top \partial_{\boldsymbol{y}_j^h} = \sum_{j \leq T_{\text{aux}}} \left( \frac{\partial \sum_{j' \leq T_{\text{aux}}} a_{j,j'} \boldsymbol{v}_{j'}^h}{\partial \boldsymbol{k}_t^h} \right)^\top \partial_{\boldsymbol{y}_j^h}
$$

$$
= \sum_{j \leq T_{\text{aux}}} \langle \boldsymbol{v}_t^h, \partial_{\boldsymbol{y}_j^h} \rangle \frac{\partial a_{j,t}^h}{\partial \boldsymbol{k}_t^h} + \sum_{j \leq T_{\text{aux}}} \sum_{j' \leq T_{\text{aux}}; j' \neq t} \langle \boldsymbol{v}_{j'}^h, \partial_{\boldsymbol{y}_j^h} \rangle \frac{\partial a_{j,j'}^h}{\partial \boldsymbol{k}_t^h} \tag{11}
$$

$$
= \sum_{j \leq T_{\text{aux}}} \langle \boldsymbol{v}_t^h, \partial_{\boldsymbol{y}_j^h} \rangle \frac{\partial}{\partial \boldsymbol{k}_t^h} \left( \frac{e^{\langle \boldsymbol{k}_t^h, \boldsymbol{q}_j^h \rangle}}{\sum_{t' \leq T_{\text{aux}}} e^{\langle \boldsymbol{k}_{t'}^h, \boldsymbol{q}_j^h \rangle}} \right) \tag{12}
$$

$$
+ \sum_{j \leq T_{\text{aux}}} \sum_{j' \leq T_{\text{aux}}; j' \neq t} \langle \boldsymbol{v}_{j'}^h, \partial_{\boldsymbol{y}_j^h} \rangle \frac{\partial}{\partial \boldsymbol{k}_t^h} \left( \frac{e^{\langle \boldsymbol{k}_{j'}^h, \boldsymbol{q}_j^h \rangle}}{\sum_{t' \leq T_{\text{aux}}} e^{\langle \boldsymbol{k}_{t'}^h, \boldsymbol{q}_j^h \rangle}} \right) \tag{13}
$$

$$
= \sum_{j \leq T_{\text{aux}}} \langle \boldsymbol{v}_t^h, \partial_{\boldsymbol{y}_j^h} \rangle \left[ \left( \frac{e^{\langle \boldsymbol{k}_t^h, \boldsymbol{q}_j^h \rangle}}{\sum_{t' \leq T_{\text{aux}}} e^{\langle \boldsymbol{k}_{t'}^h, \boldsymbol{q}_j^h \rangle}} \right) \boldsymbol{q}_j^h - \left( \frac{e^{\langle \boldsymbol{k}_t^h, \boldsymbol{q}_j^h \rangle}}{\sum_{t' \leq T_{\text{aux}}} e^{\langle \boldsymbol{k}_{t'}^h, \boldsymbol{q}_j^h \rangle}} \right)^2 \boldsymbol{q}_j^h \right] \tag{14}
$$

$$
- \sum_{j \leq T_{\text{aux}}} \sum_{j' \leq T_{\text{aux}}; j' \neq t} \langle \boldsymbol{v}_{j'}^h, \partial_{\boldsymbol{y}_j^h} \rangle \left( \frac{e^{\langle \boldsymbol{k}_{j'}^h, \boldsymbol{q}_j^h \rangle}}{\sum_{t' \leq T_{\text{aux}}} e^{\langle \boldsymbol{k}_{t'}^h, \boldsymbol{q}_j^h \rangle}} \right) \left( \frac{e^{\langle \boldsymbol{k}_t^h, \boldsymbol{q}_j^h \rangle}}{\sum_{t' \leq T_{\text{aux}}} e^{\langle \boldsymbol{k}_{t'}^h, \boldsymbol{q}_j^h \rangle}} \right) \boldsymbol{q}_j^h \tag{15}
$$

$$
= \sum_{j \leq T_{\text{aux}}} \langle \boldsymbol{v}_t^h, \partial_{\boldsymbol{y}_j^h} \rangle (a_{j,t}^h - (a_{j,t}^h)^2) \boldsymbol{q}_j^h - \sum_{j \leq T_{\text{aux}}} \sum_{j' \leq T_{\text{aux}}; j' \neq t} \langle \boldsymbol{v}_{j'}^h, \partial_{\boldsymbol{y}_j^h} \rangle a_{j,j'}^h a_{j,t}^h \boldsymbol{q}_j^h
$$

$$
= \sum_{j \leq T_{\text{aux}}} a_{j,t}^h \langle \partial_{\boldsymbol{y}_j^h}, \boldsymbol{v}_t^h - \sum_{j'} a_{j,j'}^h \boldsymbol{v}_{j'}^h \rangle \boldsymbol{q}_j^h
$$

In Equation (11), we separate the inside sum into two components, since the derivative w.r.t. $\boldsymbol{k}_t^h$ differ for the two components, as outlined in the derivation of Equation (14) from Equation (12), and Equation (15) from Equation (13). We have skipped a step going from Equations (12) and (13) to Equations (14) and (15) due to typographical simplicity. The skipped step is extremely similar to Equation (9) in the derivation of $\partial_{\boldsymbol{q}_t^h}$. Finally, $\partial_{\boldsymbol{k}_t}$ is given by $\text{VECTORIZE}(\{\partial_{\boldsymbol{k}_t^h}\}_{h \le H_{\text{aux}}})$.

**Computation of $\partial_{\boldsymbol{v}_t}$:**  Similar to the gradient computation of $\boldsymbol{q}_t$, the computation of $\partial_{\boldsymbol{v}_t}$ needs to be split across the $H_{\text{aux}}$ attention heads. However, like $\boldsymbol{k}_t$, $\boldsymbol{v}_t$ affects $\boldsymbol{y}_j$ for all $j \le T_{\text{aux}}$. For any head $h \le H_{\text{aux}}$, we follow the chain-rule step by step to get

$$\partial_{\boldsymbol{v}_t^h} = \sum_{j \le T_{\text{aux}}} \left(\frac{\partial \boldsymbol{y}_j^h}{\partial \boldsymbol{v}_t^h}\right)^\top \partial_{\boldsymbol{y}_j^h} = \sum_{j \le T_{\text{aux}}} \left(\frac{\partial \sum_{j' \le T_{\text{aux}}} a_{j,j'} \boldsymbol{v}_{j'}^h}{\partial \boldsymbol{v}_t^h}\right)^\top \partial_{\boldsymbol{y}_j^h} = \sum_{j \le T_{\text{aux}}} a_{j,t}^h \partial_{\boldsymbol{y}_j^h}$$

$\square$

**Theorem C.4.** *With the notations in Definitions 2.12, C.1 and C.2, if on a given input sequence $\{\boldsymbol{x}_t\}_{t=1}^{T_{aux}}$, with its query, key, and value vectors $\{\boldsymbol{q}_t, \boldsymbol{k}_t, \boldsymbol{v}_t\}_{t=1}^{T_{aux}}$, all the $H_{aux}$ attention heads are $\varepsilon$-hard for some $\varepsilon > 0$, then for a given sequence of gradients $\{\partial_{\boldsymbol{y}_t}\}_{t=1}^{T_{aux}}$,*

$$\|\partial_{\boldsymbol{q}_t}\|_2, \|\partial_{\boldsymbol{k}_t}\|_2 \le \mathcal{O}(\varepsilon B_x^2 B_w^2 B_y), \quad \text{for all } t \le T_{aux},$$

*where* $B_x = \max_{t \le T_{aux}} \|\boldsymbol{x}_t\|_2$, $B_y = \max_{t \le T_{aux}} \|\partial_{\boldsymbol{y}_t}\|_2$, *and* $B_w = \max\{\|\boldsymbol{W}_K\|_2, \|\boldsymbol{W}_Q\|_2, \|\boldsymbol{W}_V\|_2, \|\boldsymbol{b}_V\|_2, \|\boldsymbol{b}_K\|_2, \|\boldsymbol{b}_V\|_2\}$.

*This implies, for each position $t$, $\left\|\widehat{\partial_{\boldsymbol{x}_t}} - \partial_{\boldsymbol{x}_t}\right\|_2 \le \mathcal{O}(\varepsilon B_x^2 B_w^3 B_y)$.*

*Proof of Theorem C.4.* For typographical simplicity, we discuss the proof at an arbitrary position $t$. Recall the definition of an $\varepsilon$-hard attention head from Definition C.3. An attention head is defined to be $\varepsilon$-hard on an input sequence $\{\boldsymbol{x}_t\}_{t=1}^{T_{\text{aux}}}$, if for each position $t$, there exists a position $t_0$ such that the attention score $a_{t,t_0} \ge 1 - \varepsilon$.

For the proof, we simply focus on $\partial_{\boldsymbol{q}_t}$, and the proof for $\partial_{\boldsymbol{k}_t}$ follows like-wise.

**Bounds on $\boldsymbol{q}_t$:**  Recalling the definition of $\partial_{\boldsymbol{q}_t}$ from Definition C.2, we have

$$\partial_{\boldsymbol{q}_t} = \text{VECTORIZE}(\{\sum_j a_{t,j}^h ((\partial_{\boldsymbol{y}_t^h})^\top \boldsymbol{v}_j^h)[\boldsymbol{k}_j^h - \sum_{j'} a_{t,j'}^h \boldsymbol{k}_{j'}^h]\}_{h \le H_{\text{aux}}}).$$

Focusing on a head $h \le H_{\text{aux}}$, define $\partial_{\boldsymbol{q}_t^h} = \sum_j a_{t,j}^h ((\partial_{\boldsymbol{y}_t^h})^\top \boldsymbol{v}_j^h)[\boldsymbol{k}_j^h - \sum_{j'} a_{t,j'}^h \boldsymbol{k}_{j'}^h]$ and $t_0 \le T_{\text{aux}}$ as the token position where the $\boldsymbol{q}_t$ attends the most to, i.e. $a_{t,t_0}^h \ge 1 - \varepsilon$ and $\sum_{j \le T_{\text{aux}}; j \ne t_0} a_{t,j}^h \le \varepsilon$. Then,

$$\left\|\partial_{\boldsymbol{q}_t^h}\right\|_2 = \left\|\sum_j a_{t,j}^h ((\partial_{\boldsymbol{y}_t^h})^\top \boldsymbol{v}_j^h)[\boldsymbol{k}_j^h - \sum_{j'} a_{t,j'}^h \boldsymbol{k}_{j'}^h]\right\|_2$$

$$= \left\|a_{t,t_0}^h ((\partial_{\boldsymbol{y}_t^h})^\top \boldsymbol{v}_{t_0}^h)[\boldsymbol{k}_{t_0}^h - \sum_{j'} a_{t,j'}^h \boldsymbol{k}_{j'}^h] + \sum_{j \ne t_0} a_{t,j}^h ((\partial_{\boldsymbol{y}_t^h})^\top \boldsymbol{v}_j^h)[\boldsymbol{k}_j^h - \sum_{j'} a_{t,j'}^h \boldsymbol{k}_{j'}^h]\right\|_2$$

$$\le \underbrace{\left\|a_{t,t_0}^h ((\partial_{\boldsymbol{y}_t^h})^\top \boldsymbol{v}_{t_0}^h)[\boldsymbol{k}_{t_0}^h - \sum_{j'} a_{t,j'}^h \boldsymbol{k}_{j'}^h]\right\|_2}_{\text{Term1}} + \underbrace{\left\|\sum_{j \ne t_0} a_{t,j}^h ((\partial_{\boldsymbol{y}_t^h})^\top \boldsymbol{v}_j^h)[\boldsymbol{k}_j^h - \sum_{j'} a_{t,j'}^h \boldsymbol{k}_{j'}^h]\right\|_2}_{\text{Term2}},$$

where the final step uses a Cauchy-Schwartz inequality. We focus on the two terms separately.

1. Term1: Focusing on $\boldsymbol{k}_{t_0}^h - \sum_{j'} a_{t,j'}^h \boldsymbol{k}_{j'}^h$, we have

$$
\left\| \boldsymbol{k}_{t_0}^h - \sum_{j'} a_{t,j'}^h \boldsymbol{k}_{j'}^h \right\|_2 = \left\| (1 - a_{t,t_0})\boldsymbol{k}_{t_0}^h - \sum_{j' \neq t_0} a_{t,j'}^h \boldsymbol{k}_{j'}^h \right\|_2
$$
$$
\leq (1 - a_{t,t_0}) \left\| \boldsymbol{k}_{t_0}^h \right\|_2 + \sum_{j' \neq t_0} a_{t,j'}^h \left\| \boldsymbol{k}_{j'}^h \right\|_2
$$
$$
\leq \left( (1 - a_{t,t_0}) + \sum_{j' \neq t_0} a_{t,j'}^h \right) \max_j \left\| \boldsymbol{k}_j^h \right\|_2
$$
$$
\leq 2\varepsilon \max_j \left\| \boldsymbol{k}_j^h \right\|_2. \tag{16}
$$

We use a Cauchy-Schwartz inequality in the second and third steps and the attention head behavior in the final step.

Hence, Term1 can now be bounded as follows:

$$
\left\| a_{t,t_0}^h ((\partial_{\boldsymbol{y}_t^h})^\top \boldsymbol{v}_{t_0}^h)[\boldsymbol{k}_{t_0}^h - \sum_{j'} a_{t,j'}^h \boldsymbol{k}_{j'}^h] \right\|_2 = a_{t,t_0}^h \left| (\partial_{\boldsymbol{y}_t^h})^\top \boldsymbol{v}_{t_0}^h \right| \left\| \boldsymbol{k}_{t_0}^h - \sum_{j'} a_{t,j'}^h \boldsymbol{k}_{j'}^h \right\|_2,
$$
$$
\leq 2\varepsilon \left\| \partial_{\boldsymbol{y}_t^h} \right\|_2 \left\| \boldsymbol{v}_{t_0}^h \right\|_2 \max_j \left\| \boldsymbol{k}_j^h \right\|_2.
$$

In the final step, in addition to the bound from Equation (16), we use a Cauchy-Schwartz inequality to bound $\left| (\partial_{\boldsymbol{y}_t^h})^\top \boldsymbol{v}_{t_0}^h \right|$ and bound the attention score $a_{t,t_0}^h$ by 1.

2. Term2: Focusing on $\boldsymbol{k}_j^h - \sum_{j'} a_{t,j'}^h \boldsymbol{k}_{j'}^h$ for any $j \leq T_{\text{aux}}$, we have using two Cauchy-Schwartz inequalities:

$$
\left\| \boldsymbol{k}_j^h - \sum_{j'} a_{t,j'}^h \boldsymbol{k}_{j'}^h \right\|_2 \leq \left\| \boldsymbol{k}_j^h \right\|_2 + \left\| \sum_{j'} a_{t,j'}^h \boldsymbol{k}_{j'}^h \right\|_2 \leq (1 + \sum_{j'} a_{t,j'}^h) \max_{j'} \left\| \boldsymbol{k}_{j'}^h \right\|_2 = 2 \max_{j'} \left\| \boldsymbol{k}_{j'}^h \right\|_2.
$$
$$
\tag{17}
$$

Hence,

$$
\left\| \sum_{j \neq t_0} a_{t,j}^h ((\partial_{\boldsymbol{y}_t^h})^\top \boldsymbol{v}_j^h)[\boldsymbol{k}_j^h - \sum_{j'} a_{t,j'}^h \boldsymbol{k}_{j'}^h] \right\|_2 \leq \left( \sum_{j \neq t_0} a_{t,j}^h \right) \max_j \left| (\partial_{\boldsymbol{y}_t^h})^\top \boldsymbol{v}_j^h \right| \left\| \boldsymbol{k}_j^h - \sum_{j'} a_{t,j'}^h \boldsymbol{k}_{j'}^h \right\|_2
$$
$$
\leq 2\varepsilon \left\| \partial_{\boldsymbol{y}_t^h} \right\|_2 \left( \max_j \left\| \boldsymbol{v}_j^h \right\|_2 \right) \left( \max_{j'} \left\| \boldsymbol{k}_{j'}^h \right\|_2 \right).
$$

In the final step, in addition to the bound from Equation (17), we use a Cauchy-Schwartz inequality to bound $\left| (\partial_{\boldsymbol{y}_t^h})^\top \boldsymbol{v}_j^h \right|$ and use the $\varepsilon$-hard behavior of the attention head to bound $\sum_{j \neq t_0} a_{t,j}^h$.

Combining the bounds on both terms, we have

$$
\left\| \partial_{\boldsymbol{q}_t^h} \right\|_2 \leq 2\varepsilon \left\| \partial_{\boldsymbol{y}_t^h} \right\|_2 \left\| \boldsymbol{v}_{t_0}^h \right\|_2 \max_j \left\| \boldsymbol{k}_j^h \right\|_2 + 2\varepsilon \left\| \partial_{\boldsymbol{y}_t^h} \right\|_2 \left( \max_j \left\| \boldsymbol{v}_j^h \right\|_2 \right) \left( \max_{j'} \left\| \boldsymbol{k}_{j'}^h \right\|_2 \right)
$$
$$
\leq 4\varepsilon \left\| \partial_{\boldsymbol{y}_t^h} \right\|_2 \left( \max_j \left\| \boldsymbol{v}_j^h \right\|_2 \right) \left( \max_{j'} \left\| \boldsymbol{k}_{j'}^h \right\|_2 \right).
$$

We bound the remaining terms as follows.

- $\left\| \partial_{\boldsymbol{y}_t^h} \right\|_2 \leq B_y$, under the bounded assumption of the gradients.

- For any $j \leq T_{\text{aux}}$, we have $\left\|\boldsymbol{k}_j^h\right\|_2 \leq \|\boldsymbol{k}_j\|_2$ since $\boldsymbol{k}_j = \text{VECTORIZE}(\{\boldsymbol{k}_j^{h'}\}_{h' \in H_{\text{aux}}})$. Furthermore, from the defintion of the key vector $\boldsymbol{k}_j$, $\|\boldsymbol{k}_j\|_2 = \|\boldsymbol{W}_K \boldsymbol{x}_j + \boldsymbol{b}_K\|_2 \leq \|\boldsymbol{W}_K\|_2 \|\boldsymbol{x}_j\|_2 + \|\boldsymbol{b}_K\|_2$ with a Cauchy-Schwartz inequality. Under the bounded assumptions of $\boldsymbol{W}_K, \boldsymbol{b}_K$ and input $\boldsymbol{x}_j$, we have $\|\boldsymbol{k}_j\|_2 \leq B_w(1 + B_x)$.

- Similar procedure can be followed for bounding $\max_j \left\|\boldsymbol{v}_j^h\right\|_2$.

Thus, we have $\left\|\partial_{\boldsymbol{q}_t^h}\right\|_2 \leq 4\varepsilon \left\|\partial_{\boldsymbol{y}_t^h}\right\|_2 \left(\max_j \left\|\boldsymbol{v}_j^h\right\|_2\right) \left(\max_{j'} \left\|\boldsymbol{k}_{j'}^h\right\|_2\right) \leq 4\varepsilon B_w^2(1 + B_x)^2 B_y$.

**Bounds on $\left\|\widehat{\partial_{\boldsymbol{x}_t}} - \partial_{\boldsymbol{x}_t}\right\|_2$:** From the definitons of $\widehat{\partial_{\boldsymbol{x}_t}}$ and $\partial_{\boldsymbol{x}_t}$ from Definitions 2.12 and C.6, we have

$$\left\|\widehat{\partial_{\boldsymbol{x}_t}} - \partial_{\boldsymbol{x}_t}\right\|_2 = \left\|\boldsymbol{W}_K^\top \partial_{\boldsymbol{k}_t} + \boldsymbol{W}_Q^\top \partial_{\boldsymbol{q}_t}\right\|_2 \leq \|\boldsymbol{W}_K\|_2 \|\partial_{\boldsymbol{k}_t}\|_2 + \|\boldsymbol{W}_Q\|_2 \|\partial_{\boldsymbol{q}_t}\|_2$$
$$\leq 8\varepsilon B_w^3(1 + B_x)^2 B_y = \mathcal{O}(\varepsilon B_w^3 B_x^2 B_y),$$

where we use Cauchy-schwartz inequality in the second step. We use the assumed bounds on $\|\boldsymbol{W}_Q\|_2, \|\boldsymbol{W}_K\|_2$, and the computed bounds on $\|\partial_{\boldsymbol{q}_t}\|_2, \|\partial_{\boldsymbol{k}_t}\|_2$ in the pre-final step. $\qquad\square$

# D  LAYER NORMALIZATION

We repeat the definition of layer normalization from the main paper below.

**Definition 2.6.** [Layer Normalization] Define a normalization function $f : \mathbb{R}^d \to \mathbb{R}^d$ that performs $f(\boldsymbol{x}) = (\boldsymbol{x} - \mu)/\sigma$, where $\mu$ and $\sigma$ are the mean and standard deviation of $\boldsymbol{x}$, respectively. Then, layer normalization with parameters $\gamma, \boldsymbol{b} \in \mathbb{R}^{D_{\text{aux}}}$ takes as input $\boldsymbol{x} \in \mathbb{R}^{D_{\text{aux}}}$ and outputs $\boldsymbol{y} \in \mathbb{R}^{D_{\text{aux}}}$, which is computed as $\boldsymbol{z} = f(\boldsymbol{x}), \boldsymbol{y} = \gamma \odot \boldsymbol{z} + \boldsymbol{b}$.

In the discussions below, we consider a layer normalization layer in the auxiliary model with parameters $\{\gamma, \boldsymbol{b}\}$ that takes in input sequence $\boldsymbol{x}_1, \cdots, \boldsymbol{x}_{T_{\text{aux}}}$ and outputs $\boldsymbol{y}_1, \cdots, \boldsymbol{y}_{T_{\text{aux}}}$, with $\boldsymbol{y}_t = \gamma \odot \boldsymbol{z}_t + \boldsymbol{b}; \boldsymbol{z}_t = f(\boldsymbol{x}_t)$ for each $t \leq T_{\text{aux}}$. Since this involves a token-wise operation, we will present our constructed modules with a general token position $t$ and the prefix tokens $\{\boldsymbol{v}_j\}$. We will use $\boldsymbol{W}_\gamma$ as a diagonal matrix in $\mathbb{R}^{D_{\text{aux}} \times D_{\text{aux}}}$, containing $\gamma$ on its main diagonal.

**TINT Layer normalization Forward module**  The input embedding to this module $\boldsymbol{e}_t$ will contain $\boldsymbol{x}_t$ in its first $D_{\text{aux}}$ coordinates. The layer normalization computation can be divided into two sub-operations: (a) application of $f$, and (b) linear computation using $\gamma, \boldsymbol{b}$. We will present a TINT module for each sub-operation.

We can represent the function $f$ using a layer normalization operation itself, with its weight and bias parameters set as $\mathbf{1}$ and $\mathbf{0}$ respectively. However, since the relevant input exists only in the first $D_{\text{aux}}$ coordinates, the operation on the first $D_{\text{aux}}$ coordinates needs to be independent of the rest of the coordinates. To do so, we instead use Group normalization (Definition D.3) on $\boldsymbol{e}_t$, with groups of size $D_{\text{aux}}$.

Now, the embedding $\boldsymbol{e}_t$ contains $f(\boldsymbol{x}_t)$ in its first $D_{\text{aux}}$ coordinates. The second sub-operation can then be viewed as a Linear Layer computation, i.e. $\boldsymbol{y}_t = \boldsymbol{W}_\gamma \boldsymbol{x}_t + \boldsymbol{b}$. Hence, we simply stack the rows of $\boldsymbol{W}_\gamma$ and $\boldsymbol{b}_\gamma$ onto the prefix tokens $\{\boldsymbol{v}_j\}$ and call the TINT Linear Forward module (Appendix B).

**Auxiliary's gradient backpropagation through layer normalization**  With the definition of layer normalization and the normalization function $f$ in Definition 2.6, the auxiliary's backpropagation operation takes in the loss gradient w.r.t. output ($\partial_{\boldsymbol{y}}$) and computes the loss gradient w.r.t. input ($\partial_{\boldsymbol{x}}$).

**Definition 2.7.** [Exact Gradient for Layer Normalization] Using notations in Definition 2.6, given the gradient of the loss w.r.t the output of the Layer Normalization $\partial_{\boldsymbol{y}}$, backpropagation computes $\partial_{\boldsymbol{x}}$ as

$$\partial_{\boldsymbol{x}} = (\partial_{\boldsymbol{z}} - D_{\text{aux}}^{-1} \sum_{i=1}^{D_{\text{aux}}} \partial_{z_i} - \langle \partial_{\boldsymbol{z}}, \boldsymbol{z} \rangle \boldsymbol{z})/\sigma \qquad \partial_{\boldsymbol{z}} = \gamma \odot \partial_{\boldsymbol{y}}.$$

**Complexity of true backpropagation** The above operation is computation heavy since it involves computing (a) $\partial_{\boldsymbol{z}}$, (b) $f(\partial_{\boldsymbol{z}})$, (c) $\langle \partial_{\boldsymbol{z}}, \boldsymbol{z} \rangle \boldsymbol{z}$, and (d) multiplying by a factor of $\frac{1}{\sigma}$. $\langle \partial_{\boldsymbol{z}}, \boldsymbol{z} \rangle \boldsymbol{z}$ in itself will require two MLP layers, following Lemma A.4. In order to reduce the number of layers, we turn to first-order Taylor expansion for approximating the above operation.

**Definition 2.8.** [$\epsilon$-approximate Layer Normalization Gradient] With notations defined above, this layer takes $\partial_{\boldsymbol{y}}, \boldsymbol{x} \in \mathbb{R}^{D_{\text{aux}}}$ as input and outputs $\widehat{\partial_{\boldsymbol{x}}} = \frac{1}{\epsilon}(f(\boldsymbol{x} + \epsilon \gamma \odot \partial_{\boldsymbol{y}}) - f(\boldsymbol{x}))$.

The following theorem shows that the first-order gradient is a good approximation of the true gradient, and in the limit of $\epsilon$ tending to $0$, the approximation error tends to $0$ as well.

**Theorem D.1.** *For any $\epsilon > 0$, and a layer normalization layer with parameters $\gamma, \boldsymbol{b} \in \mathbb{R}^{D_{\text{aux}}}$, for an input $\boldsymbol{x} \in \mathbb{R}^{D_{\text{aux}}}$ and gradient $\partial_{\boldsymbol{y}} \in \mathbb{R}^{D_{\text{aux}}}$,*

$$\left\| \widehat{\partial_{\boldsymbol{x}}} - \partial_{\boldsymbol{x}} \right\|_2 \leq \mathcal{O}(\epsilon D_{aux}^{3/2} \sigma^{-2} \|\gamma\|_2^2 \|\partial_{\boldsymbol{y}}\|_2^2),$$

*where $\sigma$ denotes the standard deviation of $\boldsymbol{x}$. $\partial_{\boldsymbol{x}}, \widehat{\partial_{\boldsymbol{x}}}$ have been computed from $\boldsymbol{x}, \partial_{\boldsymbol{y}}$ and $\epsilon$ using Definitions 2.7 and 2.8.*

**TINT Layer normalization backpropagation module** The input embeddings $\boldsymbol{e}_t$ contain $\partial_{\boldsymbol{y}_t}$ at each position $t$ in the first $D_{\text{aux}}$ coordinates. Since we further need the input to the auxiliary's layer normalization layer under consideration, we copy $\boldsymbol{x}_t$ from the TINT Layer normalization Forward module at each position $t$ using residual connections. Furthermore, residual connections have been used to copy the contents of the prefix tokens $\{\boldsymbol{v}_j\}$ from the Layer normalization Forward module, which contain $\boldsymbol{W}_\gamma, \boldsymbol{b}$. Recall that for ease of presentation, we use $\boldsymbol{z}_t$ to represent $f(\boldsymbol{x}_t)$.

We set $\epsilon$ as a hyperparameter and return $\widehat{\partial_{\boldsymbol{x}}}$ as the output of this module. The computation of $\widehat{\partial_{\boldsymbol{x}}}$ can be divided into two sub-operations: (a) computation of $\partial_{\boldsymbol{z}_t} := \gamma \odot \partial_{\boldsymbol{y}_t}$, and (b) computation of $\frac{1}{\epsilon}(f(\boldsymbol{x}_t + \epsilon \partial_{\boldsymbol{z}_t}) - f(\boldsymbol{x}_t))$. We represent each sub-operation as a TINT module.

To compute $\partial_{\boldsymbol{z}_t} := \gamma \odot \partial_{\boldsymbol{y}_t} = W_\gamma \partial_{\boldsymbol{y}_t}$, we can observe that the required operation is identical to backpropagating through a linear layer with parameters $\boldsymbol{W}_\gamma$ and $\boldsymbol{b}$. Hence, we simply call the Linear Backpropagation module on the current embeddings. We use residual connections to retain $\boldsymbol{x}_t$ at each location $t$, and the contents of the prefix tokens $\{\boldsymbol{v}_j\}$.

Now, the embedding $\boldsymbol{e}_t$ contains $\partial_{\boldsymbol{z}_t}$ and $\boldsymbol{x}_t$. In order to backpropagate through $f$, we first use a linear layer to compute $\boldsymbol{x}_t + \epsilon \partial_{\boldsymbol{z}_t}$ and retain $\boldsymbol{x}_t$. Following the same procedure as the Forward module, we use a Group normalization layer with weight and bias parameters $\mathbf{1}$ and $\mathbf{0}$ respectively, to compute $f(\boldsymbol{x}_t + \epsilon \partial_{\boldsymbol{z}_t})$ and $f(\boldsymbol{x}_t)$. Finally, we use a linear layer to compute $\frac{1}{\epsilon}(f(\boldsymbol{x}_t + \epsilon \partial_{\boldsymbol{z}_t}) - f(\boldsymbol{x}_t))$.

**Auxiliary's Descent update** And finally, the auxiliary's descent operation updates parameters $\gamma, \boldsymbol{b}$ using a batch of inputs $\{\boldsymbol{x}_t\}_{t \leq T}$ and the loss gradient w.r.t. the corresponding outputs $\{\partial_{\boldsymbol{y}_t}\}_{t \leq T}$.

**Definition D.2** (Auxiliary's layer normalization descent). For parameters $\gamma, \boldsymbol{b} \in \mathbb{R}^{D_{\text{aux}}}$, descent update takes in a batch of inputs $\{\boldsymbol{x}_t \in \mathbb{R}^{D_{\text{aux}}}\}_{t \leq T_{\text{aux}}}$ and gradients $\{\partial_{\boldsymbol{y}_t} \in \mathbb{R}^{D_{\text{aux}}}\}_{t \leq T_{\text{aux}}}$ and updates the parameters as follows:

$$\gamma \leftarrow \gamma - \eta \sum_{t \leq T_{\text{aux}}} \partial_{\boldsymbol{y}_t} \odot \boldsymbol{z}_t; \qquad \boldsymbol{b} \leftarrow \boldsymbol{b} - \eta \sum_{t \leq T_{\text{aux}}} \partial_{\boldsymbol{y}_t},$$

where $\boldsymbol{z}_t$ represents $f(\boldsymbol{x}_t)$.

The update of $\gamma$ involves an elementwise multiplication between $\partial_{\boldsymbol{y}_t}$ and $\boldsymbol{z}_t$, which requires an MLP layer (Lemma A.4). With the prefix tokens containing the rows of $\boldsymbol{W}_\gamma$ and $\boldsymbol{b}$, we instead consider the update of $\boldsymbol{b}$ alone with the descent update.

**TINT Layer normalization descent module** The input embeddings contain $\partial_{\boldsymbol{y}_t}$ in the first $D_{\text{aux}}$ coordinates. The prefix tokens contain $\boldsymbol{W}_\gamma, \boldsymbol{b}$, which have been copied from the Forward module using residual connections. The update of $\boldsymbol{b}$ is identical to the auxiliary's descent update through a linear layer. Hence, we apply a TINT Linear descent module to the current embeddings, updating only the bias $\boldsymbol{b}$ and switching off the update to $\boldsymbol{W}_\gamma$.

### D.1 Additional definitions

We describe TINT group normalization layer below, which we use in different modules to simulate the auxiliary's layer normalization operations.

**Definition D.3** (TINT $D_{\text{aux}}$-Group normalization)**.** Define a normalization function $f : \mathbb{R}^d \to \mathbb{R}^d$ that performs $f(\boldsymbol{x}) = (\boldsymbol{x} - \mu)/\sigma$, where $\mu$ and $\sigma$ are the mean and standard deviation of $\boldsymbol{x}$, respectively. Then, $D_{\text{aux}}$-Group RMSnorm with parameters $\gamma^{\text{TINT}}, \boldsymbol{b}^{\text{TINT}} \in \mathbb{R}^{D_{\text{aux}}}$ takes as input $\boldsymbol{x} \in \mathbb{R}^{D_{\text{sim}}}$ and outputs $\boldsymbol{y} = \text{VECTORIZE}(\{\boldsymbol{y}^h \in \mathbb{R}^{D_{\text{aux}}}\}_{h \leq \lfloor D_{\text{sim}}/D_{\text{aux}} \rfloor})$, with

$$\boldsymbol{y}^h = \gamma^{\text{TINT}} \odot f(\boldsymbol{x}^h) + \boldsymbol{b}^{\text{TINT}},$$

where $\boldsymbol{x}^h = \text{SPLIT}_{\lfloor D_{\text{sim}}/D_{\text{aux}} \rfloor}(\boldsymbol{x})_h$.

### D.2 Proof of theorems and gradient definitions

We restate the theorems and definitions, before presenting their proofs for easy referencing.

**Definition 2.7.** [Exact Gradient for Layer Normalization] Using notations in Definition 2.6, given the gradient of the loss w.r.t the output of the Layer Normalization $\partial_{\boldsymbol{y}}$, backpropagation computes $\partial_{\boldsymbol{x}}$ as

$$\partial_{\boldsymbol{x}} = (\partial_{\boldsymbol{z}} - D_{\text{aux}}^{-1} \sum_{i=1}^{D_{\text{aux}}} \partial_{z_i} - \langle \partial_{\boldsymbol{z}}, \boldsymbol{z} \rangle \boldsymbol{z})/\sigma \qquad \partial_{\boldsymbol{z}} = \gamma \odot \partial_{\boldsymbol{y}}.$$

*Derivation of gradient in Definition 2.7 .* With the normalization function $f$ and parameters $\boldsymbol{x}, \boldsymbol{b} \in \mathbb{R}^{D_{\text{aux}}}$, recall from Definition 2.6 that given an input $\boldsymbol{x} \in \mathbb{R}^{D_{\text{aux}}}$, a layer normalization layer returns $\boldsymbol{y} = \gamma \odot \boldsymbol{z} + \boldsymbol{b}; \boldsymbol{z} = f(\boldsymbol{x})$. Let $\mu$ and $\sigma$ denote the mean and standard deviation of $\boldsymbol{x}$. They can be computed as

$$\mu = \frac{1}{D_{\text{aux}}} \sum_{i=1}^{D_{\text{aux}}} x_i, \quad \sigma = \sqrt{\frac{1}{D_{\text{aux}}} \sum_{i=1}^{D_{\text{aux}}} (x_i - \mu)^2}.$$

With the chain rule, we can compute $\partial_{\boldsymbol{x}}$ from $\partial_{\boldsymbol{y}}$ as follows.

$$\partial_{\boldsymbol{x}} = \left(\frac{\partial \boldsymbol{z}}{\partial \boldsymbol{x}}\right)^{\top} \partial_{\boldsymbol{z}}; \quad \text{with } \partial_{\boldsymbol{z}} = \left(\frac{\partial \boldsymbol{y}}{\partial \boldsymbol{z}}\right)^{\top} \partial_{\boldsymbol{y}}. \tag{18}$$

Since $\boldsymbol{y} = \gamma \odot \boldsymbol{z} + \boldsymbol{b}$, we have $\frac{\partial \boldsymbol{y}}{\partial \boldsymbol{z}} = \boldsymbol{W}_{\gamma}$, where $\boldsymbol{W}_{\gamma}$ represents a diagonal matrix with $\gamma$ on the main diagonal. Thus, $\partial_{\boldsymbol{z}} = \boldsymbol{W}_{\gamma} \partial_{\boldsymbol{y}} = \gamma \odot \partial_{\boldsymbol{y}}$.

With $\boldsymbol{z} = f(\boldsymbol{x}) = \frac{\boldsymbol{x} - \mu}{\sigma}$, we have

$$\begin{aligned}
\frac{\partial \boldsymbol{z}}{\partial \boldsymbol{x}} &= \frac{\partial}{\partial \boldsymbol{x}} \left(\frac{\boldsymbol{x} - \mu}{\sigma}\right) = \frac{1}{\sigma} \frac{\partial \boldsymbol{x}}{\partial \boldsymbol{x}} - \frac{1}{\sigma} \frac{\partial \mu}{\partial \boldsymbol{x}} - \frac{(\boldsymbol{x} - \mu)}{\sigma^2} \left(\frac{\partial \sigma}{\partial \boldsymbol{x}}\right)^{\top} \\
&= \frac{1}{\sigma} \left(\boldsymbol{I} - \frac{1}{D_{\text{aux}}} \boldsymbol{1}\boldsymbol{1}^{\top} - \boldsymbol{z}\boldsymbol{z}^{\top}\right).
\end{aligned} \tag{19}$$

In the final step, we require $\frac{\partial \mu}{\partial \boldsymbol{x}}$ and $\frac{\partial \sigma}{\partial \boldsymbol{x}}$, which are computed as follows.

- $\frac{\partial \mu}{\partial \boldsymbol{x}} \in \mathbb{R}^{D_{\text{aux}}}$ with its $j$th element given by

$$\left(\frac{\partial \mu}{\partial \boldsymbol{x}}\right)_j = \frac{\partial \mu}{\partial x_j} = \frac{\partial}{\partial x_j} \left(\frac{1}{D_{\text{aux}}} \sum_{i=1}^{D_{\text{aux}}} x_i\right) = \frac{1}{D_{\text{aux}}}.$$

- $\frac{\partial \sigma}{\partial \boldsymbol{x}} \in \mathbb{R}^{D_{\text{aux}}}$ with its $j$th element given by

$$
\begin{aligned}
\left(\frac{\partial \sigma}{\partial \boldsymbol{x}}\right)_j = \frac{\partial \sigma}{\partial x_j} &= \frac{\partial}{\partial x_j}\left(\sqrt{\frac{1}{D_{\text{aux}}}\sum_{i=1}^{D_{\text{aux}}}(x_i - \mu)^2}\right) \\
&= \frac{1}{\sqrt{\sum_{i=1}^{D_{\text{aux}}}(x_i - \mu)^2}}\sum_{i=1}^{D_{\text{aux}}}(x_i - \mu)\frac{\partial(x_i - \mu)}{\partial x_j} \\
&= \frac{1}{\sqrt{\sum_{i=1}^{D_{\text{aux}}}(x_i - \mu)^2}}\left((x_j - \mu) - \frac{1}{D_{\text{aux}}}\sum_{i=1}^{D_{\text{aux}}}(x_i - \mu)\right) = \frac{x_j - \mu}{\sigma} := z_j,
\end{aligned}
$$

where we have re-utilized the $\frac{\partial \mu}{\partial \boldsymbol{x}}$ in the pre-final step.

Hence, from Equation (18),

$$
\partial_{\boldsymbol{x}} = (\frac{\partial \boldsymbol{z}}{\partial \boldsymbol{x}})^\top \partial_{\boldsymbol{z}} = \frac{1}{\sigma}\left(\boldsymbol{I} - \frac{1}{D_{\text{aux}}}\boldsymbol{1}\boldsymbol{1}^\top - \boldsymbol{z}\boldsymbol{z}^\top\right)\partial_{\boldsymbol{z}} = \frac{1}{\sigma}\left(\partial_{\boldsymbol{z}} - \frac{1}{D_{\text{aux}}}\langle\boldsymbol{1}, \partial_{\boldsymbol{z}}\rangle\boldsymbol{1} - \langle\boldsymbol{z}, \partial_{\boldsymbol{z}}\rangle\boldsymbol{z}\right).
$$

$\square$

We repeat Theorem D.1 for easier reference.

**Theorem D.1.** *For any $\epsilon > 0$, and a layer normalization layer with parameters $\gamma, \boldsymbol{b} \in \mathbb{R}^{D_{aux}}$, for an input $\boldsymbol{x} \in \mathbb{R}^{D_{aux}}$ and gradient $\partial_{\boldsymbol{y}} \in \mathbb{R}^{D_{aux}}$,*

$$
\left\|\widehat{\partial_{\boldsymbol{x}}} - \partial_{\boldsymbol{x}}\right\|_2 \leq \mathcal{O}(\epsilon D_{aux}^{3/2}\sigma^{-2}\|\gamma\|_2^2\|\partial_{\boldsymbol{y}}\|_2^2),
$$

*where $\sigma$ denotes the standard deviation of $\boldsymbol{x}$. $\partial_{\boldsymbol{x}}, \widehat{\partial_{\boldsymbol{x}}}$ have been computed from $\boldsymbol{x}, \partial_{\boldsymbol{y}}$ and $\epsilon$ using Definitions 2.7 and 2.8.*

*Proof of Theorem D.1 .* With the normalization function $f$ and parameters $\boldsymbol{x}, \boldsymbol{b} \in \mathbb{R}^{D_{\text{aux}}}$, recall from Definition 2.6 that given an input $\boldsymbol{x} \in \mathbb{R}^{D_{\text{aux}}}$, a layer normalization layer returns $\boldsymbol{y} = \gamma \odot \boldsymbol{z} + \boldsymbol{b}; \boldsymbol{z} = f(\boldsymbol{x})$. Let $\mu$ and $\sigma$ denote the mean and standard deviation of $\boldsymbol{x}$. They can be computed as

$$
\mu = \frac{1}{D_{\text{aux}}}\sum_{i=1}^{D_{\text{aux}}}x_i, \quad \sigma = \sqrt{\frac{1}{D_{\text{aux}}}\sum_{i=1}^{D_{\text{aux}}}(x_i - \mu)^2}.
$$

We will refer to $\frac{\partial \boldsymbol{z}}{\partial \boldsymbol{x}}$ from Equation (19) and the formulation of $\partial_{\boldsymbol{x}}$ from Equation (18) for our current proof. To recall, they are

$$
\frac{\partial \boldsymbol{z}}{\partial \boldsymbol{x}} = \frac{1}{\sigma}\left(\boldsymbol{I} - \frac{1}{D_{\text{aux}}}\boldsymbol{1}\boldsymbol{1}^\top - \boldsymbol{z}\boldsymbol{z}^\top\right), \qquad \partial_{\boldsymbol{x}} = (\frac{\partial \boldsymbol{z}}{\partial \boldsymbol{x}})^\top\partial_{\boldsymbol{z}}.
$$

Using a second-order Taylor expansion of the normalization function $f$ around $\boldsymbol{x}$, we have

$$
\begin{aligned}
f(\boldsymbol{x} + \epsilon\partial_{\boldsymbol{z}}) &= f(\boldsymbol{x}) + \epsilon\frac{\partial f(\boldsymbol{x})}{\partial \boldsymbol{x}}\partial_{\boldsymbol{z}} + \int_0^\epsilon \partial_{\boldsymbol{z}}^\top\frac{\partial}{\partial \boldsymbol{x}_\theta}\left(\frac{\partial f(\boldsymbol{x}_\theta)}{\partial \boldsymbol{x}_\theta}\right)\partial_{\boldsymbol{z}}\theta d\theta \\
&= f(\boldsymbol{x}) + \epsilon\frac{\partial f(\boldsymbol{x})}{\partial \boldsymbol{x}}\partial_{\boldsymbol{z}} - \int_0^\epsilon \frac{1}{\sigma_\theta^2}\left(\|\partial_{\boldsymbol{z}}\|_2^2 - \frac{1}{D_{\text{aux}}}\sum_{i=1}^{D_{\text{aux}}}(\langle\boldsymbol{1}, \partial_{\boldsymbol{z}}\rangle)^2 - (\langle\boldsymbol{z}_\theta, \partial_{\boldsymbol{z}}\rangle)^2\boldsymbol{z}_\theta\right)\theta d\theta,
\end{aligned}
$$

where $\boldsymbol{x}_\theta$ represents $\boldsymbol{x} + \theta\partial_{\boldsymbol{z}}, \boldsymbol{z}_\theta = f(\boldsymbol{x}_\theta)$. The second step follows similar steps for computing $\frac{\partial \boldsymbol{z}}{\partial \boldsymbol{x}}$ in Equation (19). We avoid this computation since we only need to make sure that the second-order term is bounded. Furthermore, if $\epsilon \leq \mathcal{O}\left(\frac{\sigma}{\sqrt{D_{\text{aux}}}\|\partial_{\boldsymbol{z}}\|_2}\right)$, we can show the $\ell_2$-norm of the second-order term can be bounded by $\mathcal{O}(\epsilon^2 D_{\text{aux}}^{3/2}\sigma^{-2}\|\partial_{\boldsymbol{z}}\|_2^2)$. We avoid this computation as well.

Thus, from the above formulation, we have

$$\lim_{\epsilon \to 0} \frac{f(\boldsymbol{x} + \epsilon \partial_{\boldsymbol{z}}) - f(\boldsymbol{x})}{\epsilon} = \frac{\partial f(\boldsymbol{x})}{\partial \boldsymbol{x}} \partial_{\boldsymbol{z}} = \left( \frac{\partial f(\boldsymbol{x})}{\partial \boldsymbol{x}} \right)^\top \partial_{\boldsymbol{z}} = \partial_{\boldsymbol{x}}.$$

The pre-final step follows from Equation (19), where $\frac{\partial f(\boldsymbol{x})}{\partial \boldsymbol{x}} = \frac{\partial \boldsymbol{z}}{\partial \boldsymbol{x}} = \frac{1}{\sigma} \left( \boldsymbol{I} - \frac{1}{D_{\mathrm{aux}}} \boldsymbol{1}\boldsymbol{1}^\top - \boldsymbol{z}\boldsymbol{z}^\top \right)$ can be shown to be symmetric. The final step follows from the gradient formulation in Equation (18). Including the error term, we have the final bound as

$$\left\| \frac{f(\boldsymbol{x} + \epsilon \partial_{\boldsymbol{z}}) - f(\boldsymbol{x})}{\epsilon} - \partial_{\boldsymbol{x}} \right\|_2 \le \mathcal{O}(\epsilon D_{\mathrm{aux}}^{3/2} \sigma^{-2} \left\| \partial_{\boldsymbol{z}} \right\|_2^2).$$

Using $\partial_{\boldsymbol{z}} = \gamma \odot \partial_{\boldsymbol{y}}$ and a Cauchy-Schwartz inequality gives the final bound. □

# E  ACTIVATION LAYER

**Definition E.1** (Auxiliary activation). For a continuous function $\sigma_{\mathrm{act}} : \mathbb{R} \to \mathbb{R}$, an activation layer takes $\boldsymbol{x} \in \mathbb{R}^{D_{\mathrm{aux}}}$ as input and outputs $\boldsymbol{y} = \sigma_{\mathrm{act}}(\boldsymbol{x})$ with $y_i = \sigma_{\mathrm{act}}(x_i)$ for all $i \le D_{\mathrm{aux}}$.

In the discussions below, we consider an activation layer in the auxiliary model with activation function $\sigma_{\mathrm{act}}$ that takes in input sequence $\boldsymbol{x}_1, \cdots, \boldsymbol{x}_{T_{\mathrm{aux}}}$ and outputs $\boldsymbol{y}_1, \cdots, \boldsymbol{y}_{T_{\mathrm{aux}}}$, with $\boldsymbol{y}_t = \sigma_{\mathrm{act}}(\boldsymbol{x}_t)$ for each $t \le T_{\mathrm{aux}}$. Since this involves a token-wise operation, we will present our constructed modules with a general token position $t$. Since no parameters of the auxiliary model are involved in this operation, the prefix tokens $\{\boldsymbol{v}_j\}$ contain 0 in the following modules.

**TINT Activation Forward module**  The embedding $\boldsymbol{e}_t$ contains $\boldsymbol{x}_t$ in its first $D_{\mathrm{aux}}$ indices. We simply pass the embeddings into activation $\sigma_{\mathrm{act}}$, which returns $\sigma_{\mathrm{act}}(\boldsymbol{x}_t)$ in its first $D_{\mathrm{aux}}$ indices.

**Auxiliary's backpropagation through activation**  With the definition in Definition E.1, the auxiliary's backpropagation takes in the loss gradient w.r.t. output ($\partial_{\boldsymbol{y}}$) and computes the loss gradient w.r.t. input ($\partial_{\boldsymbol{x}}$). We further assume that the derivative of $\sigma_{\mathrm{act}}$ is well-defined everywhere. This assumption includes non-differentiable activation functions with well-defined derivatives like $ReLU$.

**Definition E.2** (Auxiliary activation backpropagation). For a continuous function $\sigma_{\mathrm{act}} : \mathbb{R} \to \mathbb{R}$, with a well-defined derivative $\sigma'_{\mathrm{act}}(x) = \partial \sigma_{\mathrm{act}}(x)/\partial x$ for each $x \in \mathbb{R}$, the backpropagation takes $\partial_{\boldsymbol{y}}, \boldsymbol{x} \in \mathbb{R}^{D_{\mathrm{aux}}}$ as input and outputs

$$\partial_{\boldsymbol{x}} = \sigma'_{\mathrm{act}}(\boldsymbol{x}) \odot \partial_{\boldsymbol{y}},$$

where $\sigma'_{\mathrm{act}}(\boldsymbol{x}) \in \mathbb{R}^{D_{\mathrm{aux}}}$ with $\sigma'_{\mathrm{act}}(\boldsymbol{x})_i = \sigma'_{\mathrm{act}}(x_i)$ at each $i \le D_{\mathrm{aux}}$.

**Complexity of true backpropagation**  The above operation is computation heavy since it involves $\sigma'_{\mathrm{act}}(\boldsymbol{x}) \odot \partial_{\boldsymbol{y}}$. As mentioned for the layer normalization module, the element-wise multiplication between $\sigma'_{\mathrm{act}}(\boldsymbol{x})$ and $\partial_{\boldsymbol{y}}$ will require an MLP module following Lemma A.4. Furthermore, it involves changing the activation function in TINT in specific modules to $\sigma'_{\mathrm{act}}$. To circumvent this, we instead turn to a first-order Taylor approximation.

**Definition E.3** (Approximate Activation backpropagation). For a continuous function $\sigma_{\mathrm{act}} : \mathbb{R} \to \mathbb{R}$ and a hyperparameter $\epsilon$, the layer takes $\partial_{\boldsymbol{y}}, \boldsymbol{x} \in \mathbb{R}^{D_{\mathrm{aux}}}$ as input and outputs

$$\widehat{\partial_{\boldsymbol{x}}} = \frac{1}{\epsilon} \left( \sigma_{\mathrm{act}}(\boldsymbol{x} + \epsilon \partial_{\boldsymbol{y}}) - \sigma_{\mathrm{act}}(\boldsymbol{x}) \right).$$

The following theorems show that under mild assumptions on the activation function and the input, gradient pair, the first-order gradient is a good approximation to the true gradient.

**Theorem E.4.** *For any $\epsilon > 0$, $B_y, B_{act} > 0$, consider a second-order differentiable activation function $\sigma_{act} : \mathbb{R} \to \mathbb{R}$, with $\partial^2 \sigma_{act}(x)/\partial(x^2)$ bounded by $B_{act}$ for each $x \in \mathbb{R}$. Then, for any input $\boldsymbol{x} \in \mathbb{R}^{D_{aux}}$ and gradient $\partial_{\boldsymbol{y}} \in \mathbb{R}^{D_{aux}}$ with $\left\| \partial_{\boldsymbol{y}} \right\|_2 \le B_y$, the following holds true:*

$$\left\| \partial_{\boldsymbol{x}} - \widehat{\partial_{\boldsymbol{x}}} \right\|_2 \le \mathcal{O}(B_{act} B_y^2 \epsilon),$$

*where $\partial_{\boldsymbol{x}}, \widehat{\partial_{\boldsymbol{x}}}$ have been defined using $\boldsymbol{x}, \partial_{\boldsymbol{y}}$, and $\epsilon$ in Definitions E.2 and E.3.*

For ReLU activation, which is not second-order differentiable at 0, we instead bound the difference between $\partial_{\boldsymbol{x}}, \widehat{\partial_{\boldsymbol{x}}}$ by defining some form of alignment between input and gradient pair $\boldsymbol{x}, \partial_{\boldsymbol{y}}$.

**Definition E.5** (($\epsilon, \rho$)-alignment). *Input and gradient $\boldsymbol{x}, \partial_{\boldsymbol{y}} \in \mathbb{R}^{D_{\text{aux}}}$ are said to be ($\epsilon, \rho$)-aligned, if there exist a set $C \subseteq [D_{\text{aux}}]$, with $|C| \geq (1 - \rho)D_{\text{aux}}$, such that for each $i$ in $C$, $|x_i| > \epsilon \, |(\partial_{\boldsymbol{y}})_i|$ .*

$\epsilon$ controls the fraction of coordinates where $|x_i| \leq \epsilon \, |(\partial_{\boldsymbol{y}})_i|$. As $\epsilon \to 0$, $\rho \to 0$ as well for bounded gradients.

**Example E.6.** *For any $B_{min}, B_{max} > 0$, all inputs $\boldsymbol{x}$ that satisfy $\min_i |x_i| > B_{min}$, and gradients $\partial_{\boldsymbol{y}}$ that satisfy $\max_j |(\partial_{\boldsymbol{y}})_j| \leq B_{max}$, are ($B_{min}/B_{max}, 0$)-aligned.*

**Theorem E.7.** *For any $\epsilon, \rho > 0$ and $B_y > 0$, for any input $\boldsymbol{x} \in \mathbb{R}^{D_{\text{aux}}}$ and gradient $\partial_{\boldsymbol{y}} \in \mathbb{R}^{D_{\text{aux}}}$, with $\|\partial_{\boldsymbol{y}}\|_{\infty} \leq B_y$, that are ($\epsilon, \rho$)-aligned by Definition E.5,*

$$\left\| \partial_{\boldsymbol{x}} - \widehat{\partial_{\boldsymbol{x}}} \right\|_2 \leq \mathcal{O}(B_y \sqrt{\rho D_{aux}}).$$

*where $\partial_{\boldsymbol{x}}, \widehat{\partial_{\boldsymbol{x}}}$ have been defined using $\boldsymbol{x}, \partial_{\boldsymbol{y}}, \epsilon$ and $\sigma_{act} = \text{ReLU}$ in Definitions E.2 and E.3.*

**TINT Activation backpropagation module**   The input embeddings contain $\partial_{\boldsymbol{y}_t}$ in the first $D_{\text{aux}}$ embeddings. With the requirement of the activation layer input for gradient, we copy $\boldsymbol{x}_t$ from the Forward module at each position $t$. We set $\epsilon$ as a hyper-parameter and return $\widehat{\partial_{\boldsymbol{x}_t}}$ as the output of this module.

$\widehat{\partial_{\boldsymbol{x}_t}}$ will be computed using a single-layer MLP with activation $\sigma_{\text{act}}$ as follows. The first linear layer of the MLP will be used to compute $\boldsymbol{x}_t + \epsilon \partial_{\boldsymbol{y}_t}$ and $\boldsymbol{x}_t$. After the activation $\sigma_{\text{act}}$, the embedding $\boldsymbol{e}_t$ contains $\sigma_{\text{act}}(\boldsymbol{x}_t + \epsilon \partial_{\boldsymbol{y}_t})$ and $\sigma_{\text{act}}(\boldsymbol{x}_t)$. The final linear layer of the MLP will be used to compute $\frac{1}{\epsilon} \left( \sigma_{\text{act}}(\boldsymbol{x}_t + \epsilon \partial_{\boldsymbol{y}_t}) - \sigma_{\text{act}}(\boldsymbol{x}_t) \right)$.

### E.1   PROOFS OF THEOREMS

We restate the theorems, before presenting their proofs for easy referencing.

**Theorem E.4.** *For any $\epsilon > 0$, $B_y, B_{act} > 0$, consider a second-order differentiable activation function $\sigma_{act} : \mathbb{R} \to \mathbb{R}$, with $\partial^2 \sigma_{act}(x)/\partial(x^2)$ bounded by $B_{act}$ for each $x \in \mathbb{R}$. Then, for any input $\boldsymbol{x} \in \mathbb{R}^{D_{aux}}$ and gradient $\partial_{\boldsymbol{y}} \in \mathbb{R}^{D_{aux}}$ with $\|\partial_{\boldsymbol{y}}\|_2 \leq B_y$, the following holds true:*

$$\left\| \partial_{\boldsymbol{x}} - \widehat{\partial_{\boldsymbol{x}}} \right\|_2 \leq \mathcal{O}(B_{act} B_y^2 \epsilon),$$

*where $\partial_{\boldsymbol{x}}, \widehat{\partial_{\boldsymbol{x}}}$ have been defined using $\boldsymbol{x}, \partial_{\boldsymbol{y}},$ and $\epsilon$ in Definitions E.2 and E.3.*

*Proof.* The proof follows along the lines of Theorem D.1. Recall that given an input $\boldsymbol{x}$, the activation layer outputs $\boldsymbol{y} = \sigma_{\text{act}}(\boldsymbol{x})$, where the function $\sigma_{\text{act}}$ is applied coordinate-wise on $\boldsymbol{x}$. Given input $\boldsymbol{x}$ and the output gradient $\partial_{\boldsymbol{y}}$, the gradient w.r.t. the input is given by $\partial_{\boldsymbol{x}} = \sigma'_{\text{act}}(\boldsymbol{x}) \odot \partial_{\boldsymbol{y}}$, where the $\sigma'_{\text{act}}$ function is also applied coordinate wise to $\boldsymbol{x}$. We defined $\widehat{\partial_{\boldsymbol{x}}}$ as an $\epsilon$-approximate gradient, given by $\frac{1}{\epsilon}(\sigma_{\text{act}}(\boldsymbol{x} + \epsilon \partial_{\boldsymbol{y}}) - \sigma_{\text{act}}(\boldsymbol{x}))$. Since both $\sigma_{\text{act}}$ and $\sigma'_{\text{act}}$ are applied coordinate-wise, we can look at the coordinate-wise difference between $\partial_{\boldsymbol{x}}$ and $\widehat{\partial_{\boldsymbol{x}}}$.

Consider an arbitrary coordinate $i \leq D_{\text{aux}}$. Under the assumption that $\sigma_{\text{act}}$ is second-order differentiable, we have

$$(\widehat{\partial_{\boldsymbol{x}}})_i = \frac{1}{\epsilon} \left( \sigma_{\text{act}}(x_i + \epsilon(\partial_{\boldsymbol{y}})_i) - \sigma_{\text{act}}(x_i) \right)$$

$$= \sigma'_{\text{act}}(x_i)(\partial_{\boldsymbol{y}})_i + \frac{1}{\epsilon} \int_{\theta=0}^{\epsilon} \frac{\partial^2 \sigma_{\text{act}}(x_\theta)}{\partial x_\theta^2} (\partial_{\boldsymbol{y}})_i^2 \theta d\theta$$

$$= \sigma'_{\text{act}}(x_i)(\partial_{\boldsymbol{y}})_i + \mathcal{O}(\epsilon B_{act}(\partial_{\boldsymbol{y}})_i^2),$$

where $x_\theta$ represents $x_i + \theta(\partial_{\boldsymbol{y}})_i$ in the second step. In the final step, we utilize the upper bound assumption on $\frac{\partial^2 \sigma_{\text{act}}(x)}{\partial x^2}$.

Thus, $(\partial_{\boldsymbol{x}})_i - (\widehat{\partial_{\boldsymbol{x}}})_i = \mathcal{O}(\epsilon B_{act}(\partial_{\boldsymbol{y}})_i^2)$, and so

$$\left\|\partial_{\boldsymbol{x}} - \widehat{\partial_{\boldsymbol{x}}}\right\|_2 = \mathcal{O}(\epsilon B_{act} \sum_{i=1}^{D_{aux}} (\partial_{\boldsymbol{y}})_i^2) = \mathcal{O}(\epsilon B_{act} \left\|\partial_{\boldsymbol{y}}\right\|_2^2) \leq \mathcal{O}(\epsilon B_{act} B_y^2).$$

$\square$

**Example E.6.** *For any $B_{min}, B_{max} > 0$, all inputs $\boldsymbol{x}$ that satisfy $\min_i |x_i| > B_{min}$, and gradients $\partial_{\boldsymbol{y}}$ that satisfy $\max_j |(\partial_{\boldsymbol{y}})_j| \leq B_{max}$, are $(B_{min}/B_{max}, 0)$-aligned.*

*Proof.* Recall the definition of $(\epsilon, \rho)$-alignment from Definition E.5. Input and gradient $\boldsymbol{x}, \partial_{\boldsymbol{y}} \in \mathbb{R}^{D_{aux}}$ are said to be $(\epsilon, \rho)$-aligned, if there exist a set $C \subseteq [D_{aux}]$, with $|C| \geq (1 - \rho)D_{aux}$, such that for each $i$ in $C$, $|x_i| > \epsilon |(\partial_{\boldsymbol{y}})_i|$.

Consider an arbitrary coordinate $i \leq D_{aux}$. We have $|x_i| > \epsilon |(\partial_{\boldsymbol{y}})_i|$ for any $\epsilon < |x_i| / |(\partial_{\boldsymbol{y}})_i|$. Under the assumption that $|x_i| > B_{min}$, and $|(\partial_{\boldsymbol{y}})_i| \leq B_{max}$, a bound of $B_{min}/B_{max}$ suffices. $\square$

**Theorem E.7.** *For any $\epsilon, \rho > 0$ and $B_y > 0$, for any input $\boldsymbol{x} \in \mathbb{R}^{D_{aux}}$ and gradient $\partial_{\boldsymbol{y}} \in \mathbb{R}^{D_{aux}}$, with $\left\|\partial_{\boldsymbol{y}}\right\|_\infty \leq B_y$, that are $(\epsilon, \rho)$-aligned by Definition E.5,*

$$\left\|\partial_{\boldsymbol{x}} - \widehat{\partial_{\boldsymbol{x}}}\right\|_2 \leq \mathcal{O}(B_y \sqrt{\rho D_{aux}}).$$

*where $\partial_{\boldsymbol{x}}, \widehat{\partial_{\boldsymbol{x}}}$ have been defined using $\boldsymbol{x}, \partial_{\boldsymbol{y}}, \epsilon$ and $\sigma_{act} = \mathrm{ReLU}$ in Definitions E.2 and E.3.*

*Proof.* Recall that given an input $\boldsymbol{x}$, the activation layer outputs $\boldsymbol{y} = \sigma_{\mathrm{act}}(\boldsymbol{x})$, where the function $\sigma_{\mathrm{act}}$ is applied coordinate-wise on $\boldsymbol{x}$. Given input $\boldsymbol{x}$ and the output gradient $\partial_{\boldsymbol{y}}$, the gradient w.r.t. the input is given by $\partial_{\boldsymbol{x}} = \sigma'_{\mathrm{act}}(\boldsymbol{x}) \odot \partial_{\boldsymbol{y}}$, where the $\sigma'_{\mathrm{act}}$ function is also applied coordinate wise to $\boldsymbol{x}$. We defined $\widehat{\partial_{\boldsymbol{x}}}$ as an $\epsilon$-approximate gradient, given by $\frac{1}{\epsilon}(\sigma_{\mathrm{act}}(\boldsymbol{x} + \epsilon \partial_{\boldsymbol{y}}) - \sigma_{\mathrm{act}}(\boldsymbol{x}))$. Since both $\sigma_{\mathrm{act}}$ and $\sigma'_{\mathrm{act}}$ are applied coordinate-wise, we can look at the coordinate-wise difference between $\partial_{\boldsymbol{x}}$ and $\widehat{\partial_{\boldsymbol{x}}}$. For ReLU activation, $\sigma'_{\mathrm{act}}(x) = \mathrm{sign}(x)$ for all $x \in \mathbb{R} \setminus \{0\}$, with $\sigma'_{\mathrm{act}}(0) = 1$ to avoid ambiguity.

Going by the definition of $(\epsilon, \rho)$-alignment of the input and gradient from Definition E.5, we have a set $C$ with $|C| \geq (1 - \rho)D_{aux}$ such that for each $i \in D_{aux}$, $|x_i| > \epsilon |(\partial_{\boldsymbol{y}})_i|$. For all coordinates $i \in C$, we can then observe that $\mathrm{sign}(x_i + \epsilon(\partial_{\boldsymbol{y}})_i) = \mathrm{sign}(x_i)$, implying

$$\sigma_{\mathrm{act}}(x_i + \epsilon(\partial_{\boldsymbol{y}})_i) - \sigma_{\mathrm{act}}(x_i) = \epsilon(\partial_{\boldsymbol{y}})_i \sigma'_{\mathrm{act}}(x_i) = \epsilon(\partial_{\boldsymbol{x}})_i$$

For coordinates $i \notin C$, we have three possible cases:

- $\mathrm{sign}(x_i) = \mathrm{sign}(x_i + \epsilon(\partial_{\boldsymbol{y}})_i)$: In this case, we can again show $\sigma_{\mathrm{act}}(x_i + \epsilon(\partial_{\boldsymbol{y}})_i) - \sigma_{\mathrm{act}}(x_i) = \epsilon(\partial_{\boldsymbol{y}})_i \sigma'_{\mathrm{act}}(x_i) = \epsilon(\partial_{\boldsymbol{x}})_i$.

- $\mathrm{sign}(x_i) = 0, \mathrm{sign}(x_i + \epsilon(\partial_{\boldsymbol{y}})_i) = 1$: In this case, we have $\sigma'_{\mathrm{act}}(x_i) = 0$, and so $(\partial_{\boldsymbol{x}})_i = 0$. Additionally, $\mathrm{sign}((\partial_{\boldsymbol{y}})_i) = 1$, and so

$$|\sigma_{\mathrm{act}}(x_i + \epsilon(\partial_{\boldsymbol{y}})_i) - \sigma_{\mathrm{act}}(x_i) - \epsilon(\partial_{\boldsymbol{x}})_i| = |x_i + \epsilon(\partial_{\boldsymbol{y}})_i| \leq \epsilon |(\partial_{\boldsymbol{y}})_i|,$$

  where in the final step, we use the fact that $x_i < 0$ and $|x_i| < \epsilon |(\partial_{\boldsymbol{y}})_i|$.

- $\mathrm{sign}(x_i) = 1, \mathrm{sign}(x_i + \epsilon(\partial_{\boldsymbol{y}})_i) = 0$: In this case, we have $\sigma'_{\mathrm{act}}(x_i) = 1$, and so $(\partial_{\boldsymbol{x}})_i = (\partial_{\boldsymbol{y}})_i$. Additionally, $\mathrm{sign}((\partial_{\boldsymbol{y}})_i) = 0$, and so

$$|\sigma_{\mathrm{act}}(x_i + \epsilon(\partial_{\boldsymbol{y}})_i) - \sigma_{\mathrm{act}}(x_i) - \epsilon(\partial_{\boldsymbol{x}})_i| = |-x_i - \epsilon(\partial_{\boldsymbol{y}})_i| \leq |\epsilon(\partial_{\boldsymbol{y}})_i|,$$

  where in the final step, we use the fact that $x_i \geq 0$ and $|x_i| < \epsilon |(\partial_{\boldsymbol{y}})_i|$.

Thus, from the above discussion, we have

$$
\begin{aligned}
\left\| \partial_{\boldsymbol{x}} - \widehat{\partial_{\boldsymbol{x}}} \right\|_2 &= \frac{1}{\epsilon} \left( \sum_{i=1}^{D_{\mathrm{aux}}} (\sigma_{\mathrm{act}}(x_i + \epsilon(\partial_{\boldsymbol{y}})_i) - \sigma_{\mathrm{act}}(x_i) - \epsilon(\partial_{\boldsymbol{x}})_i)^2 \right)^{1/2} \\
&= \frac{1}{\epsilon} \left( \sum_{i \notin C} (\sigma_{\mathrm{act}}(x_i + \epsilon(\partial_{\boldsymbol{y}})_i) - \sigma_{\mathrm{act}}(x_i) - \epsilon(\partial_{\boldsymbol{x}})_i)^2 \right)^{1/2} \\
&\leq \left( \sum_{i \notin C} (\partial_{\boldsymbol{y}})_i^2 \right)^{1/2} \leq \sqrt{\rho D_{\mathrm{aux}}} \sqrt{\max_{i \notin C} (\partial_{\boldsymbol{y}})_i^2} \leq \sqrt{\rho D_{\mathrm{aux}}} B_y.
\end{aligned}
$$

The final step includes a simple Cauchy Schwartz inequality and the desired bound comes from the assumed bound on $\|\partial_{\boldsymbol{y}}\|_2$. □

## F  LANGUAGE MODEL HEAD

Additionally, we provide a description of the gradient computation for the loss function that involves the language model head. This computation entails performing a $\mathrm{softmax}$ operation over the entire vocabulary. If $\mathcal{V}$ denotes the vocabulary set of the auxiliary model, and $\boldsymbol{E} \in \mathbb{R}^{|\mathcal{V}| \times D_{\mathrm{aux}}}$ denotes the embedding matrix of the auxiliary model, we directly utilize the embedding matrix for the auto-regressive loss in the TINT. Additionally, we do not update the embedding matrix of the auxiliary model; instead, we solely backpropagate the gradients through the language model head. Recent work in (Kumar et al., 2022) has shown that keeping the embedding matrix fixed while updating the model can stabilize SGD. We demonstrate that the backpropagated gradients can be expressed as the combination of the language model head and a self-attention layer.

**Definition F.1** (KL-loss gradient through auxiliary's language model head). Given an embedding matrix $\boldsymbol{E} \in \mathbb{R}^{|V| \times D_{\mathrm{aux}}}$, the language model head takes in input $\boldsymbol{x} \in \mathbb{R}^{D_{\mathrm{aux}}}$ and a target distribution $\boldsymbol{q} \in \mathbb{R}^{|V|}$ and returns gradient $\partial_{\boldsymbol{x}} \in \mathbb{R}^{D_{\mathrm{aux}}}$, with $\partial_{\boldsymbol{x}} = \boldsymbol{E}^\top (\mathrm{softmax}(\boldsymbol{E}\boldsymbol{x}) - \boldsymbol{q})$.

In the autoregressive loss on a sequence of tokens, the target output distribution at any position is the next occurring token. If $\{\boldsymbol{x}_t^{un}\}_{t=1}^{T_{\mathrm{aux}}}$ denote the uncontextualized embeddings of a sequence of tokens after encoding them via the embedding matrix, and $\{\boldsymbol{x}_t\}_{t=1}^{T_{\mathrm{aux}}}$ denote their contextualized embeddings after passing through the auxiliary model, then the gradient $\partial_{\boldsymbol{x}_t}$ at any position $t$ can be simplified as $\boldsymbol{E}^\top \mathrm{softmax}(\boldsymbol{E}\boldsymbol{x}_t) - \boldsymbol{x}_{t+1}^{un}$. We illustrate the involved TINT module w.r.t. an arbitrary position $t$.

**TINT autoregressive loss gradient module**  The current embedding $\boldsymbol{e}_t$ contains the contextualized embedding $\boldsymbol{x}_t$ in its first $D_{\mathrm{aux}}$ coordinates. Furthermore, $\boldsymbol{e}_t$ includes the uncontextualized embedding $\boldsymbol{x}_t^{un}$, copied from the input layer using residual connections. The prefix tokens $\boldsymbol{v}_j$ are assigned a value of 0 and do not participate in the subsequent computations.

The loss computation can be decomposed into two sub-operations: (a) computing $\boldsymbol{y}_t := \boldsymbol{E}^\top \mathrm{softmax}(\boldsymbol{E}\boldsymbol{x}_t)$, and (b) calculating $\partial_{\boldsymbol{x}_t} = \boldsymbol{y}_t - \boldsymbol{x}_{t+1}^{un}$.

For the first sub-operation, we use a feed-forward layer with $\mathrm{softmax}$ activation, with hidden and output weights $\boldsymbol{E}$ and $\boldsymbol{E}^\top$ respectively, that takes in the first $D_{\mathrm{aux}}$ of $\boldsymbol{e}_t$ and returns $\boldsymbol{y}_t$ in the first $D_{\mathrm{aux}}$ coordinates. We retain $\boldsymbol{x}_t^{un}$ using a residual connection.

The final sub-operation can be interpreted as a TINT self-attention layer. With $\boldsymbol{e}_t$ containing both $\boldsymbol{y}_t$ and $\boldsymbol{x}_t^{un}$, we use a linear self-attention layer (Definition A.1) with two attention heads. The first attention head assigns an attention score of 1 to pairs $\{(t, t+1)\}_{t \leq T_{\mathrm{aux}}-1}$, while assigning an attention score of 0 to the remaining pairs. At any position $t$, $-\boldsymbol{x}_t^{un}$ is considered the value vector. The second attention head assigns an attention score of 1 to pairs $\{(t, t)\}_{t \leq T_{\mathrm{aux}}}$, while assigning an attention score of 0 to the remaining pairs. At any position $t$, $\boldsymbol{y}_t$ is considered the value vector. The outputs of both attention heads are subsequently combined using a linear layer.

*Remark* F.2. We conducted experiments using mean-squared loss and Quad loss Saunshi et al. (2020), which do not necessitate softmax computations for gradient computation. As an example, in the case of mean-squared loss, if our objective is to minimize $\frac{1}{2} \sum_{t=1}^T \left\| \boldsymbol{x}_t - \boldsymbol{x}_{t+1}^{un} \right\|^2$, the gradient can

be computed as $\partial_{\boldsymbol{x}_t} = \boldsymbol{x}_t - \boldsymbol{x}_{t+1}^{un}$. Similarly, in the case of Quad loss, the gradient is $\partial_{\boldsymbol{x}_t} = \frac{1}{|V|}\sum_i \boldsymbol{e}_i - \boldsymbol{x}_{t+1}^{un}$. However, in all of our language model experiments (Section 3), both gradients resulted in minimal improvement in perplexity compared to the auxiliary model. Therefore, we continue utilizing the standard KL loss for optimization.

*Remark* F.3. For ease of implementation in the codebase, we utilize a dedicated loss module that takes in $\boldsymbol{y}_t, \boldsymbol{x}_{t+1}^{un}$ as input and directly computes $\partial_{\boldsymbol{x}_t} = \boldsymbol{y}_t - \boldsymbol{x}_{t+1}^{un}$.

## G  PARAMETER SHARING

**Feed-forward layer of auxiliary model:**   In a standard auxiliary transformer, like GPT-2, the feed-forward layer is a token-wise operation that takes in an input $\boldsymbol{x} \in \mathbb{R}^{D_{\text{aux}}}$ and returns $\boldsymbol{y} = \boldsymbol{A}\sigma(\boldsymbol{W}\boldsymbol{x})$, with $\boldsymbol{A} \in \mathbb{R}^{D_{\text{aux}} \times 4D_{\text{aux}}}$ and $\boldsymbol{W} \in \mathbb{R}^{4D_{\text{aux}} \times D_{\text{aux}}}$. A naive construction of the TINTto simulate its forward operation will have 2 Linear Forward modules (Section 2.4), separated by an activation. However, this requires $4\times$ more prefix embeddings to represent the parameters, compared to other linear operations in the auxiliary transformer that use $\mathbb{R}^{D_{\text{aux}} \times D_{\text{aux}}}$ weight parameters.

To avoid this, we can instead break down the computation into 4 sub-feed-forward layers, each with its own parameters $\{\{\boldsymbol{W}^i, \boldsymbol{A}^i\}\}_{1 \le i \le 4}$. Here $\{\boldsymbol{W}^i\}_{1 \le i \le 4}$ represent 4-shards of the rows of $\boldsymbol{W}$, and $\{\boldsymbol{A}^i\}_{1 \le i \le 4}$ represent 4-shards of the columns of $\boldsymbol{A}$.

The forward, backward, and descent operations on these 4 sub-feed-forward layers can be effectively parallelized. For example, the forward operation of each layer can be simulated by a single TINTmodule, consisting of two Linear Forward modules and activation, changing only the prefix embeddings to correspond to $\{\{\boldsymbol{W}^i, \boldsymbol{A}^i\}\}_{1 \le i \le 4}$.

## H  ADDITIONAL MODULES

We describe the forward, backward, and decent update operations of additional modules, used in different model families, like LLaMA Touvron et al. (2023) and BLOOM Scao et al. (2022). We discuss the simulation of these modules, using similar TINT modules.

### H.1  ROOT MEAN SQUARE NORMALIZATION (RMSNORM)

The operation of RMSnorm Zhang and Sennrich (2019) is very similar to layer normalization.

**Definition H.1** (RMSnorm). For an arbitrary dimension $d$, define a normalization function $f : \mathbb{R}^d \to \mathbb{R}^d$ that performs $f(\boldsymbol{x}) = \boldsymbol{x}/RMS(\boldsymbol{x})$, where $RMS(\boldsymbol{x}) = (\sum_{i=1}^d x_i^2)^{1/2}$. Then, RMSnorm with parameters $\gamma, \boldsymbol{b} \in \mathbb{R}^{D_{\text{aux}}}$ takes as input $\boldsymbol{x} \in \mathbb{R}^{D_{\text{aux}}}$ and outputs $\boldsymbol{y} \in \mathbb{R}^{D_{\text{aux}}}$, which is computed as $\boldsymbol{z} = f(\boldsymbol{x}), \boldsymbol{y} = \gamma \odot \boldsymbol{z} + \boldsymbol{b}$.

The extreme similarity between RMSnorm and layer normalization (Definition 2.6) helps us create similar TINT modules as described in Appendix D, where instead of Group normalization layers, we use Group RMSnorm layers described below.

**Definition H.2** (TINT $D_{\text{aux}}$-Group RMSnorm). For an arbitrary dimension $d$, define a normalization function $f : \mathbb{R}^d \to \mathbb{R}^d$ that performs $f(\boldsymbol{x}) = \boldsymbol{x}/RMS(\boldsymbol{x})$, where $RMS(\boldsymbol{x}) = (\sum_{i=1}^d x_i^2)^{1/2}$. Then, $D_{\text{aux}}$-Group RMSnorm with parameters $\gamma^{\text{TINT}}, \boldsymbol{b}^{\text{TINT}} \in \mathbb{R}^{D_{\text{aux}}}$ takes as input $\boldsymbol{x} \in \mathbb{R}^{D_{\text{sim}}}$ and outputs $\boldsymbol{y} = \text{VECTORIZE}(\{\boldsymbol{y}^h \in \mathbb{R}^{D_{\text{aux}}}\}_{h \le \lfloor D_{\text{sim}}/D_{\text{aux}} \rfloor})$, with

$$\boldsymbol{y}^h = \gamma^{\text{TINT}} \odot f(\boldsymbol{x}^h) + \boldsymbol{b}^{\text{TINT}},$$

where $\boldsymbol{x}^h = \text{SPLIT}_{\lfloor D_{\text{sim}}/D_{\text{aux}} \rfloor}(\boldsymbol{x})_h$.

### H.2  ATTENTION VARIANTS

In order to incorporate additional attention variants, e.g. Attention with Linear Biases (ALiBi) Press et al. (2021), and rotary position embeddings Su et al. (2021), we can change the definition of softmax attention layer in Definition A.1 likewise.

## H.3 GATED LINEAR UNITS (GLUS)

We describe the operations of GLUs Shazeer (2020) using similar GLU units available to the TINT.

**Definition H.3.** For parameters $\boldsymbol{W}, \boldsymbol{V}, \boldsymbol{W}^o \in \mathbb{R}^{D_{\text{aux}} \times D_{\text{aux}}}$, and biases $\boldsymbol{b}_W, \boldsymbol{b}_V, \boldsymbol{b}_{W^\circ} \in \mathbb{R}^{D_{\text{aux}}}$, a GLU layer with activation $\sigma_{\text{act}} : \mathbb{R} \to \mathbb{R}$, takes input $\boldsymbol{x} \in \mathbb{R}^{D_{\text{aux}}}$ and outputs $\widehat{\boldsymbol{y}} \in \mathbb{R}^{D_{\text{aux}}}$, with

$$\boldsymbol{y} = (\boldsymbol{W}\boldsymbol{x} + \boldsymbol{b}_W) \odot \sigma_{\text{act}}(\boldsymbol{V}\boldsymbol{x} + \boldsymbol{b}_V); \quad \widehat{\boldsymbol{y}} = \boldsymbol{W}^o \boldsymbol{y} + \boldsymbol{b}_{W^\circ}.$$

Typical GLUs have $8/3 \times D_{\text{aux}}$ as a hidden dimension (i.e. the dimension of $\boldsymbol{y}$). We can use similar parameter-sharing techniques discussed for feed-forward layers (Appendix G) with the TINT modules presented here. Furthermore, since $\widehat{y}$ can be expressed as a combination of the gated operation and a linear operation, we focus on the computation of $\boldsymbol{y}$ here.

For the discussion below, we consider a GLU (without the output linear layer) in the auxiliary model, with parameters $\boldsymbol{W}, \boldsymbol{V}, \boldsymbol{b}_W, \boldsymbol{b}_V$, that takes in input sequence $\boldsymbol{x}_1, \cdots, \boldsymbol{x}_T$ and outputs $\boldsymbol{y}_1, \cdots, \boldsymbol{y}_T$, with $\boldsymbol{y}_t = (\boldsymbol{W}\boldsymbol{x}_t + \boldsymbol{b}_W) \odot \sigma_{\text{act}}(\boldsymbol{V}\boldsymbol{x}_t + \boldsymbol{b}_V)$ for each $t \leq T_{\text{sim}}$. Since this involves a token-wise operation, we will present our constructed modules with a general token position $t$ and the prefix tokens $\{\boldsymbol{v}_j\}$.

**TINT GLU Forward module** The embedding $\boldsymbol{e}_t$ contains $\boldsymbol{x}_t$ in its first $D_{\text{aux}}$ coordinates. The output $\boldsymbol{y}_t$ can be computed using three sub-operations: (a) linear operation for $\boldsymbol{W}\boldsymbol{x}_t + \boldsymbol{b}_W$, (b) linear operation for $\boldsymbol{V}\boldsymbol{x}_t + \boldsymbol{b}_V$, and (c) gate operation to get $(\boldsymbol{W}\boldsymbol{x}_t + \boldsymbol{b}_W) \odot \sigma_{\text{act}}(\boldsymbol{V}\boldsymbol{x}_t + \boldsymbol{b}_V)$.

We use three TINT modules, representing each sub-operation.

(a) $\boldsymbol{W}\boldsymbol{x}_t + \boldsymbol{b}_W$ is a linear operation, hence we can use a TINT Linear Forward module (Appendix B) with the current embedding $\boldsymbol{e}_t$ and $\{\boldsymbol{v}_j\}$ containing $\boldsymbol{W}, \boldsymbol{b}_W$ to get embedding $\widetilde{\boldsymbol{e}}_t$ containing $\boldsymbol{W}\boldsymbol{x}_t + \boldsymbol{b}_W$ in its first $D_{\text{aux}}$ coordinates.

(b) $\boldsymbol{V}\boldsymbol{x}_t + \boldsymbol{b}_V$ is a linear operation, hence we can similarly use a TINT Linear Forward module (Appendix B) with the embedding $\boldsymbol{e}_t$ and $\{\boldsymbol{v}_j\}$ containing $\boldsymbol{W}_V, \boldsymbol{b}_V$ to get embedding $\widehat{\boldsymbol{e}}_t$ containing $\boldsymbol{V}\boldsymbol{x}_t + \boldsymbol{b}_V$ in its first $D_{\text{aux}}$ coordinates.

$\widehat{\boldsymbol{e}}_t$ and $\widetilde{\boldsymbol{e}}_t$ are now combined to get an embedding $\boldsymbol{e}_t$ that contains $\boldsymbol{W}\boldsymbol{x}_t + \boldsymbol{b}_W, \boldsymbol{V}\boldsymbol{x}_t + \boldsymbol{b}_V$ in its first $2D_{\text{aux}}$ coordinates.

(c) Finally, we can use a TINT GLU layer that can carry out the elementwise multiplication of $\boldsymbol{W}\boldsymbol{x}_t + \boldsymbol{b}_W, \sigma_{\text{act}}(\boldsymbol{V}\boldsymbol{x}_t + \boldsymbol{b}_V)$ to get $\boldsymbol{y}_t$ in the first $D_{\text{aux}}$ coordinates.

*Parameter Sharing:* Since (a) and (b) involve a Linear Forward module, we can additionally leverage parameter sharing to apply a single Linear Forward module for each of the two computations, changing only the prefix embeddings to correspond to $\boldsymbol{W}, \boldsymbol{b}_W$, or $\boldsymbol{W}_V, \boldsymbol{b}_V$.

**Auxiliary GLU backpropagation** For the GLU layer defined in Definition H.3, the backpropagation layer takes in the loss gradient w.r.t. output ($\partial_{\boldsymbol{y}}$) and computes the loss gradient w.r.t. input ($\partial_{\boldsymbol{x}}$).

**Definition H.4** (Auxiliary GLU backpropagation). For the weights $\boldsymbol{W}, \boldsymbol{V} \in \mathbb{R}^{D_{\text{aux}} \times D_{\text{aux}}}$, the backpropagation layer takes $\partial_{\boldsymbol{y}} \in \mathbb{R}^{D_{\text{aux}}}$ as input and outputs $\partial_{\boldsymbol{x}} \in \mathbb{R}^{D_{\text{aux}}}$, with $\partial_{\boldsymbol{x}} = \boldsymbol{W}^\top \widehat{\partial_{\boldsymbol{x}}} + \boldsymbol{V}^\top \widetilde{\partial_{\boldsymbol{x}}}$, where

$$\widehat{\partial_{\boldsymbol{x}}} = \partial_{\boldsymbol{y}} \odot \sigma_{\text{act}}(\boldsymbol{V}\boldsymbol{x} + \boldsymbol{b}_V); \qquad \widetilde{\partial_{\boldsymbol{x}}} = \sigma'_{\text{act}}(\boldsymbol{V}\boldsymbol{x} + \boldsymbol{b}_V) \odot \partial_{\boldsymbol{y}} \odot (\boldsymbol{W}\boldsymbol{x} + \boldsymbol{b}_W).$$

A direct computation of $\widetilde{\partial_{\boldsymbol{x}}}$ involves changing the activation function to $\sigma'_{\text{act}}$. Following a similar strategy for backpropagation through an activation layer (Appendix E), we instead use a first-order Taylor expansion to approximate $\widetilde{\partial_{\boldsymbol{x}}}$.

**Definition H.5** (Auxiliary GLU approximate backpropagation). For a hyper-parameter $\epsilon > 0$, for the weights $\boldsymbol{W}, \boldsymbol{V} \in \mathbb{R}^{D_{\text{aux}} \times D_{\text{aux}}}$, the approximate backpropagation layer takes $\partial_{\boldsymbol{y}} \in \mathbb{R}^{D_{\text{aux}}}$ as input and outputs $\overline{\partial_{\boldsymbol{x}}} \in \mathbb{R}^{D_{\text{aux}}}$, with $\overline{\partial_{\boldsymbol{x}}} = \boldsymbol{W}^\top \widehat{\partial_{\boldsymbol{x}}} + \boldsymbol{V}^\top \widetilde{\overline{\partial_{\boldsymbol{x}}}}$, where

$$\widehat{\partial_{\boldsymbol{x}}} = \partial_{\boldsymbol{y}} \odot \sigma_{\text{act}}(\boldsymbol{V}\boldsymbol{x} + \boldsymbol{b}_V)$$

$$\widetilde{\overline{\partial_{\boldsymbol{x}}}} = \sigma_{\text{act}}(\boldsymbol{V}\boldsymbol{x} + \boldsymbol{b}_V + \epsilon\partial_{\boldsymbol{y}}) \odot \frac{1}{\epsilon}(\boldsymbol{W}\boldsymbol{x} + \boldsymbol{b}_W) - \sigma_{\text{act}}(\boldsymbol{V}\boldsymbol{x} + \boldsymbol{b}_V) \odot \frac{1}{\epsilon}(\boldsymbol{W}\boldsymbol{x} + \boldsymbol{b}_W).$$

**TINT GLU backpropagation module**  The current embedding contains $\partial_{\boldsymbol{y}_t}$ in its first $D_{\text{aux}}$ coordinates. Furthermore, since we need $\boldsymbol{W}\boldsymbol{x}_t + \boldsymbol{b}_W$ and $\boldsymbol{V}\boldsymbol{x}_t + \boldsymbol{b}_V$ in the gradient computations, we copy them from the Forward module using residual connections. We discuss the computation of $\boldsymbol{W}^\top \widehat{\partial_{\boldsymbol{x}_t}}$ and $\boldsymbol{V}^\top \widetilde{\partial_{\boldsymbol{x}_t}}$ as separate sub-modules acting on the same embedding $\boldsymbol{e}_t$ in parallel.

1. The computation of $\boldsymbol{W}^\top \widehat{\boldsymbol{x}_t}$ involves two sub-operations: (a) gate operation to get $\widehat{\boldsymbol{x}_t} := \partial_{\boldsymbol{y}_t} \odot \sigma_{\text{act}}(\boldsymbol{V}\boldsymbol{x}_t + \boldsymbol{b}_V)$, and (b) linear backward operation to get $\boldsymbol{W}^\top \widehat{\boldsymbol{x}_t}$. Since for this operation, we require $\boldsymbol{W}$, we copy the contents of the prefix embeddings containing $\boldsymbol{W}, \boldsymbol{b}_W$ from the Forward module.

   (a) Since the current embedding $\boldsymbol{e}_t$ contains both $\partial_{\boldsymbol{y}_t}$ and $\boldsymbol{W}\boldsymbol{x}_t + \boldsymbol{b}_W$, we can use a TINT GLU layer to get an embedding $\widehat{\boldsymbol{e}}_t^{(1)}$ that contains $\widehat{\partial_{\boldsymbol{x}_t}}$.

   (b) The final linear backward operation can be performed by using a TINT Linear backpropagation module (Appendix B) with the embeddings $\widehat{\boldsymbol{e}}_t^{(1)}$ and the prefix embeddings. The final embedding $\widehat{\boldsymbol{e}}_t$ contains $\boldsymbol{W}^\top \widehat{\boldsymbol{x}_t}$ in the first $D_{\text{aux}}$ coordinates.

2. The computation of $\boldsymbol{V}^\top \widetilde{\boldsymbol{x}_t}$ involves four sub-operations: (a) gate operation to get $\frac{1}{\epsilon}(\boldsymbol{W}\boldsymbol{x}_t + \boldsymbol{b}_W) \odot \sigma_{\text{act}}(\boldsymbol{V}\boldsymbol{x}_t + \boldsymbol{b}_V + \epsilon\partial_{\boldsymbol{y}_t})$, (b) gate operation to get $\frac{1}{\epsilon}(\boldsymbol{W}\boldsymbol{x}_t + \boldsymbol{b}_W) \odot \sigma_{\text{act}}(\boldsymbol{V}\boldsymbol{x}_t + \boldsymbol{b}_V)$, (c) a linear layer to compute $\widetilde{\boldsymbol{x}_t}$, (c) linear backward operation to get $\boldsymbol{V}^\top \widetilde{\boldsymbol{x}_t}$. Since for this operation, we require $\boldsymbol{V}$, we copy the contents of the prefix embeddings containing $\boldsymbol{V}, \boldsymbol{b}_V$ from the Forward module.

   (a) Since the current embedding $\boldsymbol{e}_t$ contains $\partial_{\boldsymbol{y}_t}$, $\boldsymbol{V}\boldsymbol{x}_t + \boldsymbol{b}_W$ and $\boldsymbol{W}\boldsymbol{x}_t + \boldsymbol{b}_W$, we can use two TINT GLU layers to get an embedding $\widetilde{\boldsymbol{e}}_t^{(1)}$ that contains both $\frac{1}{\epsilon}(\boldsymbol{W}\boldsymbol{x}_t + \boldsymbol{b}_W) \odot \sigma_{\text{act}}(\boldsymbol{V}\boldsymbol{x}_t + \boldsymbol{b}_V + \epsilon\partial_{\boldsymbol{y}_t})$ and $\frac{1}{\epsilon}(\boldsymbol{W}\boldsymbol{x}_t + \boldsymbol{b}_W) \odot \sigma_{\text{act}}(\boldsymbol{V}\boldsymbol{x}_t + \boldsymbol{b}_V)$.

   (b) A linear later on $\widetilde{\boldsymbol{e}}_t^{(1)}$ can then return an embedding $\widetilde{\boldsymbol{e}}_t^{(2)}$ containing $\widetilde{\boldsymbol{x}_t}$ in the first $D_{\text{aux}}$ coordinates.

   (c) The final operation can be performed by using a TINT Linear backpropagation module (Appendix B) with the embeddings $\widehat{\boldsymbol{e}}_t^2$ and the prefix embeddings containing $\boldsymbol{V}, \boldsymbol{b}_V$. The final embedding $\widetilde{\boldsymbol{e}}_t$ contains $\boldsymbol{V}^\top \widetilde{\boldsymbol{x}_t}$ in the first $D_{\text{aux}}$ coordinates.

After the two parallel computations, we can sum up $\widehat{\boldsymbol{e}}_t$ and $\widetilde{\boldsymbol{e}}_t$ to get an embedding $\boldsymbol{e}_t$ containing $\overline{\partial_{\boldsymbol{x}_t}}$ (Definition H.5) in the first $D_{\text{aux}}$ coordinates.

**Auxiliary GLU descent**  Finally, the auxiliary's descent updates the weight and the bias parameters using a batch of inputs $\{\boldsymbol{x}_t\}_{t \leq T}$ and the loss gradient w.r.t. the corresponding outputs $\{\partial_{\boldsymbol{y}_t}\}_{t \leq T}$.

**Definition H.6** (Auxiliary GLU descent ). For weights $\boldsymbol{W}, \boldsymbol{V} \in \mathbb{R}^{D_{\text{aux}} \times D_{\text{aux}}}$ and bias $\boldsymbol{b}_W, \boldsymbol{b}_V \in \mathbb{R}^{D_{\text{aux}}}$, the linear descent layer takes in a batch of inputs $\{\boldsymbol{x}_t \in \mathbb{R}^{D_{\text{aux}}}\}_{t \leq T_{\text{aux}}}$ and gradients $\{\partial_{\boldsymbol{y}_t} \in \mathbb{R}^{D_{\text{aux}}}\}_{t \leq T_{\text{aux}}}$ and updates the parameters as follows:

$$\boldsymbol{W} \leftarrow \boldsymbol{W} - \eta \sum_{t \leq T_{\text{aux}}} \widehat{\partial_{\boldsymbol{x}_t}} \boldsymbol{x}_t^\top; \qquad \boldsymbol{b}_W \leftarrow \boldsymbol{b}_W - \eta \sum_{t \leq T_{\text{aux}}} \widehat{\partial_{\boldsymbol{x}_t}},$$

$$\boldsymbol{V} \leftarrow \boldsymbol{V} - \eta \sum_{t \leq T_{\text{aux}}} \widetilde{\partial_{\boldsymbol{x}_t}} \boldsymbol{x}_t^\top; \qquad \boldsymbol{b}_V \leftarrow \boldsymbol{b}_V - \eta \sum_{t \leq T_{\text{aux}}} \widetilde{\partial_{\boldsymbol{x}_t}},$$

where $\widehat{\partial_{\boldsymbol{x}_t}}$ and $\widetilde{\partial_{\boldsymbol{x}_t}}$ have been computed as Definition H.4.

Due to similar concerns as gradient backpropagation, we instead use $\widetilde{\partial_{\boldsymbol{x}_t}}$ (Definition H.5) in place of $\widetilde{\partial_{\boldsymbol{x}_t}}$ for each $t \leq T_{\text{aux}}$ to update $\boldsymbol{V}, \boldsymbol{b}_V$.

**TINT GLU descent module**  We discuss the two descent operations separately.

1. Update of $\boldsymbol{W}, \boldsymbol{b}_W$: We start with the embeddings $\widehat{\boldsymbol{e}}_t^{(1)}$ from the backpropagation module, that contain $\widehat{\partial_{\boldsymbol{x}_t}}$ in the first $D_{\text{aux}}$ coordinates.

For the update, we additionally require the input to the auxiliary GLU layer under consideration, and hence we copy $x_t$ from the Forward module using residual connections. Furthermore, we copy the contents of the prefix embeddings that contain $W, b_W$ from the Forward module.

With both $\widehat{\partial}_{x_t}$ and $x_t$ in the embeddings, the necessary operation turns out to be the descent update of a linear layer with parameters $W, b_W$. That implies, we can call a TINT Linear descent module (Appendix B) on the current embeddings and prefix embeddings to get the desired update.

2. We start with the embeddings $\widetilde{e}_t^{(2)}$ from the backpropagation module, that contain $\widetilde{\partial}_{x_t}$ in the first $D_{\text{aux}}$ coordinates.

   For the update, we additionally require the input to the auxiliary GLU layer under consideration, and hence we copy $x_t$ from the forward module using residual connections. Furthermore, we copy the contents of the prefix embeddings that contain $V, b_V$ from the Forward module.

   With both $\widetilde{\partial}_{x_t}$ and $x_t$ in the embeddings, the necessary operation turns out to be the descent update of a linear layer with parameters $V, b_V$. That implies we can call a TINT Linear descent module on the current embeddings and prefix embeddings to get the desired update.

*Parameter sharing*: Since both the descent updates involve a Linear descent module, we can additionally leverage parameter sharing to apply a single TINT Linear descent module for each of the two computations, changing the input to correspond to $\{\widehat{e}_t^{(1)}\}$ and prefix to correspond to $W, b_W$, or the input to correspond to $\{\widetilde{e}_t^{(2)}\}$ and prefix to correspond to $V, b_V$ respectively.

## I    CONSTRUCTION OF OTHER VARIANTS OF PRE-TRAINED MODELS

Table 3: Number of parameters of TINT for the forward, backward, and gradient update operations on various modules. For simplicity, we have ignored biases in the following computation. We set $H_{\text{sim}} = 12$ for OPT-125M and $H_{\text{sim}} = 16$ for the other models, $D_{\text{sim}} = 4D_{\text{aux}}$ for all the models, and $T_{\text{sim}} = T_{\text{aux}} + K$, with $T_{\text{aux}} = 2048$ for OPT models, and $K = D_{\text{aux}}/4$. $Q = 4Q_{split} + 3T_{\text{sim}}D_{\text{sim}}/H_{\text{sim}}$, where $Q_{split} = \frac{1}{H_{\text{sim}}}(D_{\text{sim}})^2 + H_{\text{sim}}D_{\text{sim}}$, denotes the number of parameters in a TINT Linear Forward module (Section 2.4).

| Module Name | Module Size | | | |
|---|---|---|---|---|
| | Forward | Backward | Descent | Total |
| Linear layer | $Q$ | $Q$ | $Q$ | $3Q$ |
| Layer norms | $Q$ | $Q + 2D_{\text{sim}}H_{\text{sim}}$ | $Q$ | $3Q + 2D_{\text{sim}}H_{\text{sim}}$ |
| Self-Attention | $2Q$ | $2Q$ | $2Q$ | $6Q$ |
| Activation | $Q_{split}$ | $2D_{\text{sim}}H_{\text{sim}}$ | $0$ | $Q_{split} + 2D_{\text{sim}}H_{\text{sim}}$ |
| Self-Attention block | $4Q$ | $4Q + 2D_{\text{sim}}H_{\text{sim}}$ | $4Q$ | $12Q + 2D_{\text{sim}}H_{\text{sim}}$ |
| Feed-forward block | $3Q$ | $3Q + 4D_{\text{sim}}H_{\text{sim}}$ | $3Q$ | $9Q + 4D_{\text{sim}}H_{\text{sim}}$ |
| Transformer block | $7Q$ | $7Q + 6D_{\text{sim}}H_{\text{sim}}$ | $7Q$ | $21Q + 6D_{\text{sim}}H_{\text{sim}}$ |
| Transformer | $7QL + LQ_{split}$ | $(7Q + 6D_{\text{sim}}H_{\text{sim}})L$ | $7QL$ | $(21Q + 6D_{\text{sim}}H_{\text{sim}})L$ |
| OPT-125M | 0.4B | 0.4B | 0.4B | 1.2B |
| OPT-350M | 1.2B | 1.1B | 1.1B | 3.4B |
| OPT-1.3B | 3.7B | 3.6B | 3.5B | 10.8B |
| OPT-2.7B | 7.4B | 7.2B | 7.2B | 21.8B |

Though we only conduct experiments on an OPT-125M model, our construction is generally applicable to diverse variants of pre-trained language models. Table 3 highlights many types of modules and the required size and computation for each. The size of a constructed model is influenced by various factors, including the number of layers, and embedding dimension in the auxiliary.

## J    EXPERIMENTS

**Computing environment**: All the experiments are conducted on a single A100 80G GPU.

**Hyperparameters:** In the few-shot setting, we employ three different random seeds to select distinct sets of training examples. Grid search is performed for each seed to determine the optimal learning rate for both constructed models and dynamic evaluation. The learning rates considered for the learning rate hyperparameter in the descent update operations in TINT are $1e-3, 1e-4, 1e-5$. [6] Additionally, we explore various layer-step combinations to allocate a fixed budget for one full forward pass. Specifically, we update the top 3 layers for 4 steps, the top 6 layers for 3 steps, or 12 layers for 1 step.

**Results of different settings.**    Table 4 displays the results of few-shot learning with calibration across various settings, encompassing different loss types, input formats, and layer-step configurations. Our analysis reveals that employing a label-only loss, utilizing a single-example input format, and updating all layers of the internal model for a single step yield the most favorable average result. The performance of the multi-example format is disadvantaged when dealing with tasks of long sequences such as Amazon Polarity. In general, we observe that calibrated results tend to be more consistent and stable.

## K    BROADER IMPACTS

Our findings suggest that existing transformer-based language models possess the ability to learn and adapt to context by internally fine-tuning a complex model *even during inference*. Consequently,

---

[6]When utilizing the full-context loss, the learning rates considered are $1e-5, 1e-6$, and $1e-7$ due to gradient summations in TINT.

Table 4: Few-shot ($k = 32$) results with different loss types, input formats, and layer-step configurations with a fixed compute budget, with calibration.

| Loss Type | Format | Layer | Step | Subj | AGNews | SST2 | CR | MR | MPQA | Amazon | Avg. |
|---|---|---|---|---|---|---|---|---|---|---|---|
| **Label** | Single | 12 | 1 | $66.0_{(1.9)}$ | $64.7_{(0.2)}$ | $68.7_{(1.3)}$ | $69.0_{(0.7)}$ | $63.7_{(0.2)}$ | $82.8_{(0.5)}$ | $73.7_{(0.6)}$ | $69.8_{(0.1)}$ |
| | Single | 6 | 2 | $62.7_{(0.2)}$ | $66.3_{(0.2)}$ | $68.3_{(6.1)}$ | $67.2_{(0.2)}$ | $61.8_{(1.6)}$ | $81.0_{(3.6)}$ | $74.3_{(0.5)}$ | $68.8_{(1.4)}$ |
| | Single | 3 | 4 | $63.5_{(0.0)}$ | $67.2_{(0.8)}$ | $62.5_{(0.4)}$ | $68.7_{(1.4)}$ | $61.7_{(0.6)}$ | $76.8_{(3.3)}$ | $75.2_{(0.8)}$ | $67.9_{(0.8)}$ |
| | Multi. | 12 | 1 | $83.2_{(2.5)}$ | $43.7_{(6.6)}$ | $60.7_{(5.7)}$ | $70.3_{(6.1)}$ | $62.8_{(8.9)}$ | $84.2_{(1.6)}$ | $66.3_{(12.3)}$ | $67.3_{(0.9)}$ |
| | Multi. | 6 | 2 | $83.5_{(2.9)}$ | $43.2_{(8.4)}$ | $52.0_{(1.5)}$ | $70.5_{(6.0)}$ | $58.5_{(11.3)}$ | $82.0_{(0.4)}$ | $55.8_{(7.6)}$ | $63.6_{(2.7)}$ |
| | Multi. | 3 | 4 | $84.0_{(2.3)}$ | $42.3_{(8.4)}$ | $51.5_{(1.8)}$ | $68.2_{(4.6)}$ | $58.5_{(12.0)}$ | $80.2_{(2.1)}$ | $58.5_{(7.9)}$ | $63.3_{(3.0)}$ |
| **Full-context** | Single | 12 | 1 | $64.5_{(0.4)}$ | $65.8_{(0.2)}$ | $63.2_{(0.9)}$ | $67.3_{(0.5)}$ | $60.8_{(1.4)}$ | $73.5_{(0.8)}$ | $75.0_{(0.4)}$ | $67.2_{(0.1)}$ |
| | Single | 6 | 2 | $66.7_{(2.0)}$ | $66.0_{(0.4)}$ | $62.7_{(0.6)}$ | $70.5_{(2.1)}$ | $59.7_{(0.9)}$ | $77.7_{(2.2)}$ | $76.0_{(0.0)}$ | $68.5_{(0.4)}$ |
| | Single | 3 | 4 | $64.0_{(0.0)}$ | $65.8_{(0.6)}$ | $65.0_{(1.9)}$ | $67.3_{(0.2)}$ | $59.5_{(0.4)}$ | $74.2_{(1.3)}$ | $77.0_{(1.9)}$ | $67.5_{(0.8)}$ |
| | Multi. | 12 | 1 | $83.8_{(2.9)}$ | $41.0_{(10.6)}$ | $51.2_{(0.8)}$ | $68.0_{(4.5)}$ | $58.3_{(11.1)}$ | $79.0_{(3.6)}$ | $56.0_{(8.1)}$ | $62.5_{(2.8)}$ |
| | Multi. | 6 | 2 | $85.3_{(1.9)}$ | $41.2_{(10.7)}$ | $51.2_{(1.3)}$ | $67.7_{(4.5)}$ | $57.7_{(10.8)}$ | $79.2_{(3.7)}$ | $55.8_{(7.9)}$ | $62.6_{(2.6)}$ |
| | Multi. | 3 | 4 | $83.3_{(2.5)}$ | $41.7_{(11.3)}$ | $51.0_{(1.1)}$ | $68.2_{(4.7)}$ | $57.7_{(10.8)}$ | $79.0_{(3.2)}$ | $56.0_{(8.1)}$ | $62.4_{(2.8)}$ |

Table 5: Few-shot ($k = 32$) results with different loss types, input formats, and layer-step configurations with a fixed compute budget, without calibration.

| Loss Type | Format | Layer | Step | Subj | AGNews | SST2 | CR | MR | MPQA | Amazon | Avg. |
|---|---|---|---|---|---|---|---|---|---|---|---|
| **Label** | Single | 12 | 1 | $63.3_{(0.2)}$ | $65.7_{(0.2)}$ | $71.3_{(0.6)}$ | $65.0_{(1.4)}$ | $70.7_{(0.9)}$ | $65.0_{(0.0)}$ | $76.7_{(0.2)}$ | $68.2_{(0.1)}$ |
| | Single | 6 | 2 | $63.5_{(0.0)}$ | $65.2_{(0.5)}$ | $73.3_{(1.3)}$ | $68.5_{(3.7)}$ | $71.3_{(0.2)}$ | $66.0_{(0.0)}$ | $77.5_{(0.4)}$ | $69.3_{(0.3)}$ |
| | Single | 3 | 4 | $64.2_{(0.2)}$ | $66.5_{(1.1)}$ | $73.2_{(0.6)}$ | $75.7_{(0.5)}$ | $72.0_{(0.0)}$ | $83.2_{(1.0)}$ | $78.0_{(0.4)}$ | $73.2_{(0.1)}$ |
| | Multi. | 12 | 1 | $64.5_{(7.8)}$ | $35.5_{(7.4)}$ | $56.8_{(9.7)}$ | $63.0_{(6.7)}$ | $58.7_{(8.9)}$ | $75.2_{(10.8)}$ | $62.2_{(8.3)}$ | $59.4_{(0.6)}$ |
| | Multi. | 6 | 2 | $77.7_{(7.0)}$ | $35.5_{(7.4)}$ | $57.0_{(9.9)}$ | $60.0_{(6.3)}$ | $52.3_{(2.1)}$ | $58.5_{(6.1)}$ | $55.8_{(7.9)}$ | $56.7_{(2.6)}$ |
| | Multi. | 3 | 4 | $67.5_{(11.5)}$ | $38.5_{(8.2)}$ | $55.3_{(5.2)}$ | $67.0_{(3.5)}$ | $61.0_{(8.0)}$ | $65.2_{(11.2)}$ | $62.5_{(8.9)}$ | $59.6_{(1.3)}$ |
| **Full-context** | Single | 12 | 1 | $65.5_{(1.1)}$ | $66.5_{(0.0)}$ | $70.7_{(0.2)}$ | $64.8_{(0.5)}$ | $72.0_{(1.4)}$ | $67.0_{(0.0)}$ | $76.5_{(0.0)}$ | $69.0_{(0.3)}$ |
| | Single | 6 | 2 | $64.7_{(0.6)}$ | $66.2_{(0.2)}$ | $71.2_{(0.2)}$ | $65.3_{(0.6)}$ | $71.5_{(0.4)}$ | $67.0_{(0.0)}$ | $76.7_{(0.0)}$ | $68.9_{(0.0)}$ |
| | Single | 3 | 4 | $64.2_{(0.2)}$ | $66.2_{(0.2)}$ | $71.3_{(0.2)}$ | $64.7_{(0.2)}$ | $71.0_{(0.0)}$ | $67.0_{(0.0)}$ | $76.5_{(0.0)}$ | $68.7_{(0.0)}$ |
| | Multi. | 12 | 1 | $62.2_{(7.5)}$ | $33.8_{(8.3)}$ | $52.2_{(3.1)}$ | $52.8_{(4.0)}$ | $50.8_{(1.2)}$ | $55.8_{(4.3)}$ | $55.3_{(7.2)}$ | $51.9_{(2.2)}$ |
| | Multi. | 6 | 2 | $60.0_{(5.5)}$ | $33.7_{(8.4)}$ | $50.8_{(1.2)}$ | $52.2_{(2.4)}$ | $50.2_{(0.2)}$ | $54.3_{(2.5)}$ | $55.0_{(6.7)}$ | $50.9_{(1.8)}$ |
| | Multi. | 3 | 4 | $58.7_{(4.9)}$ | $33.7_{(8.4)}$ | $50.8_{(1.2)}$ | $51.3_{(1.9)}$ | $50.0_{(0.0)}$ | $54.3_{(2.5)}$ | $55.3_{(7.2)}$ | $50.6_{(2.0)}$ |

although users are unable to directly modify deployed models, these models may still undergo dynamic updates while processing a context left-to-right, resulting in previously unseen behavior by the time the model reaches the end of the context. This has significant implications for the field of model alignment. It is challenging to impose restrictions on a model that can perform such dynamics updates internally, so malicious content can influence the output of deployed models.

Alternatively, we recognize the potential benefits of pre-training constructed models that integrate explicit fine-tuning mechanisms. By embedding the functionalities typically achieved through explicit fine-tuning, such as detecting malicious content and intent within the models themselves, the need for external modules can be mitigated. Pre-training the constructed model may offer a self-contained solution for ensuring safe and responsible language processing without relying on external dependencies.

