# OpenReview forum: "Trainable Transformer in Transformer"
_ICLR.cc/2024/Conference — Submitted to ICLR 2024_

### Official Review · Reviewer_LDni · 2023-10-28

**Soundness:** 2 fair
**Presentation:** 2 fair
**Contribution:** 2 fair
**Rating:** 5
**Confidence:** 2

**Summary:**

This paper provides a construction for a large transformer as a simulator, called TINT, such that it can simulate (on the auxiliary model / the smaller model) the forward pass, do back propagation to update the parameters, and then simulate another forward pass to output the final results. To improve the parameter efficiency, they provided the construction to approximate the layer norm, self-attention matrix, and their back-propagation process. Prior work focused on doing a forward pass one step of gradient descent on linear models or some much easier models, but they showed that actually a larger Transformer can simulate the forward and backward pass for another smaller Transformer. They showed that a GPT2 model can serve as a simulator to train a 125M OPT model in the forward  pass (of larger model) and showed that the perplexity can be improved by 0.3-0.7 on average.

This idea is very creative and worth investigating in the future. The theory result seems good and solid, but I still have some questions about the motivation and the theory part (see below). It is possible for me to change my score based on author's response and other reviewers' opinion.

**Strengths:**

1. The idea of doing forward ad backward pass of a Transformer in the single forward pass of another larger Transformer is creative.

2. They spent lots of efforts on improving the parameter efficiency of the larger Transformer (the simulator) by approximation the derivatives  of softmax-attention layer and layer norm, which is very good. The theory looks solid and correct, and the way to approximate linear Transformers using non-linear Transformers (especially the soft-max Transformers) is a topic that may be of separate interest of some community.

3. The experiments look good and there is a good improvement on the performance when trained this OPT model on larger GPT model.

**Weaknesses:**

1. The theory part, although the proof seems correct, it is better to have a theorem to include all results of approximating each part of a forward/backward pass of fine-tune a Transformer. For example, this should look like 'there is a transformer with XXX layers and XXX heads, and XXX dimensions such that, when you have an auxiliary model with XXX layers/heads/dimension and a sequence of tokens, it can approximation the objective function (something you want to approximate in the forward pass of the simulator) with an error of at most epsilon'. I think adding a result like this in your main result section (section 2) after the basic setup is essential.

The most important issue in the theorem above is that how many parameters you need in the simulator Transformers to approximate the in-context learn the fine-tuning objective of the auxiliary Transformer (with XX parameters). The proportion of the number of parameters (parameter size of big TF / param size of auxiliary one) matters since you claim that your construction is more parameter efficient than the construction in previous works [1,2].

Based on the proportion of parameter size, you can then determine that in the experiments, in order to train an OPT model with 125M params in a single forward pass of the simulator, what size model do you need to use as a simulator.

2. The structure of the paper, I believe, is not optimal, since you put lots of details of definitions and the results of approximating each part of TF in the main text, which I believe can be postponed to the appendix. This makes the paper harder to follow. I believe the authors may need to include more motivations and high-level description in the main text, as well as the the 'global theorem' I suggested in the first point. Then, the concrete way to approximating the back-propagation of soft-max and some details can be deferred in the appendix. I think the 'key component' section is especially hard to follow and I think maybe it will be slightly better to illustrate the structure of simulators in more detail in this subsection, instead of providing some formal definitions (like 2.3,2.6,2.7) in the main text (these are well-known definitions after all).

3. Why do you want to implement/approximate the forward/backward pass of Transformers in-context?

My understanding is that, you want to do something like 'in-context fine-tuning' for Transformers, because by doing this in-context, you do not need to 'do actual parameter updates' (in classical fine-tuning procedure, you need to literally 'update the parameters'). Then, the natural question is: which is the better and more efficient way for fine-tuning ----- in-context fine-tuning or direct fine-tuning?

In the paper, the authors did not show **whether it is more computationally efficient for in-context fine-tuning than direct fine-tuning**. By in-context fine-tuning, you need to do a forward pass in a much larger model; and by direct fine-tuning, you need to compute the gradient and update the parameters. Which one need a smaller number of computation (in terms of FLOPS or other metrics)?

4. Another question for the motivation: why do you use Transformers as the simulators? Are there any obvious reasons that make Transformer better than other architecture of Neural Networks here? Since what you want to do is to compute the forward/backward pass of the auxiliary Transformer in a single forward pass of the simulator model, which is a very complex task, why not considering other types of models as the outer simulator? I know that in some tasks that Transformers or attention models showed a better in-context learnability than CNN/RNN, but the tasks they considered are very much different from yours, so I am not sure whether this reason is sufficient enough.

[1] Ekin Akyurek, Dale Schuurmans, Jacob Andreas, Tengyu Ma, and Denny Zhou. What learning algorithm is in-context learning? investigations with linear models. arXiv preprint arXiv:2211.15661, 2022.
[2] Johannes von Oswald, Eyvind Niklasson, Ettore Randazzo, Jo˜ao Sacramento, Alexander Mordvintsev, Andrey Zhmoginov, and Max Vladymyrov. Transformers learn in-context by gradient descent, 2022.

**Questions:**

/

---

> ### Author Response · Authors · 2023-11-18
> **Rebuttal response**
>
> We thank the reviewer for appreciating the idea behind TINT and the effort to improve its parameter efficiency. We further appreciate the reviewer's suggestions to improve the paper's presentation. Please find our responses below.
>
> **A clear theoretical statement is missing. Something along the lines “there is a transformer with XXX layers and XXX heads, and XXX dimensions that can simulate GD on a small auxiliary transformer with error epsilon”**
>
> The difficulty with such a statement is that we have introduced a lot of approximations to be able to simulate gradient descent. These errors can compound in unpredictable ways that affect performance, so it is difficult to quantify the entire approximation error. We also note that people who use TINT may decide to remove an approximation and suffer the extra parameter cost to ensure a more faithful simulation of gradient descent. So, instead of providing a formal error bound, we instead use the experiments to validate that the approximations don’t hurt performance.
>
>
> **“The size of the simulator for different auxiliary models isn’t clear.”**
>
> In Table 3 in the appendix, we provided the sizes of different TINT architectures to simulate OPT models at different scales. We agree that this is an important figure to include in the main paper and we will move it in the subsequent version.
>
>
>
> **"Why do we want to approximate the forward/backward pass of Transformers in context?" “Unclear whether it is more computationally efficient to do in-context fine-tuning than direct fine-tuning. Comparisons between the two methods aren’t present.”**
>
> Our message behind TINT is two-fold:
>
> (a) The existence of TINT indicates that existing large-scale transformers may encode intricate sub-routines to solve complex tasks in-context.
>
> (b) We provide a framework that can explicitly encode a desirable implicit bias (i.e., the transformer should implicitly tune another smaller model) in the architecture. In particular, our framework shows that we can warm-start a large transformer model from a small transformer. The experiments further show that the large transformer improves over the small transformer at initialization.
>
> **Why do we use transformers as simulators and not any other neural network like CNN/RNN?**
>
> Transformers have demonstrated unprecedented success in solving language-related tasks with little to no additional gradient updates. TINT particularly studies the ability of the model to adapt to new inputs during inference (e.g., in-context learning). Such phenomena have not been commonly observed in CNNs or RNNs, so encoding such a tuning procedure in those architectures is not of as much interest. Ideas from TINT can inform the construction of CNN or RNN simulators.
>
>
> **Presentation**
> In the subsequent version, we will include some of the suggestions to improve the presentation of the paper. We will add more details to the simulator design in Figure 1. We will bring some of the details from the appendix to give more insights into the design of simulator modules for a linear layer (please refer to section B in the appendix which currently outlines the modules).

---

> > ### Author Response · Authors · 2023-11-22
> >
> > We hope that our response has addressed all your concerns. Please let us know if you have any more questions or feedback. We are happy to address them.

---

### Official Review · Reviewer_KkDE · 2023-11-02

**Soundness:** 3 good
**Presentation:** 1 poor
**Contribution:** 2 fair
**Rating:** 5
**Confidence:** 3

**Summary:**

This paper proposes a new construction, Transformers in Transformer (TinT), that allows a transformer to simulate and fine-tune transformer models during inference. The authors introduce innovative approximation techniques that allow TinT with less than 2 Billion parameters to simulate and fine-tune a 125 million-parameter transformer model. The authors conduct experiments to validate the internal fine-tuning procedure of TinT on various tasks.

**Strengths:**

1. Using a transformer to perform in-context-learning to fine-tune a transformer sounds like a quite fancy idea.
2. The authors performed experiments and demonstrated the effectiveness of TinT.

**Weaknesses:**

The writing of this paper is not completely clear, which makes it hard to understand the exact architecture of TinT. More specifically:
1. In Figure 1, it is not clear what are the dimensions of V_k, e_i, \partial y_j. Are there multiple back-propagation and gradient update steps in the TinT forward pass?
2. What is the meaning of notation $\partial$? There are many of them in Sections 2.5 and 2.6, but I don't understand the equations that contain this symbol.
3. What are the parameters of the auxiliary transformer, and what are the trainable parameters of the TinT?
4. I cannot see from the description of Section 2.7 PARAMETER SHARING IN THE TINT how the parameters of TinT are shared.
5. Since the architecture is unclear, I don't see where the $5 \times$, $H_{sim} \times$, $4 \times$ savings in Section 2.2 come from.

**Questions:**

1. Could the authors write down more concretely what is the architecture of the TinT?
2. Could the authors explain their experimental setup more clearly?

---

> ### Author Response · Authors · 2023-11-18
> **Rebuttal response**
>
> We thank the reviewer for appreciating the idea with the comment "Using a transformer to perform in-context-learning to fine-tune a transformer sounds like a quite fancy idea." Please find a summary of our work and our responses below.
>
> **Summary of our work**
>
> Please see Figure 1 and Section 2.1 for a straightforward illustration of the architecture of TINT. TINT is a transformer architecture that can simulate GD on an auxiliary transformer (e.g., GPT-2) during inference. The first few layers simulate the forward pass, the next few perform backpropagation while updating the parameters, and the final layers perform a forward pass using the updated parameters. The TINT architecture consists of 12 core operations that perform the forward, backward, and update operations on the 4 basic operations in a transformer: linear layer, self-attention, and layer normalization. The appendix contains all of the details and illustrations of how to simulate these 12 operations using standard transformer operations.
>
> Central to the novelty of TINT is the emphasis on the size of the resulting construction. While past works have required billions of parameters to tune even small MLPs, we show that a TINT model with fewer than two billion parameters can tune a 125M parameter auxiliary transformer (e.g., OPT-125m or GPT-2). TINT can be constructed so efficiently in part because of the approximations we make (see Section 2.2).
>
> So, we perform experiments to ensure that the approximations in the construction do not hurt the ability of TINT to tune an auxiliary model. These experiments, described in Section 3.1, quantify how well TINT can perform dynamic evaluation (Definition 3.1) and in-context learning.
>
>
> **In figure 1, what are the dimensions of v_k, e_i, and \partial y_j?**
>
> The size of the TINT models for different auxiliary models was in Table 3 in the appendix. We will move this table to the main paper in the final version.
>
> **Are there multiple backpropagations through the auxiliary model?**
>
> We can perform multiple gradient computations and updates for the auxiliary model. The size of the TINT model increases proportionally to the product of the number of auxiliary layers that we backpropagate through and the number of gradient steps.
>
> **What are the parameters of the auxiliary model and what are the trainable parameters of TINT?**
>
> The parameters of the auxiliary model are stored in the embedding layer of TINT. They are fed in as prefix tokens when necessary to TINT modules. The trainable parameters of TINT include all the parameters in the TINT modules and the embeddings (which include the auxiliary model parameters as well).
>
>
> **How are the parameters shared in TINT?**
>
> Let’s take the example of simulating forward propagation through an MLP layer in the auxiliary model (Appendix G). An MLP layer’s first layer dimension is 4 times the input dimension. However, it can be split into 4 sub-MLPs, whose first layer dimension equals the input dimension. Each sub-MLP is composed of 2 linear layers, separated by an activation layer. To simulate a sub-MLP, we can stack 3 TINT modules that simulate forward propagation through linear and activation layers. However, to simulate all 4 sub-MLPs, we use the same stack, with the contents of prefix tokens containing parameters of the 4 sub-MLPs in turns.
>
>
>
>
> **Notations: Meaning of $\partial$ unclear.**
>
> $\partial$ of a variable refers to the gradient of the loss function with respect to the variable (e.g., Definition 2.7). We will clarify this in the next version.

---

> > ### Author Response · Authors · 2023-11-22
> >
> > We hope that our response has addressed many of your concerns. Please let us know if you have any more questions or feedback. We are happy to address them.

---

### Official Review · Reviewer_k6Mj · 2023-11-06

**Soundness:** 3 good
**Presentation:** 3 good
**Contribution:** 3 good
**Rating:** 8
**Confidence:** 2

**Summary:**

This paper introduces an innovative and parameter-efficient transformer construction, TINT, capable of simulating complex models, such as pre-trained transformers, and applying fine-tuning in-context via a single forward pass.

**Strengths:**

1. The paper details a novel transformer construction that enables forward and backward operations, as well as parameter updates, within a single inference pass, and without the need for weight adjustments in the TINT model. This method surpasses previous approaches in terms of parameter efficiency and the complexity of models it can simulate.
2. The TINT architecture's efficiency is corroborated through real-data evaluations.

**Weaknesses:**

1. The TINT model requires access to auxiliary model weights, which it uses in prefix embeddings. This dependency differs from some previous works where the transformer independently and implicitly learns the auxiliary model's weights (and architecture) from the provided in-context dataset.
2. Despite its parameter efficiency, the TINT model's reliance on prefix embeddings to access auxiliary weights may lead to longer input sequence, which could potentially reduce computational and memory efficiency during inference due to the transformer's sequence length dependency.

**Questions:**

1. The capacity of the TINT structure in mimicking complex models like pre-trained transformers is well-documented, but is it equally adaptable to simpler auxiliary models, such as 2-layer MLPs?
2. Is the TINT's structure inherently dependent on the configuration of the auxiliary models it simulates, for instance in terms of number of layers, activation functions, or attention mechanisms?
3. The details of the Backward Module depicted in Figure 1 are unclear. Can you provide more insight into it, particularly regarding the connections between the Backward Module's inputs/outputs across layers and their relationship to the Forward Module's output?

---

> ### Author Response · Authors · 2023-11-20
> **Rebuttal response**
>
> We thank the reviewer for appreciating the idea behind fine-tuning via forward passes. Please find our responses below.
>
> **Is TINT adaptable to simpler auxiliary models like 2-layer MLPs?**
>
> Yes, the framework of TINT can be used to build a transformer to simulate any auxiliary model ranging from simple models like 2-layer MLP to the full transformer architecture. We create TINT modules for simulating forward, backward, and descent operations for 4 basic operations, linear, self-attention, layer norm, and activation. These can be composed according to the given auxiliary model.
>
> **Is TINT’s structure dependent on the configuration of auxiliary model?**
>
> Yes, TINT’s structure depends on the configuration of the auxiliary models. We first build TINT modules to simulate the forward, backward, and descent operations on basic operations in the auxiliary model. We then stack these TINT modules, according to the configuration of the auxiliary model. The width, depth, and other dimensions of the auxiliary model will factor into the size of TINT (please refer to Table 3 in the appendix).
>
> **Details of Backward module in Figure 1 are unclear.**
>
> Backpropagation uses the gradients of the (L+1)th layer to compute gradients with respect to the Lth layer. TINT does the same, using the activation of the previous TINT layer (i.e., the gradient of the (L+1)th layer) to compute the gradient with respect to the Lth layer. In order to save parameters, in the TINT architecture, the weight update for the (L+1)th layer is applied in parallel with the gradient computation of the Lth layer. Figure 4 and section B in the appendix [point to section]  illustrates this procedure on a linear layer.
>
>
> **"Prefix embeddings weren’t present in previous constructions, where the model learned gradient descent implicitly.”, “Computational overhead with sequence length."**
>
> We employ prefix embeddings to represent relevant auxiliary parameters for access in each layer (as defined in 2.1). This aligns closely with the concept of prefix-tuning in large language models [2]. This approach has been utilized in prior studies like [1] to construct transformers to simulate gradient descent on 2-layer MLPs. We minimize the increase in sequence length by stacking (section 2.4).
>
>
> An alternative solution was distributing auxiliary model parameters into TINT's token embeddings, eliminating the need for explicit prefix embeddings. However, this would have necessitated an increase in embedding dimension, resulting in a quadratic increase in parameters in TINT. So even though this solution could hint towards implicit gradient descent in large LLMs, it will be computationally expensive to implement and simulate this model. Exploring potential constructions that bypass the need for explicit prefix embeddings could be an interesting avenue for future research.
>
>
>
> 1:  Looped transformers as programmable computers. Giannou et al.'23
>
> 2: Prefix-Tuning: Optimizing Continuous Prompts for Generation. Li et al.'21

---

> > ### Author Response · Authors · 2023-11-22
> >
> > We hope that our response has addressed many of your concerns. Please let us know if you have any more questions or feedback. We are happy to address them.

---

### Official Review · Reviewer_zqc2 · 2023-11-10

**Soundness:** 3 good
**Presentation:** 2 fair
**Contribution:** 3 good
**Rating:** 5
**Confidence:** 2

**Summary:**

This work proposes TinT, a parameter-efficient construction to allow transformers to simulate forward and backward passes. Theoretically, the paper shows that TinT uses fewer parameters than prior construction based on the vanilla transformer architectures.  Empirically, this paper conducts experiments on language tasks to verify that TinT archives similar in-context learning performance compared to dynamic evaluation (which performs one GD step model update during inference) and larger models of a comparable parameter size.

**Strengths:**

1. New construction for simulating a backward pass within a model is provided, using fewer model parameters than prior constructions.
2. Experiments on more realistic language tasks are provided.
3. The idea of TinT might be useful in designing new language models for other applications.

**Weaknesses:**

1. Writtings could be improved in some places. For two examples,
* In definition 2.1, what are the "relevant" auxiliary model weights? The current definition is a bit difficult for me to interpret.
* In definition 2.3, are $p_t$'s referring to positional embedding? Could you explain why there aren't positional embeddings in definition 2.10.

2. Theorem 2.5 shows linear attention could be approximated by softmax attention. Can softmax attention also be approximated by linear attention? If not, I feel Theorem 2.5 alone does not suffice to justify the claim that "Thus, we often use linear attention in TINT". Let me know if I have misunderstood anything. In addition, is the claimed parameter saving based on linear attention or self-attention?

3. Definition 2.8 uses finite difference to approximate gradient. I am wondering if we can do this from end to end. That is, can we simulate a backward pass by doing finite-difference and two forward-pass? What's the disadvantage of doing so?

4. This work provides experiments on language tasks, while prior works provide experiments on simulated tasks (e.g., Akyurek et al 2022 did ICL for linear regression). So the empirical results are not directly comparable with prior works.

5. I feel an important prior work [1] is missed. Specifically, [1] also did approximation theory for ICL using transformers. How would the required number of parameters in the construction in this work compare to theirs?


[1] Bai, Yu, Fan Chen, Huan Wang, Caiming Xiong, and Song Mei. "Transformers as Statisticians: Provable In-Context Learning with In-Context Algorithm Selection." NeurIPS 2023

**Questions:**

See above.

---

> ### Author Response · Authors · 2023-11-18
> **Rebuttal response**
>
> We thank the reviewer for their comment on TINT and for pointing out the possible usefulness of TINT in designing new language models for other applications. Please find our responses below.
>
> **Issues in writing.**
>
> **a) Def 2.1: what are relevant auxiliary model weights?**
>
> **b) Def 2.3:  $p_t$ refer to position embeddings?**
>
> Each TINT layer is responsible for simulating a particular operation in the auxiliary model (e.g., the forward pass through a linear layer), which requires access to the auxiliary model weights. We provide the parameters relevant to that particular operation (e.g., the weight matrix of the linear layer) as prefix tokens to the layer. As a result, each layer in TINT is given a unique set of prefix tokens. This is similar to prefix-tuning [1], which adds a new prefix to each layer of the transformer.
>
> In def 2.3, p_t should have been $p_t^{\textrm{TINT}}$, defined as a set of one-hot positional encodings in TINT. $p_t^{\textrm{TINT}}$ has been defined above def 2.3. We slightly modify TINT's attention definition to include $p_t^{\textrm{TINT}}$  in the computation of attention scores. On the other hand, def 2.10 outlines the self-attention definition in the auxiliary transformer.
>
>
> **Why do we use Linear attention in TINT? Can any softmax attention be approximated by linear attention?**
>
> Linear attention modules are used to simulate the operations of linear layers in the auxiliary model. Theorem 2.5 shows that any linear attention can be simulated to arbitrary accuracy by a softmax attention module with a few additional parameters. So, our construction uses linear attention modules throughout to minimize the number of parameters in TINT, but one could instead use softmax attention modules without much degradation.
>
> **Def 2.8 uses finite difference to approximate gradient, why can’t we do the same end-to-end? That is, simulate backpropagation by finite difference with forward passes.**
>
> Classical analyses [1, 2] show that the variance of finite difference gradient estimates grows proportionally with the number of parameters, so if we use this method to train the entire model, we will likely have too noisy gradient estimate to successfully optimize. Recent work, dubbed MeZO [3], has shown that the end-to-end zeroth order optimization approach (i.e., multiple forward passes) can succeed in fine-tuning language models. However, MeZO relies on the inclusion of a meaningful prompt, and succeeds on relatively simple tasks. It is unknown if it can perform dynamic evaluation or in-context learning.
>
> **(Our) experiments aren’t directly comparable to Akyurek et al., since they conduct experiments on simulated tasks like linear regression.**
>
> Our goal is different from Akyurek et al., so their experimental setting would not make sense in our work. Our goal is to exhibit that TINT can tune a pre-trained language model (e.g., GPT-2 and OPT-125m) during inference. This allows us to demonstrate the complex behaviors that moderately sized transformers could exhibit on naturally occurring data. Our setting is very different from Akyurek et al. because their simulated models are simpler (e.g., linear regression) and trained on synthetic data. Our experiments are focused on validating the correctness of the approximations in our construction, not on how the in-context learning capability emerges during pre-training.
>
>
> **No comparisons to  Bai et al. who use approximation theory for ICL using transformers. How many parameters in the construction are necessary with the framework of Bai et al.?**
>
> The work of Bai et al. is complementary to ours, and we will add a discussion of it to our related works section. Our construction is more general than Bai et al. because it can implement backpropagation on many complex architectures. Also, while Bai et al. choose the number of layers in the construction to ensure the prediction error is below some threshold, we cannot straightforwardly do the same, since our construction involves a much more complicated learning algorithm (i.e., fine-tuning a transformer). On the other hand, TINT performs better on standard natural language tasks than the simple statistical algorithms in Bai et al. would.
>
> References:
>
> 1: Information-theoretic lower bounds on the oracle complexity of stochastic convex optimization. Agarwal et al.’12
>
> 2: Zeroth-Order Algorithms for Nonconvex-Strongly-Concave Minimax Problems with Improved Complexities. Wang et al.’22
>
> 3: Fine-tuning language models with just forward passes. Malladi et al. NeurIPS’23

---

> > ### Author Response · Authors · 2023-11-22
> >
> > We hope that our response has addressed many of your concerns. Please let us know if you have any more questions or feedback. We are happy to address them.

---

### Meta-Review · Area_Chair_kRPp · 2023-12-05

**Metareview:**

This paper proposes TinT, a modified transformer architecture that can efficiently simulate gradient descent over a smaller (auxiliary) transformer internally in its weights. The main theoretical result provides an explicit and parameter-efficient such construction, which can simulate the finetuning of a 125M transformer with less than 2B parameters. The paper then provides experiments to demonstrate the finetuning performance of TinT compared with standard finetuning.

The strengths of this paper are in the highly interesting conceptual message, the parameter efficiency of the construction, and the experiments on realistic tasks which demonstrate the power of the proposed TinT architecture. However, the reviewers have concerns about the soundness of the theory, such as the absence of a more quantitative global theorem (without which it may be the case that the error propagation is poor, or the extra assumptions needed in the pieces of the construction do not chain). Further, several reviewers raised concerns about the clarity of the presentation, which significantly hinders the digestibility of the new construction techniques. (I understand the authors focused on explaining the new modifications, but it may be good here to have some high-level explanations as the reviewers suggested.) Unfortunately, the concerns remain with the reviewers after rebuttal.

I encourage the authors to polish the presentation and incorporate the reviewers' suggestions in the next version. As a side question, the authors may also want to clarify some more details about the training of TinT in the experiments, i.e. "We compare a TINT model that tunes an OPT-125M pre-trained model internally to [XXX]" --- How many of the parameters of OPT125M were fed as input, in what format (a sequence of how many vectors in what dimension), and was TinT trained or hard-coded using the theoretical construction? I was not able to spot these in the experiment details (feel free to correct me if I was wrong), I believe these can be found by digging into the code, but I think it could be very helpful to provide these as well in the paper.

**Justification For Why Not Higher Score:**

While the paper explores an interesting idea, there are concerns about the soundness of the theory as well as the clarity of the presentation.

**Justification For Why Not Lower Score:**

N/A

---

### Decision · Program_Chairs · 2024-01-16

Reject